# Direct presentation of inflammation-associated self-antigens by thymic innate-like T cells induces elimination of autoreactive CD8+ thymocytes

Yuanyuan You[1,2,21], Josefine Dunst[1,2,21], Kewei Ye[1,2], Patrick A. Sandoz [3], Annika Reinhardt[1,2], Inga Sandrock[4], Natalia R. Comet[2,5], Rupak Dey Sarkar [6], Emily Yang [7], Estelle Duprez [8,9], Judith Agudo[10,11,12,13], Brian D. Brown [14,15,16], Paul J. Utz [7,17], Wolfgang Kastenmüller [6], Carmen Gerlach [2,5], Immo Prinz[4,18,19], Björn Önfelt [3,20] & Taras Kreslavsky [1,2] ✉

Upregulation of diverse self-antigens that constitute components of the inflammatory response overlaps spatially and temporally with the emergence of pathogen-derived foreign antigens. Therefore, discrimination between these inflammation-associated self-antigens and pathogen-derived molecules represents a unique challenge for the adaptive immune system. Here, we demonstrate that CD8+ T cell tolerance to T cell-derived inflammation-associated self-antigens is efficiently induced in the thymus and supported by redundancy in cell types expressing these molecules. In addition to thymic epithelial cells, this included thymic eosinophils and innate-like T cells, a population that expressed molecules characteristic for all major activated T cell subsets. We show that direct T cell-to-T cell antigen presentation by minute numbers of innate-like T cells was sufficient to eliminate autoreactive CD8+ thymocytes. Tolerance to such effector molecules was of critical importance, as its breach caused by decreased thymic abundance of a single model inflammation-associated self-antigen resulted in autoimmune elimination of an entire class of effector T cells.

The generation of a diverse antigen receptor repertoire through V(D) J recombination inevitably produces self-reactive receptors in some developing lymphocytes. To prevent autoimmunity, these self-reactive cells must be inactivated either at the site of development (central tolerance) or in secondary lymphoid organs (peripheral tolerance). Central tolerance requires the presence of antigens at the site of lymphocyte differentiation. In the thymus, several mechanisms ensure the breadth of thymocyte exposure to peripheral antigens. A wide range of tissue-restricted antigens show 'ectopic' expression in medullary thymic epithelial cells (mTECs)[1–3], and some mTECs induce large parts

of molecular programs of various peripheral cell types[4–7]. The thymus also harbors all major subsets of dendritic cells (DCs) that can present both endogenously expressed and TEC-derived antigens[8,9]. Migration of peripheral DCs into the thymus further expands the spectrum of tissue-specific antigens represented in the thymus[8,10]. Finally, another professional antigen-presenting cell (APC) type, thymic B cells, also contributes to the induction of T cell tolerance[11–13].

Self-reactive thymocytes are inactivated by negative selection[14], eviction of immature cells from the thymus[15] and diversion into lineages with regulatory or innate-like properties[16]. Differentiation of innate-like

**Fig. 1 | Thymic innate-like T cells express a broad spectrum of inflammation-associated self-antigens. a**, Expression of *Il4*, *Ifng* and *Gzmb* by mouse thymic cell subsets. Data are from the Immgen database (microarray dataset).

**b**, Uniform manifold approximation and projections (UMAPs) highlighting the expression of indicated genes in the mouse thymus cell atlas scRNA-seq dataset; AU, arbitrary units.

T cells represents the execution of one of the specialized effector programs (for example, type 1 helper T ($T_H$1) cell-, type 2 helper T ($T_H$2) cell- or interleukin-17 (IL-17)-producing helper T ($T_H$17) cell-like[17,18]) following antigen encounter in the thymus. These cells can express genes encoding inflammatory mediators and cytotoxic molecules at steady state[18–20] and often home to barrier tissues, where they contribute to the first line of defense against pathogens and regulate tissue homeostasis[21,22].

However, many innate-like T cells found in the thymus, including a large fraction of γδT cells and the majority of invariant natural killer T (iNKT) cells, are not newly developed immature cells but mature tissue-resident lymphocytes[23,24]. Inflammatory mediators secreted by these cells have a broad impact on other components of the thymic microenvironment. For example, type 2 cytokines derived from innate-like cells support the intrathymic generation of innate-like

interferon-γ (IFNγ)-producing CD8+ T cells[25,26], regulate the recruitment of thymic eosinophils[27], affect TEC function[28] and induce the activation of thymic DCs[29]. Intriguingly, other inflammatory pathways, for example, type 1/type 3 IFN signaling and MyD88-dependent signaling in TECs[30,31], are also constitutively activated in the thymus. Therefore, the thymus has evolved to harbor a complex constitutively active inflammatory network, and thymus-resident innate-like T cells represent one of its central nodes. The reasons for the activation of numerous inflammatory pathways in the organ where pathogen encounter is unlikely remain unclear.

If autoreactive T cells escape central tolerance and encounter antigen in secondary lymphoid organs in the absence of inflammation, they are inactivated by peripheral tolerance mechanisms[32]. Mature T lymphocytes will only undergo clonal expansion and differentiation into effector cells if they have received co-stimulatory signals that are provided by APCs only in the context of inflammation. The induction of T cell effector programs whose components constitute self-antigens therefore coincides both spatially and temporally with the presence of a pathogen. Breach of tolerance to such antigens would result in a 'fratricide' reaction that would curtail any further immune response involving expression of this molecule. It remains unclear how the immune system discriminates such inflammation-associated self-antigens from antigens originating from the pathogen itself.

In this study, we aimed to investigate the mechanisms of tolerance to T cell-derived inflammation-associated self-antigens. Using model inflammation-associated self-antigens, we demonstrate that CD8+ T cell tolerance to these molecules can be induced in the thymus. The high efficiency of this tolerance was ensured by multiple redundant cellular sources of these antigens in the thymus, which, in addition to TECs, included thymic innate-like T cells and thymic eosinophils. Expression of a self-antigen by minute numbers of thymic innate-like T cells was sufficient to mediate remarkably efficient elimination of autoreactive CD8+ thymocytes that occurred through direct T cell-to-T cell antigen presentation. These results suggest that constitutive expression of inflammation-associated molecules in the thymus may have evolved to ensure the induction of tolerance to this unique class of self-antigens.

## Results

### Innate-like T cells as libraries of T cell effector antigens

To start unraveling mechanisms of tolerance induction to T cell-derived inflammation-associated self-antigens, we analyzed the expression of a selection of genes encoding T cell effector molecules (*Il4*, *Il13*, *Ifng*, *Il17a*, *Il17f* and *Gzmb*) and their upstream regulators (*Eomes* and *Tbx21*) in thymic cell subsets. Analysis of bulk transcriptomics data from thymic cell types available in the Immunological Genome (Immgen) database[33,34] (Fig. 1a) and in the integrated single-cell RNA-sequencing (scRNA-seq) mouse thymus atlas[6,35,36] (Fig. 1b) demonstrated that professional thymic APCs, such as TECs

and thymic DCs, express no or low levels of many of these molecules. Instead, the highest levels of expression of most of these molecules in the thymus overlapped with the expression of *Zbtb16*, the gene encoding the transcription factor PLZF that orchestrates effector programs of many innate-like T cell populations[37–40] (Fig. 1b). Analysis of the Immgen dataset confirmed predominant expression of effector T cell molecules in iNKT and γδT cell subsets (Fig. 1a). This pattern of expression was also conserved in humans as determined by analysis of data from the human scRNA-seq thymus atlas[35] (Extended Data Fig. 1). Thus, in contrast to professional APCs, thymic innate-like T cells express high levels of effector molecules characteristic of all major effector T cell subsets and would represent perfect libraries of T cell effector program antigens.

### CD8+ T cell tolerance to a model inflammation self-antigen

To start characterizing tolerance to inflammation-associated self-antigens, we first took advantage of the predominant expression of *Il4* by innate-like T cells in the mouse thymus (Fig. 1a,b) and exploited green fluorescent protein (GFP) in *Il4*-IRES-GFP mice (referred to hereafter as IL-4−GFP mice)[41] as a generic model of antigen expressed by thymic innate-like T cells. Immunization with a major histocompatibility complex (MHC) class I (H2-K[d]) GFP$_{200-208}$ epitope[42] resulted in a prominent CD8+ T cell response in wild-type (WT) mice, but no response was detected in IL-4−GFP animals (Fig. 2a), indicating a remarkably efficient CD8+ T cell tolerance to GFP in this system. By contrast, immunization with an MHC class II (I-A[b]) GFP$_{81-95}$ epitope[43] resulted in comparable CD4+ T cell responses in WT and IL-4−GFP mice (Fig. 2b). Thus, expression of IL-4−GFP resulted in strong CD8+, but not CD4+, T cell tolerance.

Next, to assess the contribution of hematopoietic and stromal cells to the induction of CD8+ T cell tolerance, we established WT → WT, IL-4−GFP → WT, WT → IL-4−GFP and IL-4−GFP → IL-4−GFP bone marrow (BM) chimeras and immunized them with the GFP$_{200-208}$ peptide. A clear expansion of GFP-specific CD8+ T cells was observed only in the WT → WT group, whereas the frequency of tetramer+ cells in all other groups was similar to that in unimmunized controls (Fig. 2c). Together, these results reveal efficient induction of CD8+ T cell tolerance to a model inflammation-associated self-antigen, with a redundant role of hematopoietic and radioresistant cells in this process.

### Central tolerance to a model inflammation self-antigen

We next sought to assess if tolerance to IL-4−GFP was already established in the thymus. Analysis of IL-4−GFP expression revealed that only $0.5 \pm 0.2\%$ (mean ± s.d.) of hematopoietic cells in the thymus were GFP+, of which $76.9 \pm 3.3\%$ were innate-like T cells and $17.7 \pm 3.4\%$ represented thymic eosinophils (Fig. 3a and Extended Data Fig. 2a,b). GFP expression was also detectable in $0.8 \pm 0.2\%$ of mTECs (Extended Data Fig. 2c). Immunofluorescence microscopy revealed that GFP+SiglecF+

**Fig. 2 | Redundant role of radioresistant and radiosensitive cells in the induction of CD8+ T cell tolerance to a model inflammation-associated self-antigen. a**, WT and IL-4−GFP mice on a C57BL/6J × BALB/c F1 (CB6F1) genetic background were immunized intraperitoneally with GFP$_{200-208}$ peptide using anti-CD40 and poly(I:C) as adjuvant or were left unimmunized (WT only). Six days after immunization, splenic H2-K[d] GFP$_{200-208}$ tetramer-binding CD8+ T cells were quantified by flow cytometry. Representative flow cytometry plots gated on CD8+CD44+ T cells (left) and quantification of frequencies (from total CD8+ T cells) and absolute numbers (right) are shown (*n* = 5 mice per group). The data are representative of four independent experiments. **b**, WT and IL-4−GFP mice on a CB6F1 genetic background were immunized subcutaneously with GFP$_{81-95}$ peptide in complete Freund's adjuvant or were left unimmunized (WT only). Fourteen days after immunization, I-A[b] GFP$_{81-95}$ tetramer-binding cells were magnetically enriched from pooled spleens and lymph nodes and quantified by flow cytometry. Representative flow cytometry plots gated on CD4+ T cells (left) and absolute numbers of CD4+CD44+tetramer+ T cells (right)

are shown (*n* = 2 mice for nonimmunized, *n* = 5 for the other groups). Data are representative of two independent experiments; LN, lymph node. **c**, WT and IL-4−GFP mice (on a CB6F1 background) were lethally irradiated and reconstituted with syngeneic WT or IL-4−GFP BM cells depleted of T and NK cells. The resulting four groups of BM chimeras were immunized with GFP$_{200-208}$ peptide ≥7 weeks after reconstitution and analyzed 6 days after immunization, as described in **a**. WT CB6F1 mice were used as an unimmunized control. Representative plots demonstrating tetramer binding by CD8+CD44+ T cells (left) and quantification of the frequency of CD8+CD44+tetramer+ cells from total CD8+ T cells and their absolute numbers (right) are shown (*n* = 6 mice for nonimmunized, *n* = 7 for WT → WT, *n* = 7 for IL-4−GFP → WT, *n* = 5 for WT → IL-4−GFP and *n* = 6 for IL-4−GFP → IL-4−GFP). Results are pooled from two independent experiments. Data are presented as mean ± s.d. (NS, not significant (*P* > 0.05); **P* < 0.05 and *****P* < 0.0001) and were analyzed by two-tailed Student's *t*-test (**b**) or one-way analysis of variance (ANOVA) with a Holm−Sidak multiple comparisons test (**a** and **c**).

eosinophils and GFP⁺SiglecF⁻ innate-like lymphocytes were present both in the thymic cortex and medulla, with more GFP⁺ lymphocytes located in the medulla (Fig. 3b and Extended Data Fig. 2d).

We next tested if the expression of IL-4–GFP by hematopoietic and/or stromal cells would result in the inactivation of autoreactive CD8⁺ T cells already in the thymus. We took advantage of Jedi T cell antigen receptor-αβ (Jedi-TCRαβ) mice, in which T cells express pre-rearranged chains of the H2-Kᵈ GFP$_{200–208}$-specific TCR from the endogenous *Tcra* and *Tcrb* loci[44]. We generated BM chimeras by transferring a mix of Jedi-TCRαβ and WT or Jedi-TCRαβ and IL-4–GFP BM cells into

WT or IL-4–GFP recipients and analyzed T cell development in these four groups of BM chimeras. As expected, in the absence of IL-4–GFP, the Jedi thymocyte compartment in WT + Jedi-TCRαβ → WT chimeras included CD4/CD8-double-negative (DN), CD4/CD8-double-positive (DP) and CD8-single-positive (CD8SP) cells (Fig. 3c). Strikingly, expression of IL-4–GFP solely by thymic hematopoietic cells in the IL-4–GFP + Jedi-TCRαβ → WT group or by radioresistant cells in the WT + Jedi-TCRαβ → IL-4–GFP group in both cases resulted in a virtually complete elimination of GFP-specific CD8SP thymocytes (Fig. 3c). Residual DP thymocytes in chimeras with IL-4–GFP donors and/or

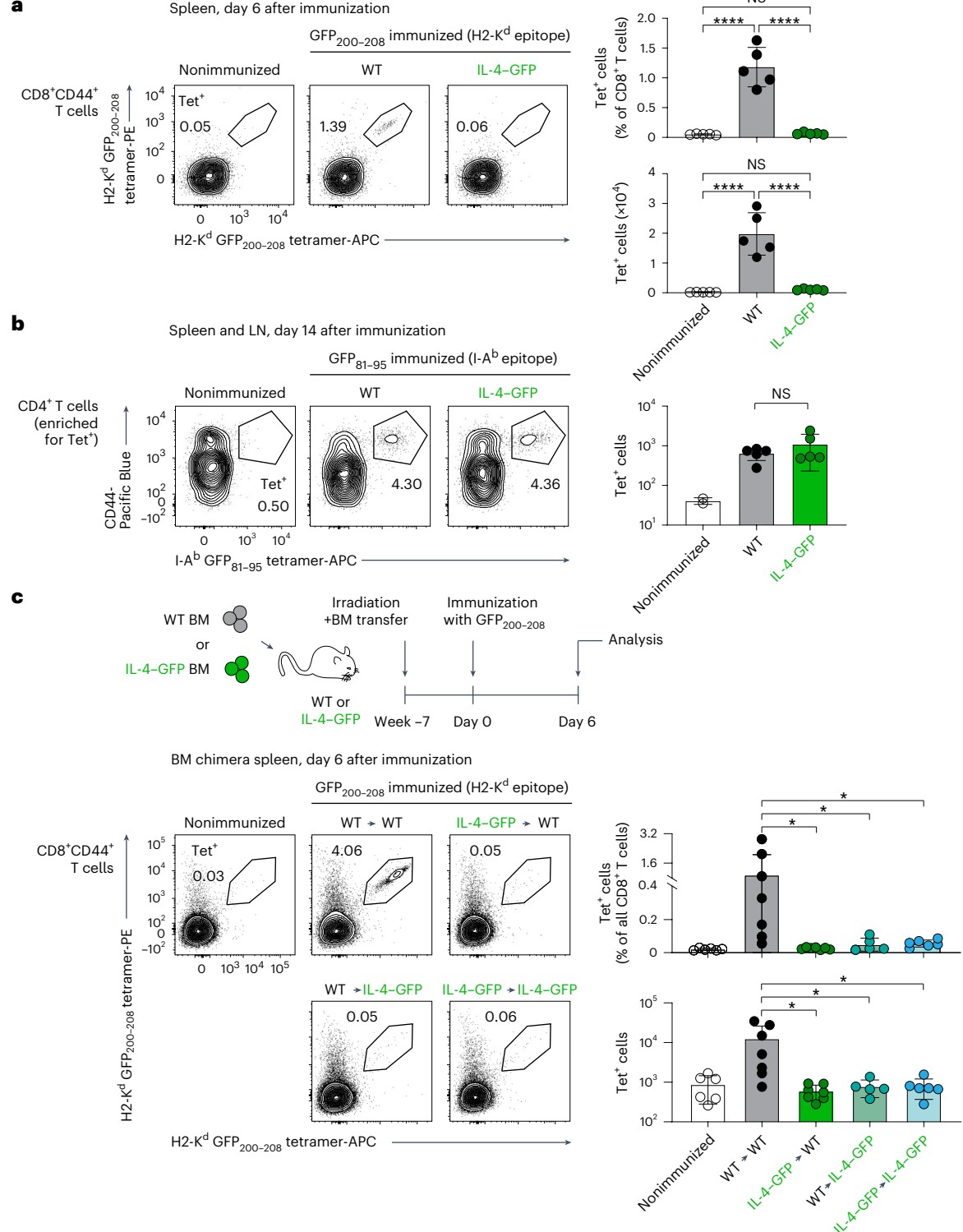

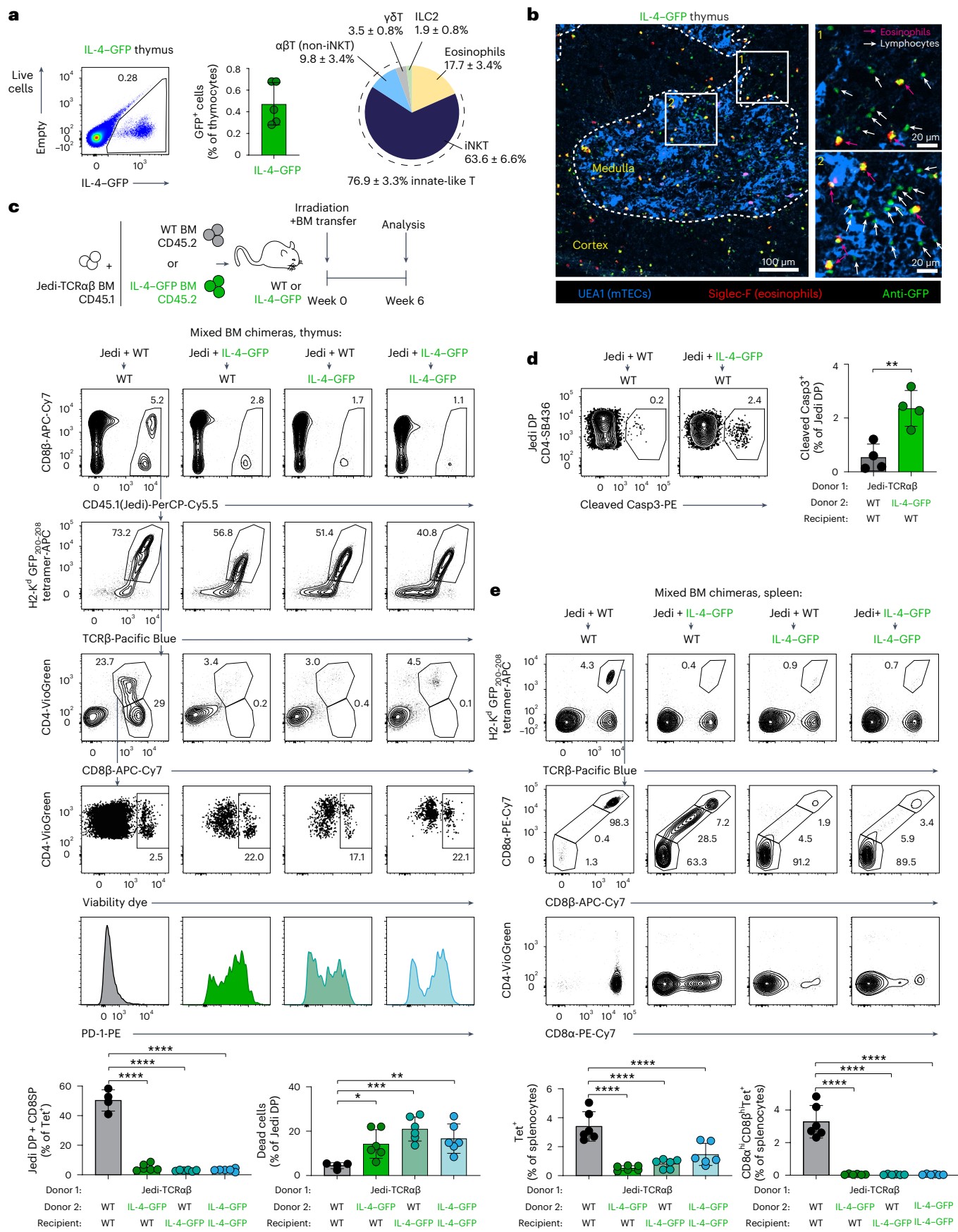

**Fig. 3 | Efficient central CD8⁺ T cell tolerance to a model inflammation self-antigen. a**, GFP expression by thymic nonstromal populations in IL-4–GFP mice. Frequency of GFP⁺ cells ($n = 5$ mice) and pie chart showing frequencies of different subsets within the GFP⁺ compartment (average percentage ± s.d.; $n = 3$ mice). See also Extended Data Fig. 2a–c. **b**, Images of IL-4–GFP thymus sections as analyzed by confocal immunofluorescence microscopy. The medulla was identified by staining with UEA1 lectin (blue). IL-4–GFP⁺ eosinophils were distinguished from IL-4–GFP⁺ lymphocytes (green) by co-staining with anti-Siglec-F (red). See also Extended Data Fig. 2d. **c**, WT and IL-4–GFP mice (on a BALB/c (H2ᵈ) background; CD45.2) were lethally irradiated and reconstituted with syngeneic WT or IL-4–GFP BM cells mixed with BM cells from Jedi mice (B10.D2 genetic background (H2ᵈ); CD45.1). Thymi of the chimeras were analyzed 6 weeks after reconstitution ($n = 4$ chimeras for analysis of thymi in WT + Jedi-TCRαβ → WT; $n = 6$ for the other groups). Jedi thymocytes were gated as CD45.1⁺tetramer⁺ cells. Expression of CD4 and CD8β on CD45.1⁺tetramer⁺ cells and viability dye and PD-1 staining of CD45.1⁺tetramer⁺CD4⁺CD8β⁺ (Jedi DP) cells

are shown. The frequencies of CD4⁺CD8β⁺ (DP) and CD4⁻CD8β⁺ (CD8SP) cells of CD45.1⁺tetramer⁺ Jedi thymocytes and the frequencies of dead Jedi DP cells were quantified (bottom). Data are representative of three independent experiments; Tet, tetramer. **d**, Analysis of the frequency of cleaved caspase-3⁺ cells among tetramer⁺CD4⁺CD8β⁺ (Jedi DP) cells in WT + Jedi-TCRαβ → WT and IL-4–GFP + Jedi-TCRαβ → WT mixed BM chimeras (from the experiment shown in Supplementary Fig. 2). Data are from one experiment with four chimeras per group; Casp3, caspase-3. **e**, Representative flow cytometric analysis of splenocytes from four groups of BM chimeras (same experiment as in **c**). Jedi cells were identified by H2-Kᵈ GFP₂₀₀–₂₀₈ tetramer staining, and CD8α, CD8β and CD4 expression on these cells was analyzed. The frequencies of total and CD8αʰⁱCD8βʰⁱ tetramer-binding cells among splenocytes were quantified (bottom). Data are presented as mean ± s.d. (*$P < 0.05$, **$P < 0.01$, ***$P < 0.001$ and ****$P < 0.0001$) and were analyzed by two-tailed Student's $t$-test (**d**) or one-way ANOVA with a Holm–Sidak multiple comparisons test (**c** and **e**).

recipients upregulated PD-1 expression and contained a large fraction of dying cells (Fig. 3c) and cells positive for active caspase-3 (Fig. 3d), indicating that these autoreactive cells were eliminated by negative selection. Strikingly, in IL-4–GFP + Jedi-TCRαβ → WT chimeras, this elimination of autoreactive thymocytes was mediated by as few as $0.1 ± 0.02\%$ of GFP⁺ hematopoietic cells present in the thymi of these mice (Extended Data Fig. 2e). By contrast, WT + Jedi-TCRαβ → IL-4–GFP chimeras contained, on average, $0.01\%$ of residual GFP⁺ hematopoietic cells, and some had virtually no detectable GFP⁺ thymocytes, indicating that, in this setting, induction of tolerance is likely to be mediated solely by thymic stroma (Extended Data Fig. 2e). The disappearance of Jedi CD8⁺ thymocytes was reflected in the periphery by an overall decrease in Jedi-TCRαβ cells and a near complete loss of GFP-specific conventional CD8⁺ T cells in the spleen (Fig. 3e). As the TCR in Jedi-TCRαβ mice was expressed prematurely at the DN stages of T cell development (Fig. 3c), we performed a similar analysis with Jedi-TCRβ mice that only carry the TCRβ chain of the Jedi TCR and thus rely on recombination of the endogenous *Tcra* loci at the DP stage of T cell development. A small distinct population of Jedi-TCRβ GFP-specific CD8⁺ T cells was completely eliminated in the presence of antigen (see Supplementary Note 1 and Supplementary Figs. 1 and 2). We conclude that most CD8⁺ T cells that recognize IL-4–GFP are inactivated already in the thymus, and TECs and hematopoietic cells play redundant roles in the induction of central tolerance to this inflammation-associated self-antigen.

### iNKT cells and eosinophils can mediate negative selection

As the thymic GFP⁺ population in IL-4–GFP mice, although dominated by innate-like T lymphocytes, represent a complex mix of cells (Fig. 3a), we next aimed to test if antigen expression solely by innate-like T cells was sufficient to induce tolerance. To this end, we established reaggregate thymus organ cultures (RTOCs; Fig. 4a) with small numbers of Jedi DP

thymocytes and iNKT cells sorted from WT or IL-4–GFP thymi. The presence of as few as $0.2\%$ of GFP⁺ iNKT cells in these thymus organoids resulted in a near complete deletion of Jedi cells with few remaining cells acquiring the CD4⁻ CD8αˡᵒCD8βˡᵒPD-1ʰⁱ cell-surface phenotype (Fig. 4b). Thus, antigen expression by minute numbers of innate-like T cells is sufficient to eliminate autoreactive thymocytes in the RTOC system.

Of note, this highly efficient induction of negative selection was not a unique property of iNKT cells, as small numbers of conventional ACTB–GFP-transgenic CD4SP thymocytes and thymic eosinophils from IL-4–GFP mice likewise induced elimination of Jedi-TCRαβ DP thymocytes in RTOCs (Extended Data Fig. 3a–c). These results suggest that in addition to professional thymic APCs, such as TECs, negative selection of autoreactive CD8⁺ thymocytes can be induced by expression of antigen in a wide range of 'amateur' APC cell types, including developing conventional T lymphocytes, thymic innate-like T cells and thymic eosinophils.

### Effects of co-stimulation, antigen localization and affinity

As co-stimulation by CD80/CD86 can be important for effective negative selection for some, but not all, TCR specificities[45–49], we next assessed the expression of these molecules by thymic eosinophils and innate-like T cells (Extended Data Fig. 4a). Thymic eosinophils expressed high levels of CD80 but not CD86, whereas a large fraction of thymic iNKT cells and some γδT cells displayed both co-stimulatory molecules on their surface. As data from the Immgen database[20,34] suggest that iNKT cells express little if any *Cd80* or *Cd86* mRNA, and as T cells often acquire CD80 and CD86 from APCs through trogocytosis[50,51], it seems likely that iNKT cells acquire these molecules from professional thymic APCs. Addition of anti-CD80/CD86 to RTOCs containing Jedi-TCRαβ DP thymocytes and WT or IL-4–GFP iNKT cells had no effect on negative selection (Extended Data Fig. 4b), suggesting that co-stimulation might be dispensable for negative selection in the case

**Fig. 4 | Antigen expression exclusively by small numbers of iNKT cells is sufficient to induce elimination of autoreactive CD8⁺ thymocytes.
a**, Schematic representation of the RTOC experiments shown in **b**–**d**. Jedi-TCRαβ and OT-I DP thymocytes were sorted as CD4⁺CD8⁺ cells, and ex vivo iNKT cells were sorted from the thymi of WT and IL-4–GFP mice as CD1d tetramer-binding cells (without additional gating on GFP). Transduced thymic iNKT cells were sorted as Thy1.1⁺ (**c**) or GFP⁺ (**d**) CD1d tetramer-binding cells; E14.5–15.5, embryonic days 14.5–15.5. **b**, Flow cytometric analysis of RTOCs with ex vivo thymic WT or IL-4–GFP iNKT cells 5 days after initiation of reaggregation. Representative plots show the distribution of Jedi cells (CD45.1⁺), iNKT cells (CD45.2⁺) and fetal thymocytes (CD45.1⁺CD45.2⁺), frequencies of GFP-expressing cells, frequencies of H2-Kᵈ GFP₂₀₀–₂₀₈ tetramer-binding Jedi thymocytes (quantification is shown on the right) and expression of the indicated cell-surface markers by the latter cells. Results are pooled from three independent experiments ($n = 4$ RTOCs for WT iNKT cells and $n = 7$ for IL-4–GFP iNKT cells).

**c**, Thymic iNKT cells were transduced with Thy1.1-IRES-i.c.GFP, Thy1.1-IRES-secGFP or Thy1.1-only retroviruses. Sorted Thy1.1⁺ iNKT cells were added to RTOCs together with Jedi-TCRαβ DP thymocytes. Flow cytometric analysis and quantification is shown as in **b** ($n = 5$ RTOCs for empty vector, $n = 10$ for i.c.GFP and $n = 9$ for secGFP). Results are pooled from two independent experiments. **d**, Thymic iNKT cells were transduced with cOVA-IRES-GFP retroviruses encoding the indicated versions of the OVA₂₅₇–₂₆₄ epitope. Sorted GFP⁺ iNKT cells were added to RTOCs together with OT-I DP thymocytes. RTOCs with nontransduced iNKT cells were used as a no-antigen control. Flow cytometric analysis and quantification is shown as in **b** ($n = 6$ RTOCs for uninfected, $n = 3$ for N4, $n = 6$ for Q4, $n = 3$ for T4 and $n = 5$ for V4). Representative results from two independent experiments are shown. Data are presented as mean ± s.d. (*$P < 0.05$, **$P < 0.01$, ***$P < 0.001$ and ****$P < 0.0001$) and were analyzed by two-tailed Student's $t$-test (**b**) or one-way ANOVA with a Holm–Sidak multiple comparisons test (**c** and **d**).

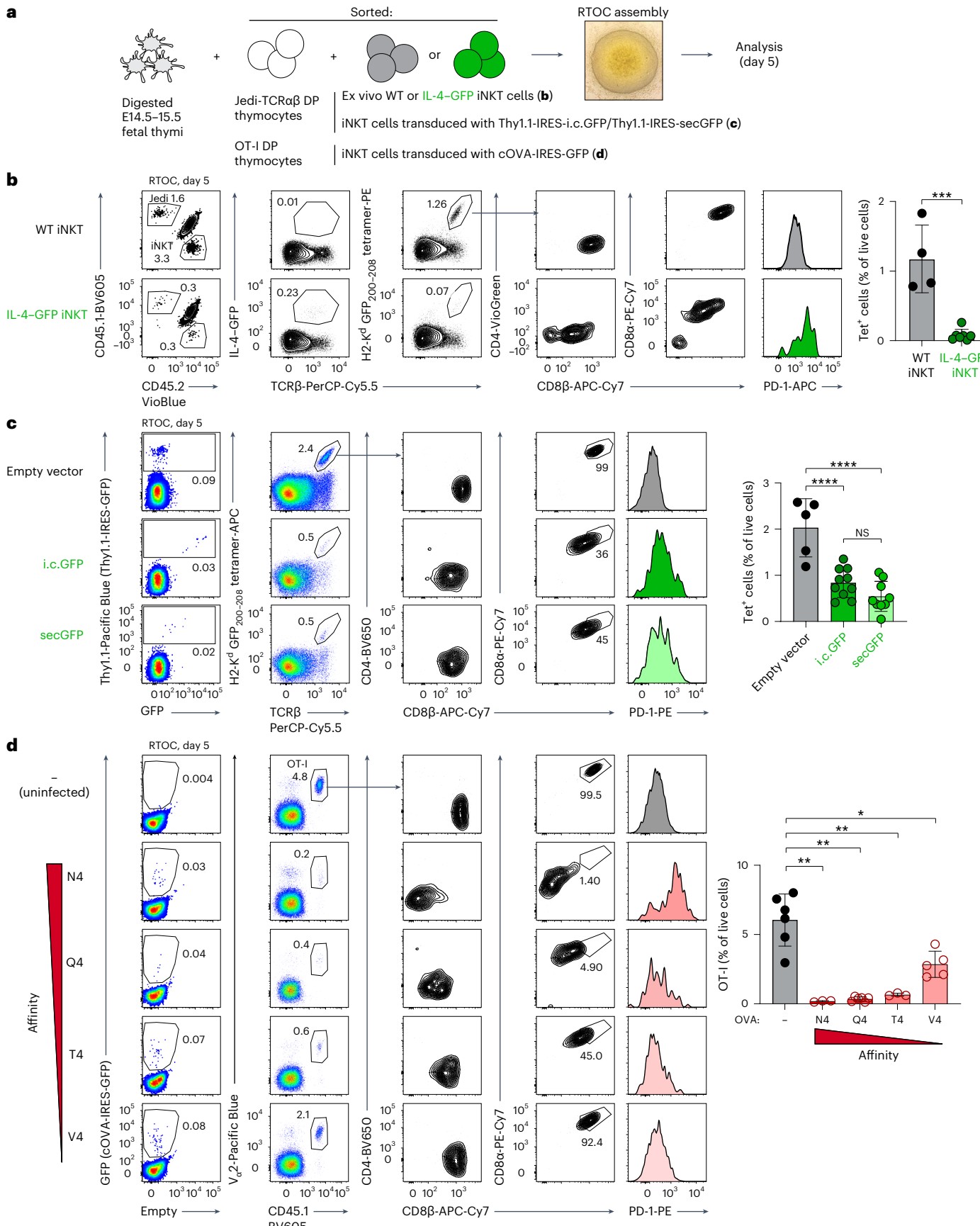

of Jedi thymocytes. However, as the co-stimulation requirement for negative selection is not absolute and depends on the epitope[45], and as thymic innate-like T cells are equipped with CD80 and CD86, it is conceivable that co-stimulation could contribute to the induction of tolerance to some self-antigens expressed by these cells.

As many endogenous inflammation-associated self-antigens are secreted molecules, we next tested if innate-like T cells can induce tolerance to a secreted antigen. To this end, we generated retroviral constructs encoding intracellular (i.c.GFP) and secreted (secGFP) versions of GFP and confirmed efficient GFP secretion in the latter setting (Extended Data Fig. 4c). We next transduced thymic iNKT cells with Thy1.1-IRES-i.c.GFP, Thy1.1-IRES-secGFP and Thy1.1-only retroviruses, sorted Thy1.1+ iNKT cells and added them to RTOCs together with Jedi-TCRαβ DP thymocytes. A comparable loss of Jedi cells was observed with i.c.GFP and secGFP constructs, and the remaining cells upregulated PD-1 expression and downregulated CD8 expression (Fig. 4c).

Finally, to test how antigen affinity affects negative selection, we took advantage of altered peptide ligands described for the OT-I TCR. It was shown that Q4, T4 and V4 substitutions in the SIINFEKL epitope (N4) reduced OT-I CD8+ T cell reactivity by 20-, 70- and 700-fold, respectively[52]. We therefore retrovirally transduced thymic iNKT cells with constructs encoding cytoplasmic ovalbumin (cOVA)-IRES-GFP containing these epitopes and added GFP+ iNKT cells to RTOCs together with sorted OT-I DP thymocytes. N4-, Q4- and T4-expressing iNKT cells induced a near complete elimination of OT-I thymocytes (33-, 18- and 9-fold decreases, respectively), and even very low-affinity V4 epitope was sufficient to induce elimination of about half of the OT-I thymocytes (Fig. 4d).

We conclude that expression of either intracellular or secreted antigens by innate-like T cells can be sufficient to induce elimination of autoreactive CD8+ thymocytes and that this elimination can take place across a wide range of affinities.

## Antigen expression by iNKT cells induces tolerance in vivo

We next sought to test if antigen expression by innate-like T cells is also sufficient to mediate tolerance induction in vivo. As a first approach to this question, we used IL-17A–GFP mice in which, in contrast to the IL-4–GFP system, GFP expression in the thymic hematopoietic compartment was restricted to innate and innate-like lymphocytes (Fig. 5a and Extended Data Fig. 5a,b). Analysis of stromal populations in these mice also revealed GFP expression in a small subset of mTECs (Extended Data Fig. 5c). Therefore, to dissect the roles of GFP-expressing stromal and hematopoietic populations, we established WT → WT, IL-17A–GFP → WT, WT → IL-17A–GFP and IL-17A–GFP → IL-17A–GFP BM chimeras and immunized these mice with the GFP200–208 peptide. Fully mirroring the results obtained in the IL-4–GFP system (Fig. 2), IL-17A–GFP expression either by radioresistant or radiosensitive cells was sufficient to abrogate CD8+ T cell responses to GFP200–208 (Fig. 5b), whereas no CD4+ T cell tolerance

to GFP81–95 MHC class II epitope immunization was observed in IL-17A–GFP mice (Extended Data Fig. 5d). Unexpectedly, GFP-expressing cells were very poorly reconstituted in the thymi of IL-17A–GFP → WT BM chimeras (0.009 ± 0.003% GFP+ cells of total thymocytes in IL-17A–GFP → WT BM chimeras versus 0.06 ± 0.04% in steady-state IL-17A–GFP mice and 0.5 ± 0.2% in IL-4–GFP mice; Extended Data Fig. 5e). This near absence of GFP-expressing cells in the thymi of IL-17A–GFP BM chimeras precluded us from assessing negative selection of supraphysiological numbers of autoreactive GFP-specific thymocytes in Jedi:IL-17A–GFP mixed BM chimeras, a setting that in fact resulted in breach of tolerance (see below). As an alternative approach, we therefore turned to RTOC experiments, which demonstrated that iNKT cells from IL-17A–GFP mice can induce efficient elimination of Jedi-TCRαβ DP thymocytes (Fig. 5c).

The immunization experiments in IL-17A–GFP → WT BM chimeras described above showed that expression of a model inflammation-associated self-antigen solely by innate-like lymphocytes is sufficient to induce tolerance in vivo, and RTOC experiments suggested that it can happen through negative selection of autoreactive thymocytes. To directly test if exclusive antigen expression by thymic innate-like T cells is sufficient to induce inactivation of autoreactive CD8+ thymocytes in vivo, we injected sorted thymic IL-4–GFP iNKT cells into the thymi of Jedi-TCRβ mice, which, before injection, contained 0.007 ± 0.003% GFP-specific thymocytes (Fig. 5d). GFP+ cells were detectable in the thymi of the recipient mice 7 days after transfer (Fig. 5d), but their frequency was much lower than that in IL-4–GFP mice (Fig. 3a) and similar to what was observed in steady-state IL-17A–GFP mice (Fig. 5a). The presence of these minute numbers of GFP+ iNKT cells resulted in a decreased frequency of GFP-specific thymocytes and an acquisition of a CD4−CD8αloCD8βloPD-1hi cell-surface phenotype by the remaining GFP-specific cells (Fig. 5d). Thus, antigen expression exclusively by small numbers of thymic innate-like T cells is sufficient to induce elimination of autoreactive CD8+ thymocytes in vivo.

## T cell-to-T cell antigen presentation mediates negative selection

The efficient negative selection of autoreactive thymocytes by small numbers of antigen-expressing innate-like T cells suggested possible cross-presentation by thymic professional APCs. To start exploring this possibility, we sorted DCs, iNKT cells and eosinophils from the thymi of WT, IL-4–GFP and IL-17A–GFP mice and cocultured them with an NFAT reporter cell line[53] expressing the Jedi TCR. Reporter expression was upregulated in cocultures with iNKT cells from IL-4–GFP and IL-17A–GFP mice and eosinophils from IL-4–GFP mice but not in cocultures with DCs (Extended Data Fig. 6a, note the higher baseline reporter expression in all DC cocultures), suggesting that IL-4–GFP and IL-17A–GFP may not undergo efficient cross-presentation by thymic DCs.

To directly test if cross-presentation could contribute to negative selection, we studied the fate of Jedi-TCRαβ DP thymocytes in RTOCs

**Fig. 5 | Expression of antigen by innate-like T cells is sufficient to mediate induction of tolerance in vivo. a**, GFP expression by thymic nonstromal populations in IL-17A–GFP mice (as in Fig. 3a; *n* = 5 mice). See also Extended Data Fig. 5a–c; MAIT, mucosal-associated invariant T. **b**, Reciprocal BM chimeras were established and immunized with GFP200–208 peptide as in Fig. 2c but using IL-17A–GFP donor and recipients (all mice were on a CB6F1 background). WT mice were used as an unimmunized control. Tetramer binding by CD8+CD44+ T cells, frequencies of CD8+CD44+tetramer+ cells from total CD8+ T cells and their absolute numbers are shown. Results were pooled from two independent experiments (WT → WT and IL-17A–GFP → WT groups) or from one experiment (the rest; *n* = 4 mice for nonimmunized, *n* = 8 for WT → WT, *n* = 11 for IL-17A–GFP → WT, *n* = 3 for WT → IL-17A–GFP and *n* = 3 for IL-17A–GFP → IL-17A–GFP). **c**, RTOCs with sorted Jedi DP thymocytes and thymic iNKT cells were established and analyzed as in Fig. 4b but using IL-17A–GFP iNKT cells (sorted as CD1d tetramer-binding cells (without gating on GFP); *n* = 2 RTOCs for WT iNKT cells and *n* = 3 for IL-17A–GFP iNKT cells). Representative results of three independent

experiments are shown. **d**, Jedi-TCRβ mice were injected intrathymically (i.t.) with iNKT cells sorted from the thymi of IL-4–GFP mice (both donor and recipient mice were on a BALB/c background). Thymi of the injected mice were analyzed 7 days later. Uninjected Jedi-TCRβ mice were used as controls. Representative plots show frequencies of GFP-expressing cells, frequencies of H2-Kd GFP200–208 tetramer-binding Jedi thymocytes and expression of the indicated cell-surface markers by the latter cells. Quantification of the frequencies of all tetramer-binding cells of total thymocytes, CD8α+CD8β+ tetramer-binding cells of total thymocytes and CD8α+CD8β+ cells of tetramer-binding cells are shown. Results are pooled from three intrathymic injection experiments (*n* = 8 lobes for the uninjected group and *n* = 6 lobes for the IL-4–GFP group; 2 lobes from the uninjected group were not stained for CD8 expression). Data are presented as mean ± s.d. (*P < 0.05, ***P < 0.001 and ****P < 0.0001) and were analyzed by two-tailed Student's *t*-test (**d**) or one-way ANOVA with a Holm–Sidak multiple comparisons test (**b**).

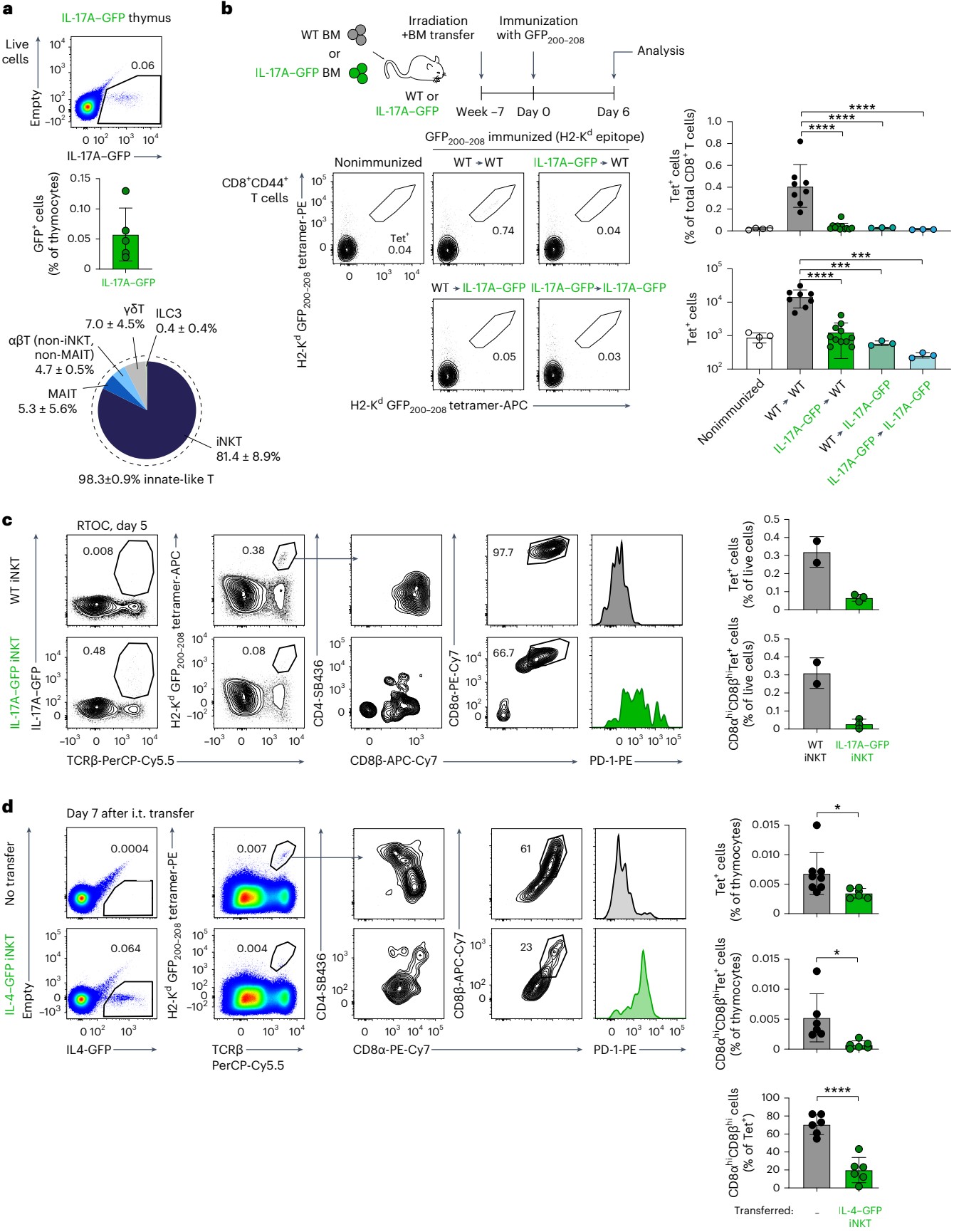

established with iNKT cells sorted from the thymi of IL-4−GFP mice either on an H2$^{b/d}$ background (capable of presentation to Jedi thymocytes) or on an H2$^{b/b}$ background that makes direct presentation to Jedi thymocytes impossible. Negative selection and phenotypic changes of Jedi cells were completely abrogated when iNKT cells were unable to directly present the GFP peptide to Jedi thymocytes due to this MHC mismatch, even when such iNKT cells were present at highly supra-physiological numbers (Fig. 6a). No evidence for cross-presentation was obtained when MHC-mismatched iNKT cells from IL-17A−GFP mice or MHC-mismatched iNKT cells expressing a secreted version of GFP were used in RTOC experiments (Extended Data Fig. 6b,c).

To assess the contribution of direct versus cross-presentation of IL-4−GFP in vivo, we next established three groups of BM chimeras: (1) WT H2$^{b/d}$ + Jedi-TCRβ H2$^{b/d}$ → WT H2$^{b/d}$ (no antigen), (2) IL-4−GFP H2$^{b/d}$ + Jedi-TCRβ H2$^{b/d}$ → WT H2$^{b/d}$ (both direct and cross-presentation is possible) and (3) IL-4−GFP H2$^{b/b}$ + Jedi-TCRβ H2$^{b/d}$ → WT H2$^{b/d}$ (only cross-presentation by APCs of Jedi and recipient origin is possible). Importantly, in the latter setting, more than half of the thymic DCs expressed H2-K$^d$ (Extended Data Fig. 6d) and therefore were theoretically capable of cross-presentation of GFP$_{200−208}$ epitope to Jedi cells. Although GFP-specific thymocytes were eliminated when direct presentation was possible, 'direct presentation-deficient' IL-4−GFP H2$^{b/b}$ + Jedi-TCRβ H2$^{b/d}$ → WT chimeras had numbers and cell-surface phenotypes of GFP-specific thymocytes indistinguishable from those of the no-antigen group (Fig. 6b). Finally, intrathymic injections of H2$^{b/b}$ (no direct presentation) or H2$^{b/d}$ (direct presentation is possible) iNKT cells expressing a secreted version of GFP into Jedi-TCRβ H2$^{b/d}$ recipients demonstrated that PD-1 upregulation and CD8$^+$ downregulation was evident only when transferred iNKT cells were capable of direct antigen presentation (Extended Data Fig. 6e). We conclude that in the case of GFP as an innate-like T cell-derived model antigen, it undergoes direct T cell-to-T cell presentation but not cross-presentation by thymic APCs.

Finally, to visualize these T cell-to-T cell antigen presentation events, we analyzed the interactions between fluorescently labeled iNKT cells and Jedi-TCRαβ thymocytes loaded with Calbryte-520 (Cal-520) AM Ca$^{2+}$ sensor dye in RTOCs by confocal microscopy. Jedi DP cells for these experiments were sorted for low levels of PD-1 expression to enrich TCR-expressing cells (Extended Data Fig. 6f). RTOCs were allowed to assemble overnight and were then imaged by confocal microscopy for 13 h (Fig. 6c−g). Both iNKT cells and viable Jedi-TCRαβ DP thymocytes were highly motile in these organoids (Supplementary Videos 1−4). In line with the expression of the Jedi TCR by about 70% of DP thymocytes at the beginning of the culture and their ongoing negative selection, approximately one-third of the contacts between Jedi thymocytes and IL-4−GFP iNKT cells resulted in Ca$^{2+}$ flux in Jedi-TCRαβ DP cells (Fig. 6d−f, Extended Data Fig. 6g and Supplementary Video 1).

No such contact-induced Ca$^{2+}$ signaling was observed in RTOCs with WT iNKT cells (Fig. 6d−f and Supplementary Video 2). Contacts between IL-4−GFP iNKT cells and Jedi-TCRαβ DP cells that lead to Ca$^{2+}$ signaling were longer than those that did not result in Ca$^{2+}$ flux (Fig. 6g), and, in some cases, Jedi thymocytes exhibited chasing behavior toward IL-4−GFP iNKT cells (Supplementary Video 3). In few cases, contact-induced Ca$^{2+}$ flux in Jedi-TCRαβ thymocytes was followed by a loss of motility and changes in cellular morphology suggestive of cell death (Supplementary Video 4). Thus, direct antigen presentation by iNKT cells to autoreactive thymocytes was confirmed by visualization of TCR signaling in autoreactive CD8$^+$ thymocytes.

## Lack of tolerance to IL-17A−GFP results in a fratricide reaction

We next sought to investigate the consequences of failure to disarm CD8$^+$ T cells reactive to an antigen expressed by effector T cells. Our initial analysis of *PLZF*$^{lu/lu}$ and *Tcrd*$^{−/−}$*Cd1d*$^{−/−}$*Mr1*$^{−/−}$ mice, which lack some innate-like T cell populations, did not reveal autoimmune phenotypes, in line with the unexpected abundance of all functional effector subsets in the thymi of these mice (see Supplementary Note 2 and Supplementary Figs. 3 and 4). We therefore next decided to take advantage of the fact that the thymic IL-17A−GFP$^+$ compartment was not properly reconstituted in IL-17A−GFP → WT BM chimeras, and GFP$^+$ cells were nearly absent in the thymi of these mice, whereas the peripheral IL-17A−GFP effector subsets were less affected in this setting (Extended Data Figs. 5e and 7a). We hypothesized that if we tip the balance between scarce thymic IL-17A−GFP$^+$ cells and developing GFP-specific CD8$^+$ thymocytes toward the latter cells in this system, some autoreactive CD8$^+$ T cells should escape to the periphery and may cause autoimmune elimination of peripheral T$_H$17 cells. To test this hypothesis, we established the following three groups of BM chimeras: (1) IL-17A−GFP → WT (no overproduction of autoreactive T cells), (2) WT + Jedi-TCRβ → WT (no antigen) and (3) IL-17A−GFP + Jedi-TCRβ → WT (predominantly peripheral antigen and overproduction of autoreactive T cells; Fig. 7a). As expected, small numbers of GFP-specific CD8$^+$ T cells with naive cell-surface phenotype were found in peripheral lymphoid organs of WT + Jedi-TCRβ → WT chimeras (Fig. 7b and Extended Data Fig. 7b). By contrast, lymphoid organs of IL-17A−GFP + Jedi-TCRβ → WT chimeras contained strongly expanded GFP-specific CD8$^+$ T cells with a CD44$^+$PD-1$^+$ activated cell-surface phenotype (Fig. 7b and Extended Data Fig. 7b). Thymi of IL-17A−GFP + Jedi-TCRβ → WT BM chimeras also exhibited an accumulation of GFP-specific CD44$^+$PD-1$^+$CD8$^+$ T cells (Extended Data Fig. 7c), possibly reflecting recruitment of activated cells from the periphery. Of note, this was in stark contrast to the previous observations made in IL-4-GFP + Jedi-TCRβ → WT BM chimera thymi that exhibited a complete loss of GFP-specific cells (Fig. 6b and Supplementary Fig. 2). This accumulation of activated autoreactive CD8$^+$ effector T cells coincided with a near complete loss of IL-17A−GFP$^+$

**Fig. 6 | Innate-like T cells can mediate elimination of autoreactive CD8$^+$ thymocytes through direct T cell-to-T cell antigen presentation. a**, RTOCs with sorted Jedi-TCRαβ DP thymocytes and iNKT cells were established and analyzed as in Fig. 4b but using iNKT cells sorted from WT mice (H2$^{b/d}$ background) or IL-4−GFP mice on H2$^{b/d}$ and H2$^{b/b}$ backgrounds. Results from one experiment ($n = 2$ RTOCs per group) are shown. **b**, CB6F1 WT recipient mice (H2$^{b/d}$) were irradiated and transferred intravenously (i.v.) with a mix of BM cells from Jedi-TCRβ mice on a CB6F1 background (H2$^{b/d}$) and WT H2$^{b/d}$, IL-4−GFP H2$^{b/d}$ or IL-4−GFP H2$^{b/b}$ mice (on mixed C57BL/6J and BALB/c genetic backgrounds). Thymi of chimeras were analyzed 6 weeks after reconstitution. Representative plots demonstrate the frequencies of GFP-expressing thymocytes, frequencies of H2-K$^d$ GFP$_{200−208}$ tetramer-binding thymocytes (quantification shown on the right side) and expression of the indicated cell-surface markers by the latter cells ($n = 3$ chimeras per group). Data are representative of two independent experiments. **c**, Schematic representation of RTOC experiments shown in **d**−**g**; O/N, overnight. **d**−**g**, Confocal microscopy analysis of RTOCs assembled with sorted Jedi-TCRαβ DP thymocytes (labeled with CellTrace Violet (CTV)

and Cal-520 AM Ca$^{2+}$ sensor dye) and thymic GFP$^+$ iNKT cells from IL-4−GFP mice or total iNKT cells from WT thymi (labeled with CellTrace Yellow (CTY)). **d**, Representative images showing interactions between Jedi thymocytes and IL-4−GFP iNKT cells (top and middle) and a WT iNKT cell (bottom); scale bars, 2 µm. **e**, Quantification of Ca$^{2+}$ signaling events in Jedi thymocytes contacting WT or IL-4−GFP iNKT cells. The numbers in the circles indicate the total number of interactions analyzed. **f**, Normalized Ca$^{2+}$ signaling intensity in Jedi thymocytes interacting with iNKT cells. Quantification was performed as described in the Methods. Mean and 95% confidence interval are shown ($n = 10$ Jedi cells per group). Individual curves for each cell are shown in Extended Data Fig. 6g. **g**, Quantification of the interaction time with iNKT cells for 90 Jedi thymocytes from RTOCs with WT iNKT cells (gray), 35 Jedi cells that underwent Ca$^{2+}$ flux in RTOCs with IL-4−GFP iNKT cells (orange) and 44 Jedi cells that did not undergo Ca$^{2+}$ flux in RTOCs with IL-4−GFP iNKT cells (blue). Data are presented as mean ± s.d. (**$P < 0.01$, ***$P < 0.001$ and ****$P < 0.0001$) and were analyzed by one-way ANOVA with a Holm–Sidak multiple comparisons test (**b**) or Kruskal–Wallis test (**g**).

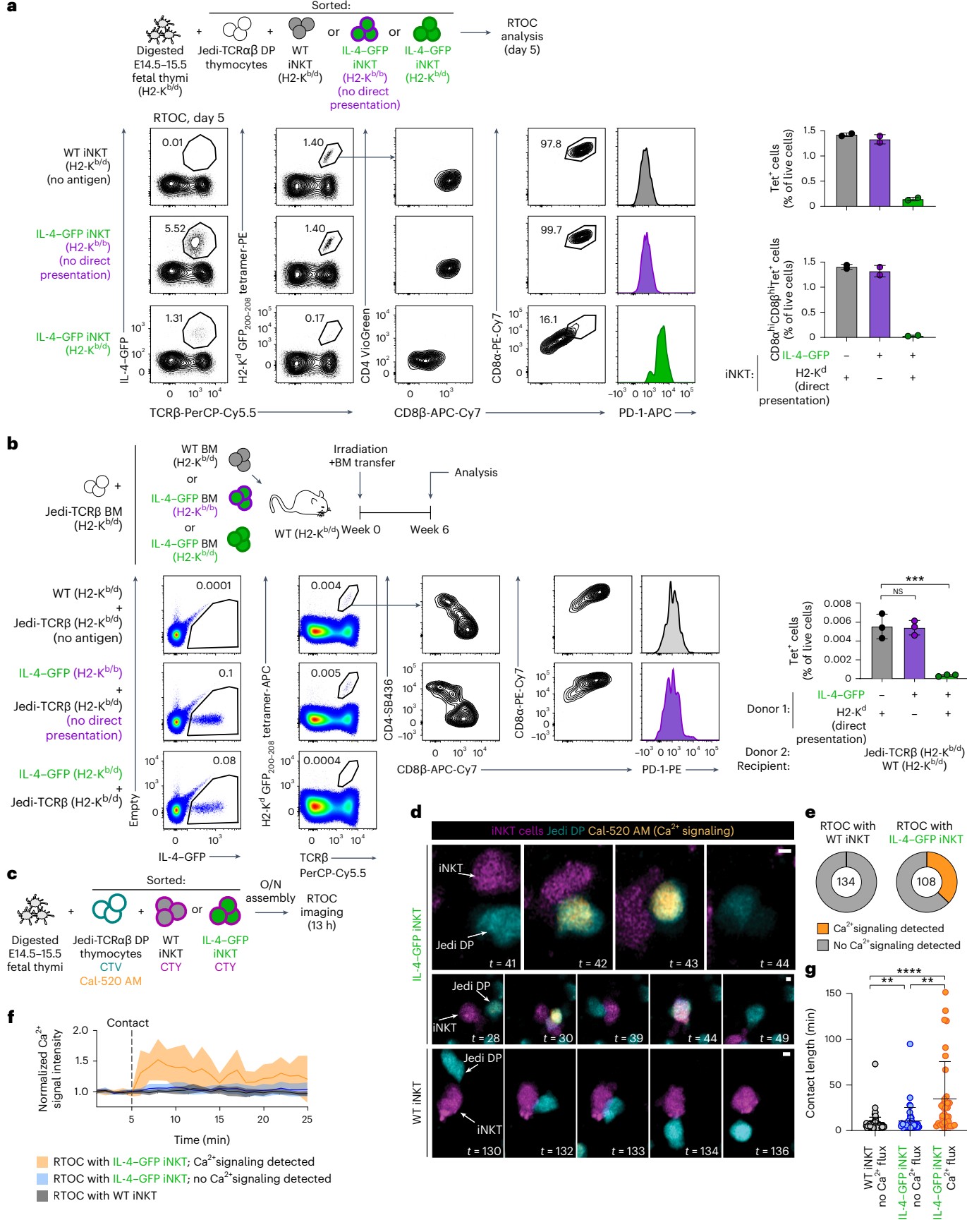

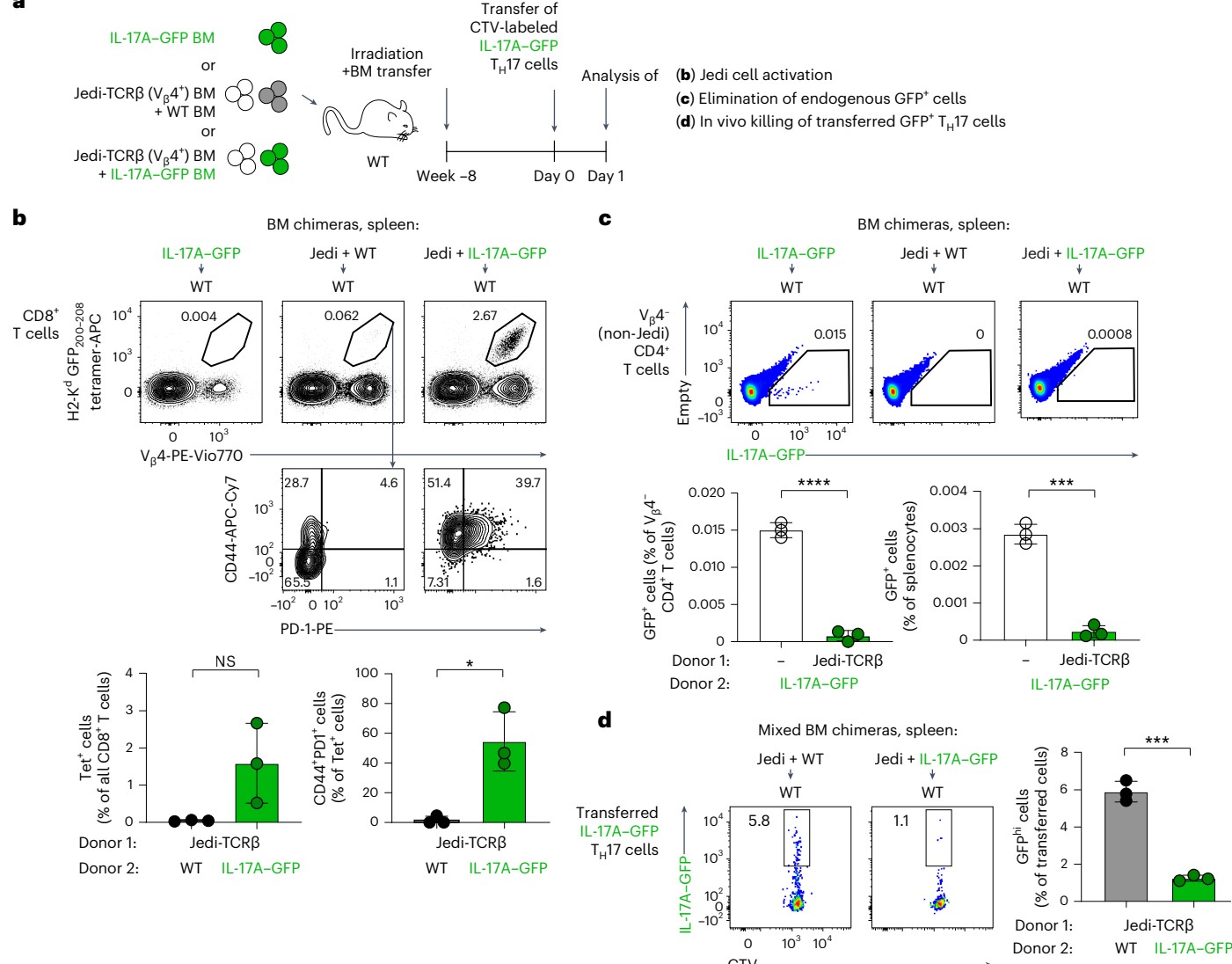

**Fig. 7 | Decreased thymic abundance of IL-17A–GFP results in autoimmune elimination of peripheral GFP+ TH17 cells. a**, Schematic representation of experiments shown in **b**–**d**. WT recipient mice were irradiated and transferred i.v. with T cell- and NK cell-depleted BM cells from IL-17A–GFP mice or Jedi-TCRβ mice and WT mice or Jedi-TCRβ mice and IL-17A–GFP mice (all on a CB6F1 background). Spleens were analyzed 8 weeks after reconstitution. Note the near absence of thymic IL-17A–GFP-expressing cells in BM chimeras (see Extended Data Fig. 7a). **b**, Frequency and cell-surface phenotype of H2-K$^d$ GFP$_{200-208}$ tetramer-binding CD8+ T cells in the spleens of BM chimeras. **c**, Frequency of

IL-17A–GFP-expressing cells among total lymphocytes and V$_β$4$^-$ (non-Jedi) CD4+ T cells. **d**, Twenty hours before collection, CTV-labeled TH17 cells in vitro differentiated from IL-17A–GFP CD4+ T cells were i.v. transferred into the indicated groups of chimeras. Elimination of the GFP$^{hi}$ fraction of CTV+ cells was assessed by flow cytometry. Representative results of two independent experiments (**b** and **c**) or one experiment (**d**; $n = 3$ chimeras per group) are shown. Data are presented as mean ± s.d. (*$P < 0.05$, ***$P < 0.001$ and ****$P < 0.0001$) and were analyzed by two-tailed Student's $t$-tests.

TH17 cells (Fig. 7c and Extended Data Fig. 7d,e; gating on V$_β$4$^-$ cells was applied to exclude Jedi cells from the analysis). These results suggest that breach in tolerance induction results in an autoimmune attack that may eliminate peripheral IL-17A-expressing effector T cells. To formally test this, we performed an in vivo killing assay by transferring TH17 cells in vitro differentiated from CD4+ T cells of IL-17A–GFP mice. Only a fraction of these cells induced the expression of GFP (which correlated well with the production of IL-17A; Extended Data Fig. 7f) and therefore would represent possible targets for activated Jedi cells. These GFP+ cells were selectively eliminated in IL-17A–GFP + Jedi-TCRβ → WT, but not in WT + Jedi-TCRβ → WT, chimeras (Fig. 7d). Together, these results demonstrate that failure to induce CD8+ T cell tolerance to a single inflammation-associated self-antigen can result in a fratricide reaction that leads to elimination of the entire class of effector T cells expressing this molecule.

## Induction of tolerance to an endogenous effector molecule

Finally, we sought to test if expression of an endogenous effector T cell molecule exclusively in the hematopoietic compartment is sufficient to induce CD8+ T cell tolerance to this antigen. We focused on IFNγ, the production of which among thymocytes was restricted to T cell/ NK cell/innate lymphoid cell (ILC) lineages, as evidenced by Thy1 and NK1.1 expression and lack of MHC class II expression by IFNγ+ cells (Extended Data Fig. 8a). We identified an H2-K$^b$ epitope in IFNγ (IFNγ$_{69-78}$ QIISFYLRLF), the immunization of which resulted in a strong CD8+ T cell response in *Ifng*$^{-/-}$, but not WT, animals (Extended Data Fig. 8b). We next established the following four groups of BM chimeras: (1) WT → WT, (2) *Ifng*$^{-/-}$ → WT, (3) WT → *Ifng*$^{-/-}$ and (4) *Ifng*$^{-/-}$ → *Ifng*$^{-/-}$ and immunized them with IFNγ$_{69-78}$ and an irrelevant foreign peptide (ovalbumin (OVA$_{257-264}$) SIINFEKL). Intracellular staining for tumor necrosis factor (TNF) after a brief peptide restimulation was used as a readout

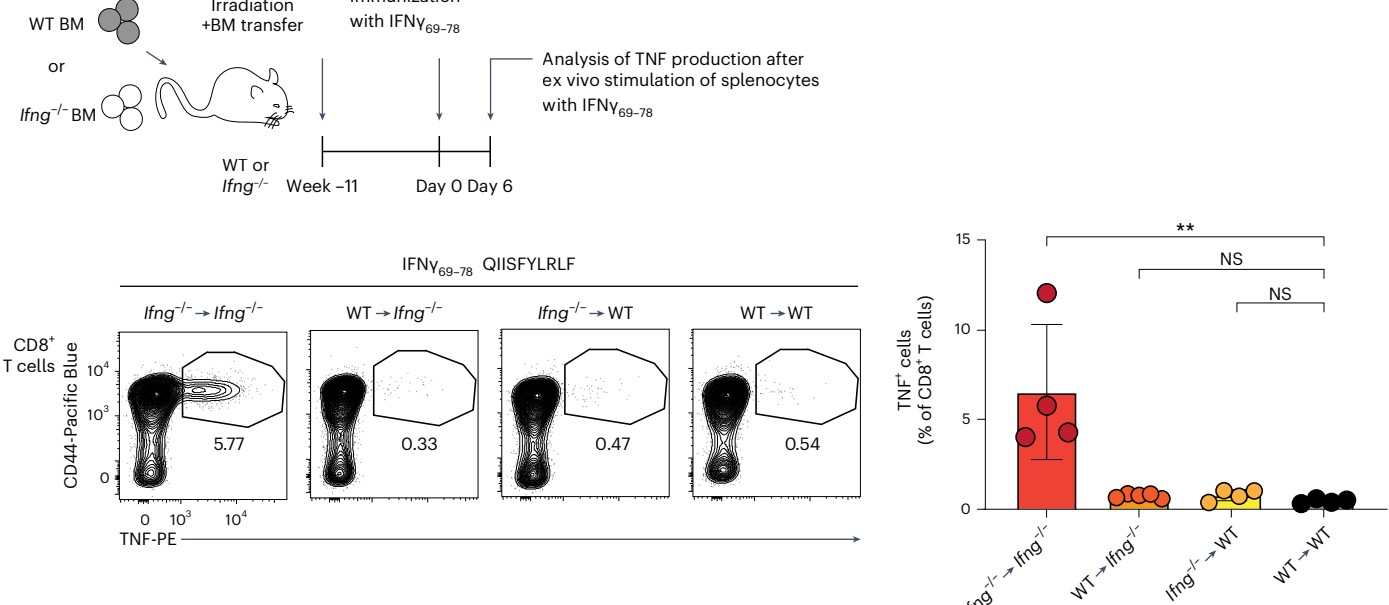

**Fig. 8 | Expression of IFNγ solely by hematopoietic cells is sufficient to induce CD8⁺ T cell tolerance.** WT and *Ifng⁻ᐟ⁻* mice on a C57BL/6J background were lethally irradiated and reconstituted with syngeneic WT or *Ifng⁻ᐟ⁻* BM cells depleted of T and NK cells. Eleven weeks after reconstitution, the resulting four groups of BM chimeras were immunized with IFNγ$_{69–78}$ and OVA$_{257–264}$ peptides using anti-CD40 and poly(I:C) as adjuvant (see also Extended Data Fig. 8c). For quantification of antigen-specific CD8⁺ T cell responses, splenocytes were isolated 6 days after immunization and stimulated with IFNγ$_{69–78}$ peptide for 6 h (Methods). Production of TNF by CD8⁺ T cells was analyzed by flow cytometry (*n* = 5 for WT → *Ifng⁻ᐟ⁻* and *n* = 4 for the other groups). Data are presented as mean ± s.d. (**P < 0.01) and were analyzed by one-way ANOVA with a Holm–Sidak multiple comparisons test. Data are representative of two independent experiments.

for antigen-specific CD8⁺ T cell responses. All four groups of chimeras showed a response to the foreign epitope immunization (Extended Data Fig. 8c). By contrast, only *Ifng⁻ᐟ⁻* → *Ifng⁻ᐟ⁻* chimeras exhibited a strong response to IFNγ$_{69–78}$ immunization, whereas all other groups of chimeras were equally tolerant to this epitope (Fig. 8). Thus, expression of IFNγ solely by hematopoietic cells can induce CD8⁺ T cell tolerance to this molecule. Together with the T cell/NK cell/ILC-restricted expression, these results suggest that antigen expression by these effector subsets is sufficient for induction of tolerance to this endogenous inflammation-associated self-antigen.

## Discussion

Self/non-self discrimination by the adaptive immune system largely relies on the context of antigen encounter. For T lymphocytes, a variety of tolerance mechanisms ensure inactivation of autoreactive cells in the thymus or in the periphery in the absence of inflammation induced by pathogen invasion. Only if a mature T lymphocyte encounters an antigen in the context of inflammation does it receive all signals required to induce clonal expansion and execution of effector programs. The expression of numerous self-molecules involved in T cell effector programs therefore overlaps in time and space with the emergence of pathogen-derived antigens. In this study, we aimed to investigate how the immune system distinguishes such inflammation-associated self-antigens from pathogen-derived molecules. We demonstrated that CD8⁺ T cell tolerance to inflammation-associated self-antigens can be induced in the thymus through antigen presentation by a variety of thymic cell types. These include two cell types not previously implicated in T cell tolerance induction—thymus-resident innate-like T cells and thymic eosinophils. We showed that minute numbers of thymic innate-like T lymphocytes, cells that express self-antigens characteristic of all major T cell effector programs, were sufficient to efficiently eliminate autoreactive CD8⁺ thymocytes. We demonstrated that CD8⁺ T cell tolerance induction by innate-like T lymphocytes can be achieved without the involvement of professional APCs through direct T cell-to-T cell antigen presentation. We also provided evidence for the contribution of effector lymphocytes to the induction of tolerance to an endogenous inflammation-associated self-antigen. Finally, we showed that decreased abundance of a model inflammation-associated self-antigen in the thymus can result in a spontaneous fratricide reaction, causing complete elimination of peripheral effector T cell subsets expressing this antigen.

This unprovoked and extremely efficient elimination of a whole class of effector T cells after breach of tolerance to a single inflammation-associated self-antigen highlights the ultimate importance of tolerance to this group of molecules. The importance of this tolerance seems to be further reflected by the high redundancy of cell types that express such antigens in the thymus. Such redundant expression was also characteristic for IFNγ that was expressed by thymic innate and innate-like lymphocytes and a small group of TECs. Well in line with our observations with model antigens, both radioresistant and radiosensitive cells were able to mediate CD8⁺ T cell tolerance to this endogenous inflammation-associated self-antigen. Together with the T cell/NK cell/ILC-restricted expression of IFNγ in the hematopoietic compartment, these results provide evidence for the role of effector lymphocytes in the induction of tolerance to this proinflammatory cytokine. Although by themselves these experiments cannot discriminate between central and peripheral tolerance mechanisms, the abundance of IFNγ-expressing cells in the thymus and our results obtained with model antigens suggest that central tolerance is likely to contribute to the neutralization of IFNγ-reactive CD8⁺ T cells.

Although cytotoxic responses to inflammation-associated molecules have not, to our knowledge, been studied in the context of autoimmunity, many human autoimmune diseases are associated with autoantibodies against a broad range of proinflammatory cytokines[54], suggesting that CD4⁺ T cell tolerance to these self-antigens is broken in such patients. Interestingly, in our study using IL-4−GFP and IL-17A−GFP as model inflammation-associated self-antigens, we observed profound CD8⁺, but not CD4⁺, T cell tolerance. We also did not find

any evidence of GFP cross-presentation by thymic DCs in this system, even when innate-like T cells expressed a secreted form of GFP. However, we did not test if secreted GFP may be subjected to MHC class II presentation by professional APCs. Moreover, the situation may be different for some endogenous secreted antigens, for example, if professional thymic APCs would express a receptor for such a molecule. It remains to be investigated if in these scenarios secreted inflammation-associated self-antigens would be taken up and presented by DCs, enabling the induction of CD4$^+$ T cell tolerance. Alternatively, it is also possible that induction of CD4$^+$ T cell tolerance is less stringent for inflammation-associated self-antigens, making them frequent targets in autoimmunity. Finally, unlike their mouse counterparts, human thymocytes can express MHC class II molecules[55], and the expression of genes encoding MHC class II chains was relatively high in some of the thymocyte populations that expressed T cell effector genes (Extended Data Fig. 1). It is therefore conceivable that in humans, these cells could also present antigens to developing CD4$^+$ thymocytes and therefore contribute to the induction of CD4$^+$ T cell tolerance.

Although the role of direct antigen presentation by professional APCs in the elimination of autoreactive thymocytes is well documented[8,10], the contribution of antigen presentation by non-professional APCs to this process under physiological circumstances remained unclear. In in vitro systems, peptide-pulsed thymocytes are capable of direct antigen presentation and induction of cell death in autoreactive thymocytes[56,57]. In an in vivo setting, mature T cells were shown to mediate the induction of central tolerance to an alloantigen[58]. In addition, it was demonstrated that autoantigens expressed by T cells[59] and thymic fibroblasts[60] can contribute to the induction of T cell tolerance. However, it remains unclear if the latter happens through direct antigen presentation of autoantigens or through their cross-presentation by professional APCs.

In this study, we have demonstrated that endogenous antigen expression by minute numbers of thymic-resident innate-like T cells, thymic eosinophils and conventional developing thymocytes was sufficient to eliminate autoreactive CD8$^+$ thymocytes. Moreover, we demonstrated that innate-like T cells can induce CD8$^+$ T cell tolerance by directly presenting self-antigens to autoreactive thymocytes. These results substantially expand the array of known cell types that can mediate the induction of T cell tolerance and suggest that most cell types that express MHC class I and are present in the thymus are likely able to contribute to the induction of CD8$^+$ T cell tolerance. In addition to B cells and eosinophils, mouse and/or human thymi were reported to harbor several NK cell and ILC subsets, plasma cells, mast cells, erythroblasts, megakaryocytes and recirculating memory T cells[35,61–63]. Thus, in addition to ectopic antigen expression by mTECs, recruitment and/or retention of these 'ectopic cell types' may represent a mechanism that ensures representation of hematopoietic system-derived self-antigens in the thymus.

Recent studies have demonstrated that large fractions of iNKT cells and γδT cells in the thymus are tissue-resident cells[23,24]. Moreover, thymus-resident innate-like T cells are in the center of a complex 'ecosystem' that actively ensures their presence in the thymus and production of inflammatory mediators by these cells. Indeed, a tuft cell-like subset of mTECs is required for iNKT cell accumulation in the thymus[5], whereas production of IL-4 by these cells is ensured by their CD1d-dependent activation by thymic myeloid cells[64]. In turn, type 2 cytokines secreted by iNKT cells support the intrathymic generation of IFNγ-producing CD8$^+$ T cells[25,26], regulate the recruitment of thymic eosinophils[27] and induce the activation thymic DCs[29]. In addition, proinflammatory cytokines of both stromal (type 1 IFNs, IL-15, IL-6 and thymic stromal lymphopoetin) and innate-like T cell (IFNγ) origin were shown to regulate thymic T cell development[65,66], indicating that these mediators are present in the thymic microenvironment in biologically meaningful amounts. Moreover, several inflammatory pathways, including type 1/type 3 IFN signaling and Toll-like receptor and/or IL-1β signaling, are constitutively active in TECs[30,31].

In this study, we demonstrated that at least two cellular components of this inflammatory network (thymic innate-like T cells and eosinophils) were sufficient to mediate the induction of remarkably efficient tolerance to antigens that they express. Taking these observations together, it is tempting to speculate that constitutive activation of a plethora of inflammatory pathways in the thymus may have evolved to ensure induction of tolerance to inflammation-associated self-antigens, a class of molecules that otherwise largely mirror the spatial and temporal distribution of pathogen-derived antigens.

## Online content

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

[1]Division of Immunology and Allergy, Department of Medicine Solna, Karolinska Institutet, Karolinska University Hospital, Stockholm, Sweden. [2]Center for Molecular Medicine, Karolinska Institutet, Stockholm, Sweden. [3]Department of Applied Physics, Science for Life Laboratory, KTH Royal Institute of Technology, Stockholm, Sweden. [4]Institute of Immunology, Hannover Medical School, Hannover, Germany. [5]Division of Rheumatology, Department of Medicine Solna, Karolinska Institutet, Karolinska University Hospital, Stockholm, Sweden. [6]Max Planck Research Group, Würzburg Institute of Systems Immunology, Julius-Maximilians-Universität Würzburg, Würzburg, Germany. [7]Department of Medicine, Division of Immunology and Rheumatology, Stanford University, Stanford, CA, USA. [8]Epigenetic Factors in Normal and Malignant Hematopoiesis Lab, CRCM, CNRS, INSERM, Institut Paoli Calmettes, Aix Marseille University, Marseille, France. [9]Equipe Labellisée Ligue Nationale Contre le Cancer, Paris, France. [10]Department of Cancer Immunology and Virology, Dana-Farber Cancer Institute, Boston, MA, USA. [11]Parker Institute for Cancer Immunotherapy, Dana-Farber Cancer Institute, Boston, MA, USA. [12]Department of Immunology, Harvard Medical School, Boston, MA, USA. [13]Ludwig Center at Harvard, Boston, MA, USA. [14]The Precision Immunology Institute, Icahn School of Medicine at Mount Sinai, New York, NY, USA. [15]Icahn Genomics Institute, Icahn School of Medicine at Mount Sinai, New York, NY, USA. [16]Department of Immunology and Immunotherapy, Icahn School of Medicine at Mount Sinai, New York, NY, USA. [17]Institute for Immunity, Transplantation and Infection, Stanford University School of Medicine, Stanford, CA, USA. [18]Hamburg Center for Translational Immunology (HCTI), University Medical Center Hamburg-Eppendorf, Hamburg, Germany. [19]Institute of Systems Immunology, University Medical Center Hamburg-Eppendorf, Hamburg, Germany. [20]Center for Infectious Medicine, Department of Medicine Huddinge, Karolinska Institutet, Stockholm, Sweden. [21]These authors contributed equally: Yuanyuan You, Josefine Dunst. ✉e-mail: taras.kreslavskiy@ki.se

## Methods

### Mice

IL-4−GFP[41], IL-17A−GFP and Ifng[−/−][67] mice were purchased from the Jackson Laboratory. Rag2[−/−] mice were purchased from Janvier Labs. PLZF[lu/lu][68], OT-I[69], Jedi-TCRαβ[44] and Tcrd[−/−]Cd1d[−/−]Mr1[−/−][71] mice were described previously. WT C57BL/6J and BALB/c mice were obtained from Janvier Labs or were bred in-house. Jedi-TCRβ-only mice were generated from Jedi-TCRαβ mice by crossing with WT BALB/c mice and were further backcrossed to the BALB/c background for more than ten generations. WT C57BL/6J × BALB/c F1 mice (referred to as CB6F1), IL-4−GFP CB6F1, IL-17A−GFP CB6F1, ACTB−GFP CB6F1 and Jedi-TCRβ CB6F1 mice were generated by crossing mice on C57BL/6J and BALB/c backgrounds. For MHC mismatch experiments, IL-4−GFP CB6F1 mice were crossed with WT C57BL/6J mice to generate IL-4−GFP H2[b/d] and IL-4−GFP H2[b/b] mice. Tcrd[−/−]Cd1d[−/−]Mr1[−/−] mice were bred and maintained at the Würzburg Institute for Systems Immunology. All other mice were bred and maintained at the Comparative Medicine Biomedicum facility of Karolinska Institutet (Stockholm, Sweden) and were kept under specific pathogen-free conditions. All mouse experiments were performed according to valid ethical permits, which were approved and regularly controlled by the Swedish and/or German Veterinary Authority.

### Flow cytometry

Mouse spleen tissue, lymph nodes and thymi were collected, and single-cell suspensions were obtained by mincing through 70-µm cell strainers. Isolation of intestinal intraepithelial lymphocytes (iIELs) was performed as described previously[72]. Isolation of TECs was performed as described previously[73] with minor modifications. In short, each thymus was digested in 0.5–1 ml of RPMI 1640 medium with 1 mg ml[−1] collagenase IV and 100 U ml[−1] DNase I for 20 min at 37 °C before staining with APC-labeled anti-EPCAM, followed by enrichment using anti-APC microbeads (Miltenyi Biotec) and magnets from the EasySep Cell Isolation and Separation Technology platform (STEMCELL Technologies). For detection of antigen-specific T cells, single-cell suspensions were incubated with APC- and/or PE-conjugated H2-K[d] GFP$_{200-208}$ or I-A[b] GFP$_{81-95}$, PE- or BV421-conjugated CD1d-PBS-57 or APC-conjugated MR1-5-OP-RU tetramers (all provided by the National Institutes of Health tetramer core facility) for 1 h before enrichment and further cell-surface staining (for I-A[b] GFP$_{81-95}$) or they were directly included into the surface antibody staining mix (for all other tetramers). Tetramer-binding CD4[+] T cells were enriched by magnetic enrichment as described by Moon et al.[74]. Dead cells were detected using a Live/Dead Fixable Aqua Dead Cell Stain kit for 405-nm excitation (Thermo Fisher Scientific) or the fixable viability dye eFluor780 (eBioscience) according to manufacturers' instructions and were excluded from further analysis unless stated otherwise.

For the detection of cleaved caspase-3, cells were fixed and permeabilized with BD Cytofix/Cytoperm (BD Bioscience) and stained with rabbit anti-mouse active caspase-3 (5A1E) on ice for 30 min, followed by incubation with PE-conjugated anti-rabbit IgG (H + L) F(ab')$_2$ fragment antibody on ice for 30 min. Intracellular cytokine staining was performed using a BD Cytofix/Cytoperm kit (BD Bioscience) following manufacturer's instructions.

Data were acquired on an LSR Fortessa, FACSCanto II (BD Biosciences) or Cytek Aurora (Cytek Biosciences) flow cytometer and analyzed with FlowJo software v.10 (BD Biosciences). Cell sorting was performed using FACS Aria III and FACS Fusion cell sorters (BD Biosciences).

### Flow cytometry staining reagents

Monoclonal antibodies specific for CD4 (GK1.5, RM4-8), CD3e (145-2C11, 17A2), CD8α (53-6.7), CD8β (YT5156.7.7), PD-1 (REA802, 29F.1A12), CD24 (M1/69), CD44 (IM7), CD19 (1D3, 6D5), B220 (RA3-6B2), CD45.1 (A20), CD45.2 (104), F4/80 (BM8), CD11c (N418), NK1.1 (PK136), TCRβ (H57-597), TCRγδ (GL3), H2-K[b] (AF 6-885), H2-K[d] (SF1-1.1), cKit (2B8), Ter119 (Ter119), SiglecF (E50-2440), MHC class II (M5/114.15), Ly51

(REA988), EPCAM (REA977), TCRV$_α$2 (B20.1), TCRV$_β$4 (REA729), TNF (MAb11), Thy1.2 (30-H12), Thy1.1 (OX7), Ly49 (14B11), CD122 (TM-β1), CCR7 (4B12), CD25 (REA568), CD80 (16-10A1), CD86 (GL-1), IFNγ (REA638, XMG1.2), IL-4 (11B11) and IL-17A (TC11-18H10) were purchased from BioLegend, BD Biosciences, Miltenyi Biotec or Thermo Fisher Scientific. Monoclonal antibody targeting caspase-3 (5A1E) and anti-rabbit IgG (H + L) F(ab')$_2$ fragments were purchased from Cell Signaling Technology. Streptavidin-APC (405207) was purchased from BioLegend. Biotinylated UEA1 was purchased from Vector Laboratories. All antibodies and staining reagents were used at dilutions specified by the manufacturer or determined experimentally.

### Generation of BM chimeras

For all BM chimera experiments, BM cells from donor mice were stained with APC-labeled CD4, CD8, CD3, TCRβ, TCRγδ, NK1.1 and Ter119 antibodies, followed by magnetic depletion of T and NK cells with anti-APC microbeads (Miltenyi Biotec). BM cells ($2 \times 10^6$–$5 \times 10^6$) were then transferred i.v. into lethally irradiated (split dose of 5 Gy, twice, using an Xstrahl CIX3 X-ray irradiator) recipients. For mixed BM chimera experiments shown in Fig. 3c, Jedi-TCRαβ BM cells were mixed with WT or IL-4−GFP BM cells at a 3:1 ratio before transfer. For mixed BM chimera experiments shown in Fig. 6b, Jedi-TCRβ H2[b/d] BM cells were mixed at a 1:1 ratio with IL-4−GFP H2[b/b] or IL-4−GFP H2[b/d] BM cells before transfer. For mixed BM chimera experiments shown in Fig. 7, Jedi-TCRβ BM cells were mixed at a 1:20 ratio with IL-17A−GFP or WT BM cells before transfer into WT recipients (all on a CB6F1 genetic background). For mixed BM chimera experiments shown in Supplementary Fig. 2, Jedi-TCRβ or Jedi-TCRαβ BM cells were mixed at a 1:1 ratio with IL-4−GFP or WT BM cells before transfer into WT recipients (all on a BALB/c genetic background).

### In vitro T$_H$2 cell and T$_H$17 cell differentiation

For T$_H$17 cell differentiation, CD4[+] T cells from spleens of IL-17A−GFP mice on a CB6F1 genetic background were isolated by staining with APC-conjugated anti-CD4, followed by magnetic enrichment with anti-APC magnetic beads (Miltenyi Biotec). Enriched CD4[+] T cells were cultured in six-well plates (precoated with 4 µg ml[−1] anti-CD3e (Biolegend) in PBS at 4 °C overnight) in IMDM supplemented with 10% fetal calf serum (FCS), 50 µM 2-mercaptoethanol, 2 mM L-glutamine, 100 U ml[−1] penicillin, 100 U ml[−1] streptomycin, 0.5 µg ml[−1] anti-CD28 (Biolegend), 50 ng ml[−1] IL-6 (Peprotech), 2 ng ml[−1] TGFβ (Peprotech), 2.5 µg ml[−1] anti-IL-4 (Biolegend) and 2.5 µg ml[−1] anti-IFNγ (Biolegend). The cells were used in in vivo killing assays 4 days after the start of the cultures.

For T$_H$2 cell differentiation, CD4[+] T cells from spleens of WT C57BL/6J mice were isolated as described above. Enriched CD4[+] T cells were cultured in six-well plates (precoated with 4 µg ml[−1] anti-CD3e in PBS at 4 °C overnight) in IMDM supplemented with 10% FCS, 50 µM 2-mercaptoethanol, 2 mM L-glutamine, 100 U ml[−1] penicillin, 100 U ml[−1] streptomycin, 0.5 µg ml[−1] anti-CD28, 10 ng ml[−1] IL-2 (Peprotech), 20 ng ml[−1] IL-4 (Peprotech) and 2.5 µg ml[−1] anti-IFNγ. Induction of the T$_H$2 molecular program was confirmed by side-by-side analysis of GFP expression in T$_H$2 cells differentiated from IL-4−GFP CD4[+] T cells (80–90% of GFP[+] cells). The cells were used for repetitive transfer into the indicated BM chimeras or in in vivo killing assays 4 days after the start of the cultures.

### In vivo killing assays

In vitro-differentiated IL-17A−GFP T$_H$17 cells were labeled with 1 µM CTV (Thermo Fisher) in PBS with 0.5% bovine serum albumin (BSA) in a water bath at 37 °C for 8 min. Labeled cells were washed and i.v. transferred ($0.5 \times 10^6$) into the indicated groups of BM chimeras. Disappearance of GFP[hi]CTV[+] cells was assessed by flow cytometry in the spleens of the recipient mice 20 h after transfer.

In vitro-differentiated T$_H$2 cells were labeled with 1 µM CTV and mixed with 10 µM CTV-labeled freshly isolated naive T cells at a 1:3

ratio. Labeled cells were washed with PBS and i.v. transferred into the indicated groups of BM chimeras ($2 \times 10^6$ cells per mouse). The ratio between CTV$^{int}$ (T$_H$2) and CTV$^{hi}$ (naive) cells was assessed in the spleens of recipient mice 20 h after transfer by flow cytometry.

## Construction of retroviral vectors

DNA fragments encoding full-length cytoplasmic OVA with N4 (SIINFEKL), Q4 (SIIQFEKL), T4 (SIITFEKL) and V4 (SIIVFEKL) versions of the OVA$_{257-264}$ epitope were synthesized by Twist Bioscience and cloned into a retroviral pMIG II vector[75] using BglII and XhoI sites. To generate the Thy1.1-IRES-i.c.GFP construct, the Thy1.1-coding sequence was cloned into pMIG II using BglII and XhoI sites. To generate the Thy1.1-IRES-secGFP construct, the DNA sequence encoding a secretory signal peptide of the FSH-β subunit was inserted at the 5′ end of the GFP-coding sequence in the Thy1.1-IRES-i.c.GFP construct as described previously[76] with minor modifications. In brief, 5′- TGAAAAACAC-GATAATACCATGATGAAGTCGATCCAGCTTTGCATCCTACTCTGGT-GCTTGAGAGCAGTCTGCTGCCATATGGTGAGCAAGGGCGAGG -3′ and 5′-CCTCGCCCTTGCTCACCATATGGCAGCAGACTGCTCTCAAG CACCAGAGTAGGATGCAAAGCTGGATCGACTTCATCATGGTATT ATCGTGTTTTTCA -3′ oligonucleotides were annealed and mixed with linearized pMIG II vector (digested with NcoI) and incubated at 50 °C for 30 min with NEBuilder HiFi DNA Assembly Master Mix (New England Biolabs). To generate the Thy1.1-only construct, the Thy1.1-IRES-i.c.GFP construct was digested with NotI and NcoI to remove the IRES-i.c.GFP cassette, followed by removal of overhangs using Klenow fragment (New England Biolabs) and self-ligation of the digested vector. To generate the Bcl-2-P2A-Thy1.1-IRES-secGFP construct, the Thy1.1-coding sequence in the Thy1.1-IRES-secGFP construct was replaced with a DNA fragment encoding Bcl-2-P2A-Thy1.1 (synthesized by Twist Bioscience) using BglII and XhoI sites.

## Transfection and retrovirus production

HEK293T cells were cultured in DMEM supplemented with 10% FCS, 50 μM 2-mercaptoethanol, 2 mM L-glutamine, 100 U ml$^{-1}$ penicillin and 100 U ml$^{-1}$ streptomycin at 37 °C and 5% CO$_2$ until the confluency reached 70%. To test GFP expression and secretion, HEK293T cells were transiently transfected with Thy1.1-IRES-i.c.GFP or Thy1.1-IRES-secGFP constructs by calcium phosphate transfection, as previously described[77]. The culture medium was replaced with DMEM (without phenol red) 6 h after transfection. Supernatants were collected 8, 24 and 48 h after transfection, and fluorescence intensity was measured with a Varioskan LUX Multimode Microplate Reader (Thermo Fisher) at an excitation wavelength of 480 nm and emission wavelength of 520 nm (with a bandwidth of 12 nm).

Retrovirus-containing supernatants were generated by transient cotransfection of HEK293T cells with the retroviral constructs described above and pCL-Eco packaging vector using calcium phosphate transfection. Viral supernatants were collected 48 h after transfection, mixed thoroughly with 5× PEG solution (40% polyethylene glycol 10,000 (Sigma) and 2.4% NaCl dissolved in water) and incubated overnight at 4 °C. The next day, virus particles were precipitated by centrifugation at 2,000$g$ and 4 °C for 1 h, resuspended in fresh culture medium to reach 100-times the concentration and stored at −80 °C.

## Immunization experiments

To study antigen-specific CD8$^+$ T cell responses, mice were intraperitoneally injected with 200 μg of GFP$_{200-208}$ peptide (HYLSTQSAL, H2-K$^d$ epitope; Genscript) dissolved in PBS plus 50 μg of anti-CD40 (clone FGK4.5, BioXcell) and 50 μg of poly(I:C) (Invivogen).

For CD4$^+$ T cell peptide immunizations, mice were subcutaneously injected with 100 μl of an emulsion containing 100 μg of GFP$_{81-95}$ peptide (HDFFKSAMPEGYVQE, I-A$^b$ epitope; Genscript). Emulsions were prepared by mixing the peptide in PBS in a 1:1 ratio with complete Freund's adjuvant (Sigma-Aldrich).

For IFNγ and OVA peptide immunization, BM chimeric mice were intraperitoneally injected with 200 μg of IFNγ$_{69-78}$ peptide (QIISFYLRLF, H2-K$^b$ epitope; Genscript) and 50 μg of OVA$_{257-264}$ peptide (SIINFEKL, H2-K$^b$ epitope; Genscript) with 50 μg of anti-CD40 and 50 μg of poly(I:C).

## Cytokine production assays

For characterization of thymic effector subsets in WT and *PLZF*$^{lu/lu}$ mice, isolated thymocytes were incubated in RPMI 1640 culture medium supplemented with 10% FCS, 50 μM 2-mercaptoethanol, 2 mM L-glutamine, 100 U ml$^{-1}$ penicillin, 100 U ml$^{-1}$ streptomycin, 50 ng ml$^{-1}$ phorbol 12-myristate 13-acetate (PMA; Sigma), 1 μg ml$^{-1}$ ionomycin (Sigma) and 3 μg ml$^{-1}$ brefeldin A (Thermo Fisher) at 37 °C and 5% CO$_2$. After 5 h, thymocytes were collected for surface antibody and viability staining, followed by intracellular staining using a BD Cytofix/Cytoperm kit (BD Biosciences) according to manufacturer's instructions. For detection of intracellular cytokines in *Tcrd*$^{-/-}$*Cd1d*$^{-/-}$*Mr1*$^{-/-}$ mice, isolated thymocytes were cultured in RPMI 1640 culture medium with 25 ng ml$^{-1}$ PMA, 1 μg ml$^{-1}$ ionomycin and 1 μg ml$^{-1}$ brefeldin A for 4 h at 37 °C and 5% CO$_2$.

For detection of antigen-specific CD8$^+$ responses in peptide-immunized *Ifng*$^{-/-}$/WT BM chimeras, spleens were collected 6 days after immunization and processed to a single-cell suspension. After erythrocyte lysis, splenocytes were resuspended in IMDM containing the indicated peptides (final concentration of 10 μg ml$^{-1}$) and cultured in a six-well plate at a density of $2 \times 10^6$ cells per ml ($1 \times 10^7$ cells per well) at 37 °C and 5% CO$_2$. After a 1-h incubation, brefeldin A (3 μg ml$^{-1}$) was added to the culture. Cells were collected, stained as described above and analyzed 5 h after the addition of brefeldin A. Antigen-specific CD8$^+$ T cells were identified as TCRβ$^+$CD8$^+$CD44$^{hi}$TNF$^+$ in Fig. 8 and Extended Data Fig. 8b,c.

## Generation of the Jedi TCR-expressing reporter cell line

NFAT-sFT reporter-containing 16.2c11 cells[53] (a kind gift of J. Kisielow, Repertoire Immune Medicines) were retrovirally transduced with constructs encoding Jedi TCR, CD8α and CD8β. Retroviral transfection was performed as described previously[77]. Cells were sorted and expanded to generate a CD8α$^+$CD8β$^+$ Jedi TCR-expressing reporter cell line.

## In vitro cocultures with Jedi TCR-expressing reporter cells

Thymic iNKT cells (gated as GFP$^+$ CD1d tetramer-binding cells for IL-4−GFP and IL-17A−GFP mice and as CD1d tetramer-binding cells for WT mice), eosinophils (gated as SiglecF$^+$SSC$^{hi}$) and DCs (gated as CD11c$^+$MHC class II$^+$) from WT, IL-4−GFP and IL-17A−GFP mice were sorted and mixed with CD8α$^+$CD8β$^+$ Jedi TCR-expressing 16.2c11 reporter cells in a 10:1 (iNKT cell:Jedi reporter cell and eosinophil:Jedi reporter cell) or 1:1 (DC:Jedi reporter cell) ratio in IMDM supplemented with 10% FCS, 50 μM 2-mercaptoethanol, 2 mM L-glutamine, 100 U ml$^{-1}$ penicillin and 100 U ml$^{-1}$ streptomycin. Cells were cultured in 96-well V-bottom plates at 37 °C and 5% CO$_2$ for approximately 20 h and analyzed by flow cytometry.

## Retroviral infection of thymic iNKT cells

Single-cell suspensions of thymocytes were stained with APC-labeled antibodies to CD24 and CD8 (anti-CD11c was added for the experiment shown in Extended Data Fig. 6e), followed by magnetic depletion with anti-APC microbeads (Miltenyi Biotec) to enrich for iNKT cells. Enriched thymic iNKT cells were cultured at a density of $1 \times 10^6$ cells per ml in IMDM supplemented with 10 ng ml$^{-1}$ IL-7 (Peprotech), 100 ng ml$^{-1}$ IL-15 (Peprotech), and 0.5 μg ml$^{-1}$ CD1d-PBS-57 tetramer. After 2 days, iNKT cells were centrifuged at 400$g$ at room temperature for 5 min and resuspended in fresh IMDM containing 10 ng ml$^{-1}$ IL-7, 100 ng ml$^{-1}$ IL-15, 4 μg ml$^{-1}$ polybrene (Sigma) and 20 μl ml$^{-1}$ concentrated virus particles, transferred to plates precoated with 20 μg ml$^{-1}$ RetroNectin (Takara) and spin infected at 1,000$g$ and 32 °C for 1 h. One more round of

infection was performed 6 h after the first round of infection. Eighteen hours after the second round of infection, the culture medium was replaced with fresh IMDM containing 10 ng ml$^{-1}$ IL-7 and 100 ng ml$^{-1}$ IL-15. Forty-eight hours after the first round of infection, iNKT cells were used in RTOC and intrathymic transfer experiments.

## Immunofluorescence microscopy of thymus sections

Organ sections were prepared as described previously[78]. For GFP detection, slides were incubated with Alexa Fluor 488-conjugated GFP booster (Chromotek), and staining was performed according to the manufacturer's instructions. For mTEC detection, slides were incubated with biotinylated UEA1 (Vector Laboratories) followed by incubation with APC-conjugated streptavidin (BioLegend) according to the manufacturer's instructions. Confocal images were acquired on an LSM 800 system (Carl Zeiss) at the Biomedicum Imaging Core at the Karolinska Institute. Images were processed using Zen 2.3 Black Edition (Carl Zeiss) or Imaris (Bitplane) imaging software.

## RTOCs

RTOCs were performed as described previously[79]. Thymi from embryonic day 14.5–15.5 CB6F1 embryos were isolated, and single-cell suspensions were prepared by digesting in PBS containing 0.125% trypsin and 1.325 mM EDTA for 15–30 min at 37 °C. In total, $6 \times 10^4$–$9 \times 10^4$ (or $1.6 \times 10^4$ for live-cell imaging) Jedi DP thymocytes (gated as CD4$^+$CD8$^+$ cells) sorted from the thymi of Jedi-TCRαβ mice (on a B10.D2 genetic background) and sorted iNKT cells (gated as CD1d tetramer-binding cells without additional gating on GFP) from the thymi of WT and IL-4– GFP or IL-17A–GFP mice or CD4SP thymocytes from WT and ACTB–GFP mice or thymic eosinophils (sorted as SiglecF$^+$SSC$^{hi}$ cells) from WT and IL-4–GFP mice were mixed with $2 \times 10^5$–$3 \times 10^5$ cells from digested fetal CB6F1 thymi. RTOCs comparing i.c.GFP and secGFP were established as described above but with sorted Thy1.1$^+$ thymic iNKT cells (on a CB6F1 background) transduced with the indicated constructs. RTOCs with OT-I cells were established as described above, but digested embryonic day 15.5 WT thymi were reaggregated with OT-I DP thymocytes (sorted as CD4$^+$CD8$^+$ cells) and sorted GFP$^+$ cOVA-IRES-GFP-transduced thymic iNKT cells (all on a C57BL/6J background). Cell pellets were resuspended in less than 1.5 µl of RPMI 1640 (supplemented with 10% FCS, 50 µM 2-mercaptoethanol, 10 mM HEPES, 2 mM L-glutamine, 1× nonessential amino acids, 1 mM sodium pyruvate, 100 U ml$^{-1}$ penicillin and 100 U ml$^{-1}$ streptomycin) per RTOC. The cell suspension was transferred onto the membrane (Millipore Millicell 0.4-µm cell culture inserts) and placed in a six-well plate (Sarstedt) filled with the same medium as described above. For eosinophil-containing RTOCs, 5 ng ml$^{-1}$ IL-33 (Peprotech) was added to the culture medium. For CD80/CD86 blocking experiments, 10 µg ml$^{-1}$ anti-CD80 (16-10A1, BioLegend) and anti-CD86 (GL-1, BioLegend) or the corresponding isotype controls (HTK888, BioLegend; RTK2758, BioLegend) were added to the RTOC culture medium. Reaggregation and further culture were performed in humidified chambers placed in the cell culture incubator at 37 °C and 5% CO$_2$. Flow cytometric analysis was performed on day 5 of the culture. Live-cell imaging was performed 16–20 h after setup.

## RTOC live-cell imaging

For live-cell imaging, RTOCs were prepared as described above with minor modifications. Jedi DP thymocytes (gated as CD4$^+$CD8$^+$PD-1$^{lo/int}$ cells) isolated from the thymi of Jedi mice and iNKT cells (gated as GFP$^+$CD1d tetramer-binding cells) from the thymi of IL-4–GFP mice were sorted. After sorting, Jedi DP thymocytes were loaded with 5 µM Cal-520 AM (AAT Bioquest) in HBSS containing 2% FCS for 1 h at 37 °C. DP Jedi cells were then washed twice with FCS-containing RPMI 1640 culture medium and labeled with 2.5 µM CTV (Thermo Fisher) in PBS/0.5% BSA for 8 min in a water bath at 37 °C, while thymic iNKT cells were labeled with 2.5 µM CTY (Thermo Fisher) under the same labeling conditions. RTOCs were allowed to assemble for 16–20 h before transfer

to compartmentalized glass-bottom dishes (ibidi). For imaging, RTOCs were first embedded in PureCol EZ Gel solution (Sigma-Aldrich) for 45 min at 37 °C. RTOC complete medium was then added on top of the collagen matrix. z-stacks of two RTOCs were first imaged for each condition at a rate of 1 frame per min for 1–3 h using a ×20/0.8-NA Plan-Apochromat objective and a fast Airyscan detector on an LSM880 microscope (Carl Zeiss) with the incubation chamber set to 37 °C and 5% CO$_2$. Afterward, longer acquisitions of 12 h at 1 frame per min were conducted using standard confocal settings.

## RTOC live-cell imaging analysis

Maximum intensity projections of confocal datasets were generated and analyzed in Zen Lite software (Carl Zeiss). Cell contacts were defined by colocalization of an iNKT cell and a Jedi thymocyte for at least three frames (one frame = 1 min). Contact identified using maximum intensity projections was then validated using the full three-dimensional datasets. Consecutive detachment and attachment between two cells with short intervals (shorter than five frames) were counted as one contact. In total, 134 contacts (for RTOCs with WT iNKT cells) and 108 contacts (for RTOCs with IL-4–GFP iNKT cells) were analyzed manually for an increase in Cal-520 AM fluorescence to enumerate Ca$^{2+}$ signaling events (Fig. 6e). For quantification of contact length, the number of frames where cell colocalization had been observed was counted. Contacts where cells could not be tracked throughout the entire contact (for example, cells masked by other cells and cells disappearing from the field of view) were excluded from this analysis. Contacts lasting for 2 frames or less were also excluded from quantification of contact length.

For quantification of Cal-520 AM signal intensity over time, ten contacts from each group (WT iNKT cells, IL-4–GFP iNKT cells with Ca$^{2+}$ signaling detected and IL-4–GFP iNKT cells with no Ca$^{2+}$ signaling detected) were randomly selected and analyzed in ImageJ. The Cal-520 AM signal intensity and CTV signal intensity in Jedi cells were quantified by measuring the mean intensity of each signal in the region of interest (ROI; corresponding to a Jedi cell) drawn manually for each frame. For each frame, Ca$^{2+}$ signal intensity of the Jedi cell was calculated as Ca$^{2+}$ signal intensity (per frame) = (mean intensity of Cal-520 AM in ROI)/(mean intensity of CTV in ROI). The baseline Ca$^{2+}$ signal intensity for each Jedi cell was then calculated by averaging the Ca$^{2+}$ signal intensity (per frame) for five frames right before contact initiation. Normalized Ca$^{2+}$ signal intensities plotted in Fig. 6f and Extended Data Fig. 6g were calculated for each frame using the following equation: normalized Ca$^{2+}$ signal intensity = (Ca$^{2+}$ signal intensity (per frame))/(baseline Ca$^{2+}$ signal intensity).

## Intrathymic injections

Intrathymic injections were performed as previously described[24]. Four- to 5-week-old Jedi-TCRβ recipient mice were anesthetized by intraperitoneal injection of a ketamine (80 mg per kg (body weight))/xylazine (8 mg per kg (body weight)) solution. Subcutaneous injection of carprofen (5 mg kg$^{-1}$ (body weight)) was performed 30–60 min before surgery. When fully anesthetized, the fur was shaved, and a small incision in the skin above the sternum was made. For experiments shown in Fig. 5d, up to $1 \times 10^6$ iNKT cells (gated as CD1d tetramer-binding cells without additional gating on GFP) sorted from the thymi of IL-4–GFP mice were resuspended in 10 µl of PBS, and 5 µl per thymus lobe was injected. For experiments shown in Extended Data Fig. 6e, total cells from Bcl-2-P2A-Thy1.1-IRES-secGFP transduction cultures containing $6 \times 10^5$ transduced (Thy1.1$^+$) iNKT cells were resuspended in 10 µl of PBS, and 5 µl per thymus lobe was injected. Bcl-2 was added to the construct to improve survival of intrathymically transferred iNKT cells. The skin incision was closed with sutures, and mice were intraperitoneally injected with atipamezole (4 mg per kg (body weight)). Carprofen (5 mg per kg (body weight)) was subcutaneously injected once a day until 72 h after surgery. Flow cytometric analysis was performed 7 days after injection.

**Autoantibody profiling using bead-based protein microarrays**

Serum from *PLZF*<sup>lu/lu</sup> mice was tested for the presence of antibodies against autoantigens and proteins from pathogens. A custom microbead-based antigen array was created to profile serum samples for antibodies against cytokines, autoimmune-associated antigens and viral antigens, as previously described[80]. The array was constructed by conjugating antigens to uniquely barcoded carboxylated magnetic beads (MagPlex-C, Luminex). Beads were qualified using prototype plasma or serum samples with known reactivities. Although the available array was constructed using human antigens, protein homology with mouse antigens still allowed for comparisons in mouse autoantibody development[81,82]. Mouse serum samples were tested at a dilution of 1:100 in 0.05% PBS-Tween supplemented with 1% (wt/vol) BSA. Bound antibody was detected using R-PE-conjugated Fcγ-specific goat anti-mouse IgG F(ab')₂ fragment (Jackson ImmunoResearch, 115-116-071) before analysis using a FlexMap3D instrument (Luminex). Binding events were displayed as median fluorescence intensity (MFI). For normalization, the MFI value for the 'bare bead' ID (no conjugated antigen) was subtracted from the MFI value for the antigen-conjugated bead ID. Samples were run in duplicate. Serum samples from MR/*Fas*<sup>lpr/lpr</sup> mice (a mouse model for systemic lupus erythematosus) served as positive controls.

**Analysis of public expression datasets**

AnnData objects containing previously described human and mouse thymus cell atlases[35] were downloaded from https://developmental.cellatlas.io/thymus-development and were replotted using Scanpy[83] (version 1.9.1). The combined human effector T cell signature used in Extended Data Fig. 1 was generated using the sc.tl.score_genes Scanpy function and the following list of effector program genes: *IL13*, *IL-17A*, *IL17F*, *IFNG*, *GZMB* and *GZMK*. Analysis of the Immgen dataset was described previously[33].

**Statistical analysis**

Statistical analyses were performed with GraphPad Prism 9 software. Error bars represent standard deviation of biological replicates (or individual RTOCs) unless stated otherwise. Two-tailed unpaired Student's *t*-tests or two-tailed Mann–Whitney tests were used to assess the statistical significance of one observed parameter between two experimental groups. An unpaired ANOVA with a Holm–Sidak multiple comparisons test or a Kruskal–Wallis test was used when more than two experimental groups were compared.

**Reporting summary**

Further information on research design is available in the Nature Portfolio Reporting Summary linked to this article.

## Data availability

Source data are provided with this paper.

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

## Acknowledgements

We thank M. Aichinger and G. Wirnsberger for a discussion that facilitated the initiation of this project. We thank the National Institutes of Health tetramer core facility for the preparation of all tetramers used in this study. This study was supported by the Swedish Research Council (grants 2017-01118 and 2021-01468 to T.K.), Cancerfonden (grants CAN 2018/710 and 21 1602 Pj to T.K.), Barncancerfonden (grants PR2021-017 and PR2023-0091 to T.K.), Radiumhemmets Forskningsfonder (grants 211192 and 231233 to T.K.), a stipend from the German Research Foundation (DU 1964/1-1 to J.D.) and a stipend from the China Scholarship Council (to Y.Y.). 16.2c11 reporter cells were provided by J. Kisielow (Repertoire Immune Medicines). We thank S. Edwards and the Advanced Light Microscopy facility at Science for Life Laboratory for their help setting up the imaging of the RTOCs. We thank J. Coquet, S. Nylén and A. Gigliotti Rothfuchs (Karolinska Insitutet) and members of their laboratories for their help with experiments that were not included in the manuscript. We thank all members of the Kreslavsky laboratory for their support, discussions of the project and critical reading of the manuscript.

## Author contributions

Y.Y. performed most of the experiments. J.D. designed and supervised some of the experiments and performed most of the intrathymic transfer experiments and the initial RTOC experiments. K.Y. contributed to the RTOC and peptide immunization experiments. P.A.S. and B.Ö. performed RTOC imaging. N.R.C. and C.G. provided expertise and training for the intrathymic transfer experiments. I.S. and I.P. performed the initial experiments with IL-17A–GFP mice. E.D. provided *PLZF*<sup>lu/lu</sup> mice and collected sera from these animals. J.A. and B.D.B. generated and provided Jedi mice. R.D.S. and W.K. performed analyses of *Tcrd*<sup>−/−</sup>*Cd1d*<sup>−/−</sup>*Mr1*<sup>−/−</sup> mice. E.Y. and P.J.U. performed antigen array experiments. A.R. and K.Y. analyzed the data and prepared the figures. T.K. suggested the project idea, supervised the study and wrote the manuscript. Y.Y., J.D., K.Y., A.R., C.G. and T.K. edited the manuscript.

## Funding

## Competing interests

The authors declare no competing financial interests.

## Additional information

**Extended data** is available for this paper at

**Supplementary information** The online version
contains supplementary material available at

**Correspondence and requests for materials** should be addressed to
Taras Kreslavsky.

**Peer review information** *Nature Immunology* thanks Ludger Klein
and Jung Hyun Park for their contribution to the peer review of this
manuscript. Primary Handling Editor: S. Houston, in collaboration with
the *Nature Immunology* team. Peer reviewer reports are available.

**Combined effector T cell signature**

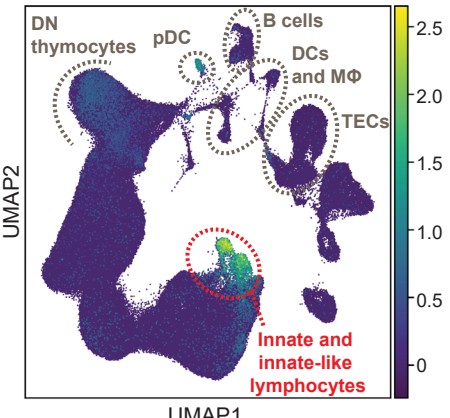

**Effector T cell program-related genes**

*IFNG*  *IL17A*  *IL4*  *GNLY*  *EOMES*

*TBX21*  *IL17F*  *IL13*  *GZMB*  *ZBTB16*

**MHC class II-encoding genes**

*HLA-DPA1*  *HLA-DPB1*  *HLA-DRA*  *HLA-DRB1*  *HLA-DQB1*

**Extended Data Fig. 1 | Human thymic innate and innate-like lymphocytes express a broad spectrum of inflammation-associated self-antigens.** UMAP highlighting the expression of combined effector T cell gene signature (generated as describe in Methods) (top) and UMAPs highlighting the expression of the indicated genes in the human thymus cell atlas scRNA-seq dataset.

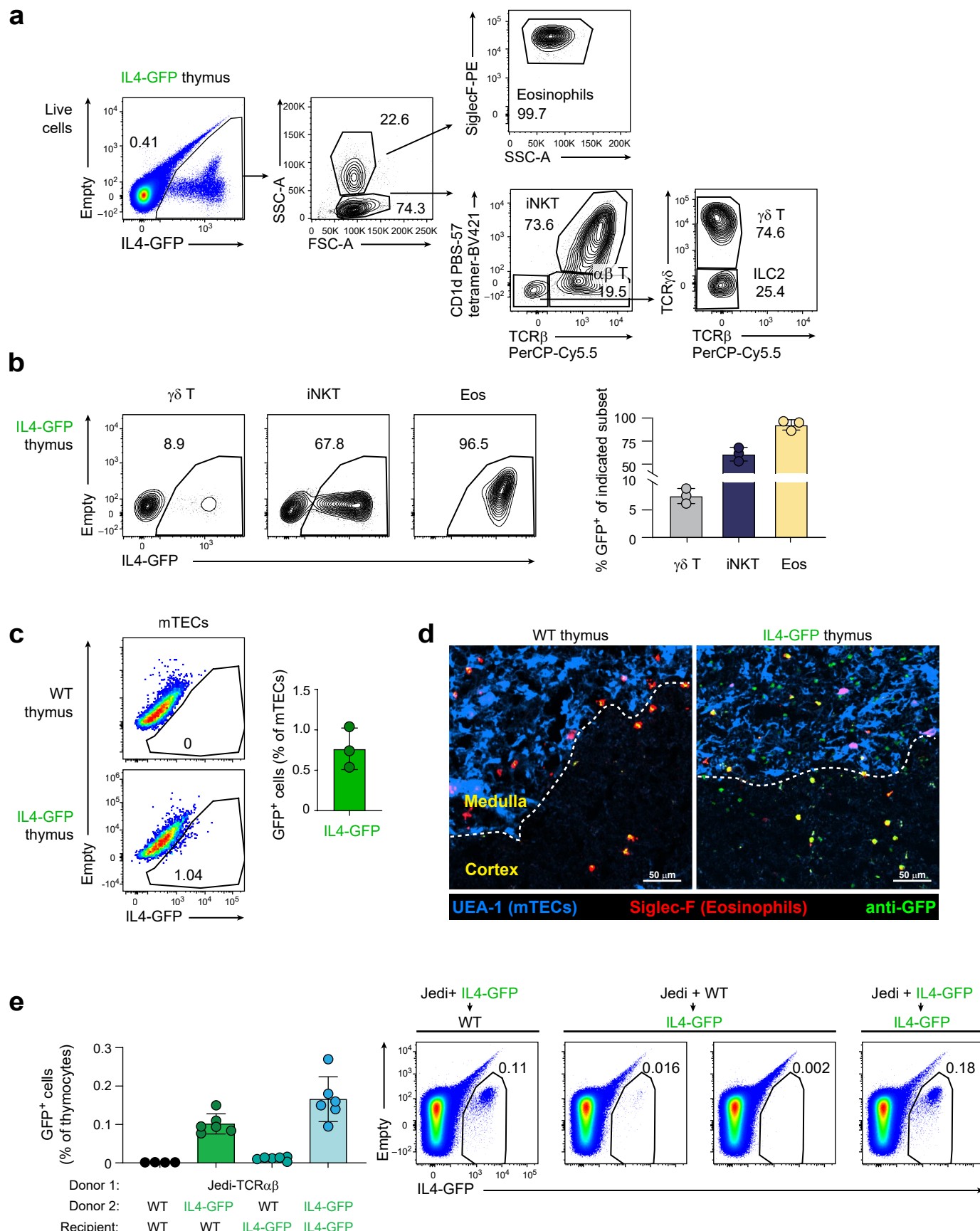

**Extended Data Fig. 2 | See next page for caption.**

**Extended Data Fig. 2 | Characterization of GFP-expressing cell types in IL-4-GFP thymi. a**. Gating strategy for characterization of GFP-expressing cell types in IL4-GFP thymi. **b**. Frequency of GFP$^+$ cells in the indicated thymic subsets in IL4-GFP mice (n = 3 mice). **c**. Frequency of GFP$^+$ cells among mTECs (gated as EPCAM$^+$CD45$^-$UEA1$^+$Ly51$^-$) in IL4-GFP mice (n = 3 mice). **d**. Images of IL4-GFP and WT thymi sections as analyzed by confocal immunofluorescence microscopy as in Fig. 3b. An area of the same image as in Fig. 3b is shown for the IL4-GFP mouse. **e**. Frequency of GFP$^+$ hematopoietic cells in the thymi of mixed BM chimeras shown in Fig. 3c, e. Quantification (left) and representative flow cytometric plots (right) are shown (n = 4 chimeras for [WT+Jedi-TCRαβ]→WT thymi, n = 6 for the other groups). Data are presented as the mean ± s.d.

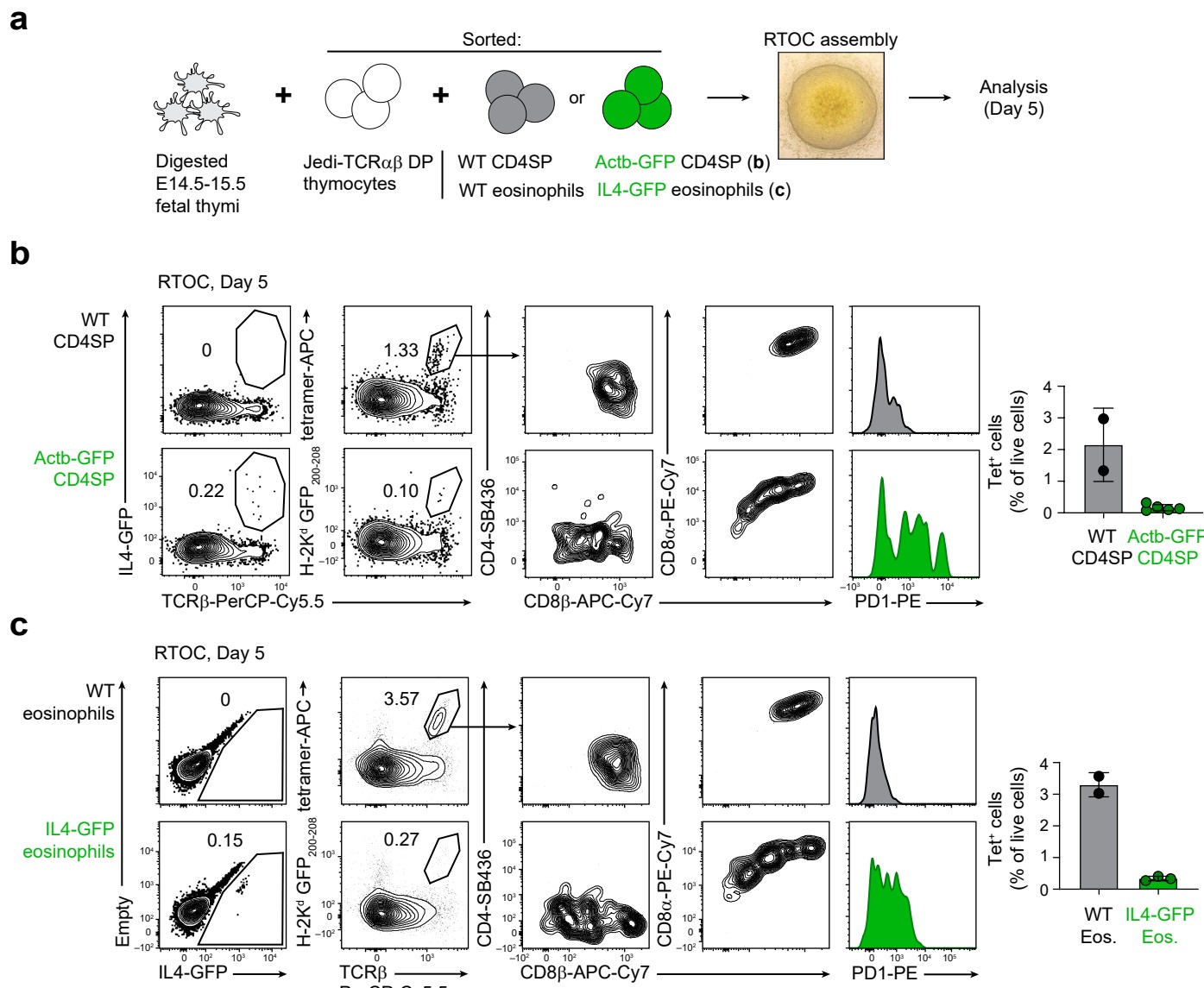

**Extended Data Fig. 3 | Antigen expression exclusively by small numbers of conventional thymocytes or thymic eosinophils is sufficient to induce elimination of autoreactive thymocytes. a**. Schematic representation of RTOC experiments shown in b-c. **b-c** RTOCs with Jedi-TCRαβ DP thymocytes established and analyzed as in Fig. 4a, b but with WT or Actb-GFP CD4SP thymocytes (b) and WT or IL4-GFP eosinophils (c) used as antigen source. Representative plots showing frequencies of GFP-expressing cells, frequencies of H2-K$^d$ GFP$_{200-208}$ tetramer-binding Jedi thymocytes (quantification shown on the right) and expression of the indicated cell surface markers by the latter cells. Representative results of three independent experiments (b) (n = 2 RTOCs for WT CD4SP, n = 5 for Actb-GFP CD4SP) or results of one experiment (c) (n = 2 for WT eosinophils, n = 3 for IL4-GFP eosinophils). Data are presented as the mean ± s.d.

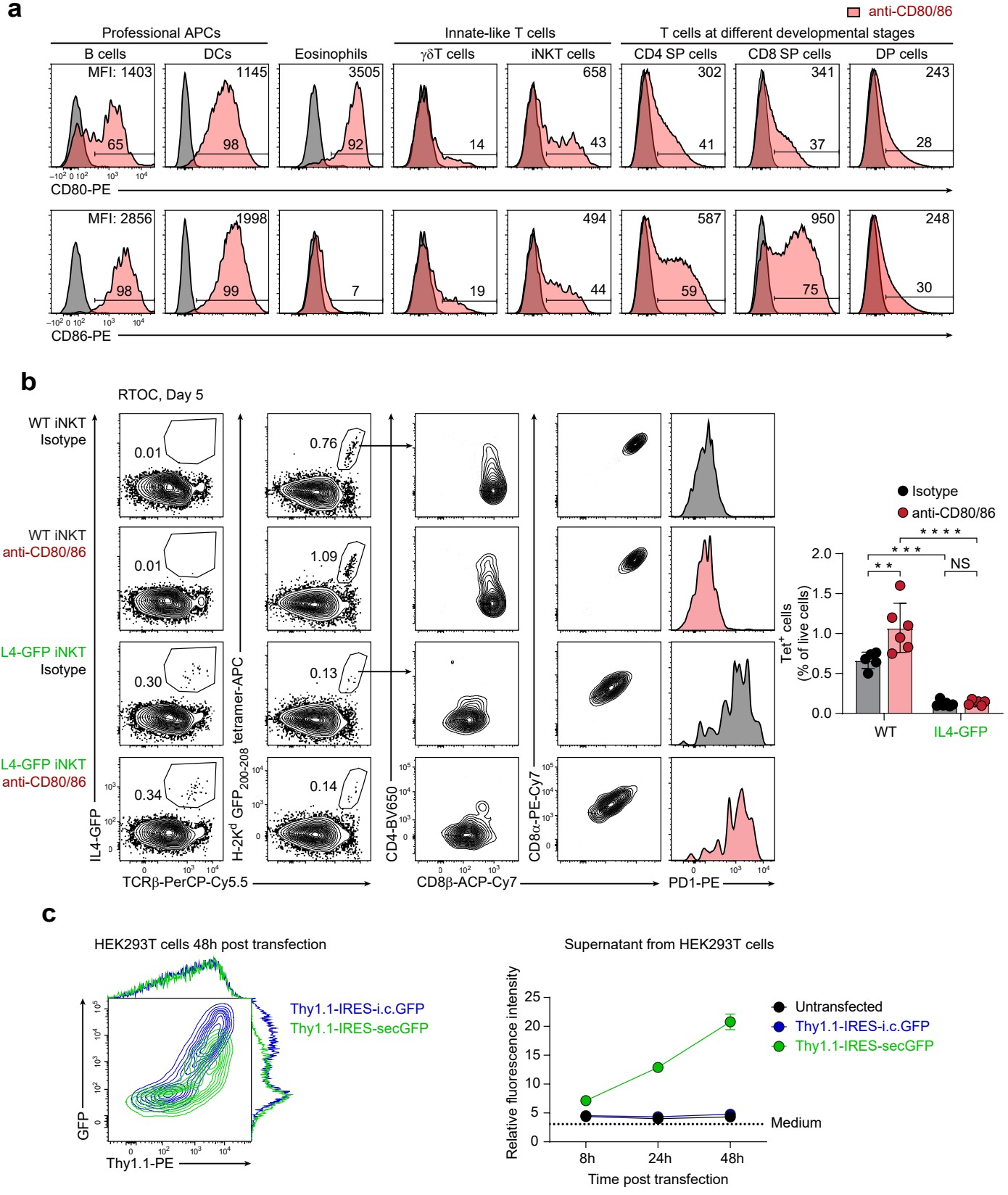

**Extended Data Fig. 4 | See next page for caption.**

**Extended Data Fig. 4 | Assessment of possible effects of co-stimulation on induction of tolerance by innate-like T cells. a.** Expression of CD80 (top) and CD86 (bottom) by the indicated thymic populations (red) in an IL4-GFP mouse. Isotype control for CD80 is shown in grey (same in both rows). Percentage of positive cells and median fluorescent intensity (MFI) in the positive gate are indicated. The following thymic populations were analyzed: B cells (CD19⁺MHC-II⁺), DCs (CD11c⁺MHC-II⁺), eosinophils (IL4-GFP⁺SSC^hi), γδT cells (CD3⁺TCRγδ⁺), iNKT (TCRβ⁺CD1d-PBS-57 tetramer⁺), CD4SP (CD4⁺), CD8SP (CD8⁺), and DP cells (CD4⁺CD8⁺). **b.** RTOCs were established and analyzed as in Fig. 4b, but blocking antibodies against CD80 and CD86 or isotype control antibodies were added to culture medium. Representative results of two

independent experiments (n = 5 RTOCs for WT isotype group, n = 6 for the other groups). Data are presented as the mean ± s.d. with NS: non-significant ($P > 0.05$), $**P < 0.01$, $***P < 0.001$, and $****P < 0.0001$. Data were analyzed by two-way ANOVA with Holm-Sidak's multiple comparisons test. **c.** HEK293T cells were transfected with Thy1.1-IRES-i.c.GFP or Thy1.1-IRES-secGFP retroviral constructs. Expression of Thy1.1 and GFP by HEK293T cells 48 hours after transfection (left, contour plot overlay as well as histogram overlays for Thy1.1 and GFP are shown) and the mean value of GFP fluorescence in culture supernatants of the same cells at different timepoints (right) are shown. Error bars represent s.d. of technical replicates (n = 2 for untransfected group, n = 3 for the other groups).

**a**

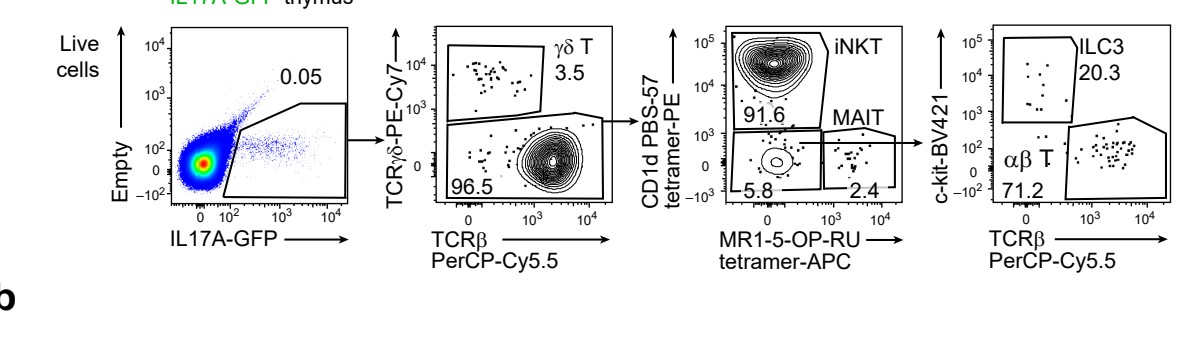

**b**

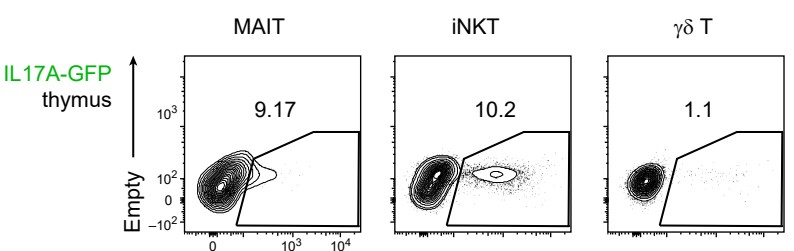

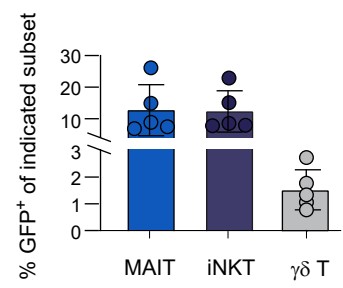

**c**

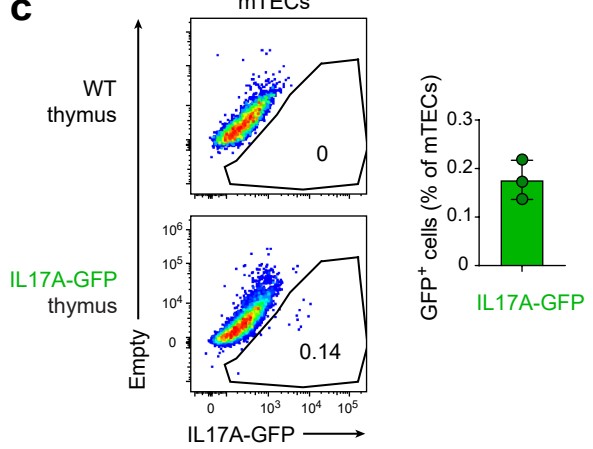

**d**

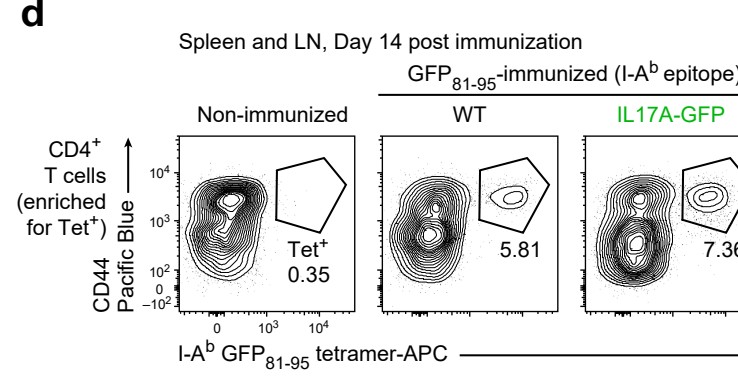

**e**

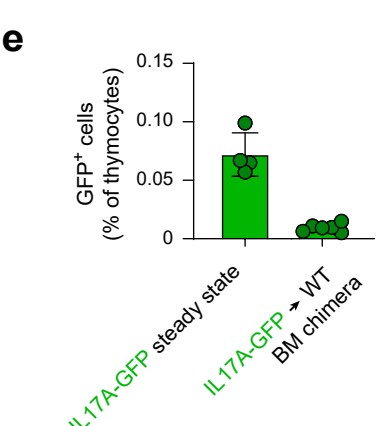

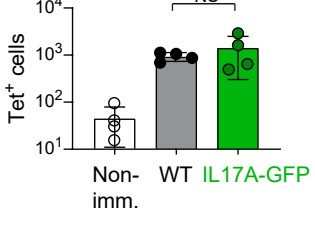

**Extended Data Fig. 5 | See next page for caption.**

**Extended Data Fig. 5 | Characterization of GFP-expressing cell types in IL17A-GFP thymi. a**. Gating strategy for characterization of GFP-expressing cell types in IL17A-GFP thymi. **b**. Frequency of GFP⁺ cells in the indicated thymic subsets in IL17A-GFP mice (n = 5 mice). **c**. Frequency of GFP⁺ cells among mTECs (gated as EPCAM⁺CD45⁻UEA1⁺Ly51⁻) from IL17A-GFP mice (n = 3 mice). **d**. WT and IL17A-GFP mice (on C57BL/6J background) were immunized s.c. with GFP$_{81-95}$ peptide in CFA or left unimmunized (WT only). 14 days after immunization, I-A$^b$ GFP$_{81-95}$ tetramer-binding cells were magnetically enriched from pooled lymph nodes and spleens and quantified by flow cytometry. Representative flow cytometry plots gated on CD4 T cells (left) and absolute numbers of CD4⁺CD44⁺Tetramer⁺ T cells (right) are shown. Representative results of two independent experiments (n = 4 mice per group). **e**. Comparison of the frequencies of GFP⁺ hematopoietic cells in the thymi of IL17A-GFP mice and in IL17A-GFP→WT BM chimeras (analyzed ≥ 7 weeks after reconstitution) (n = 4 for IL17A-GFP mice, n = 6 for chimeras). Data are presented as the mean ± s.d. with NS: non-significant (P > 0.05). Data were analyzed by two-tailed Student's t test (d).

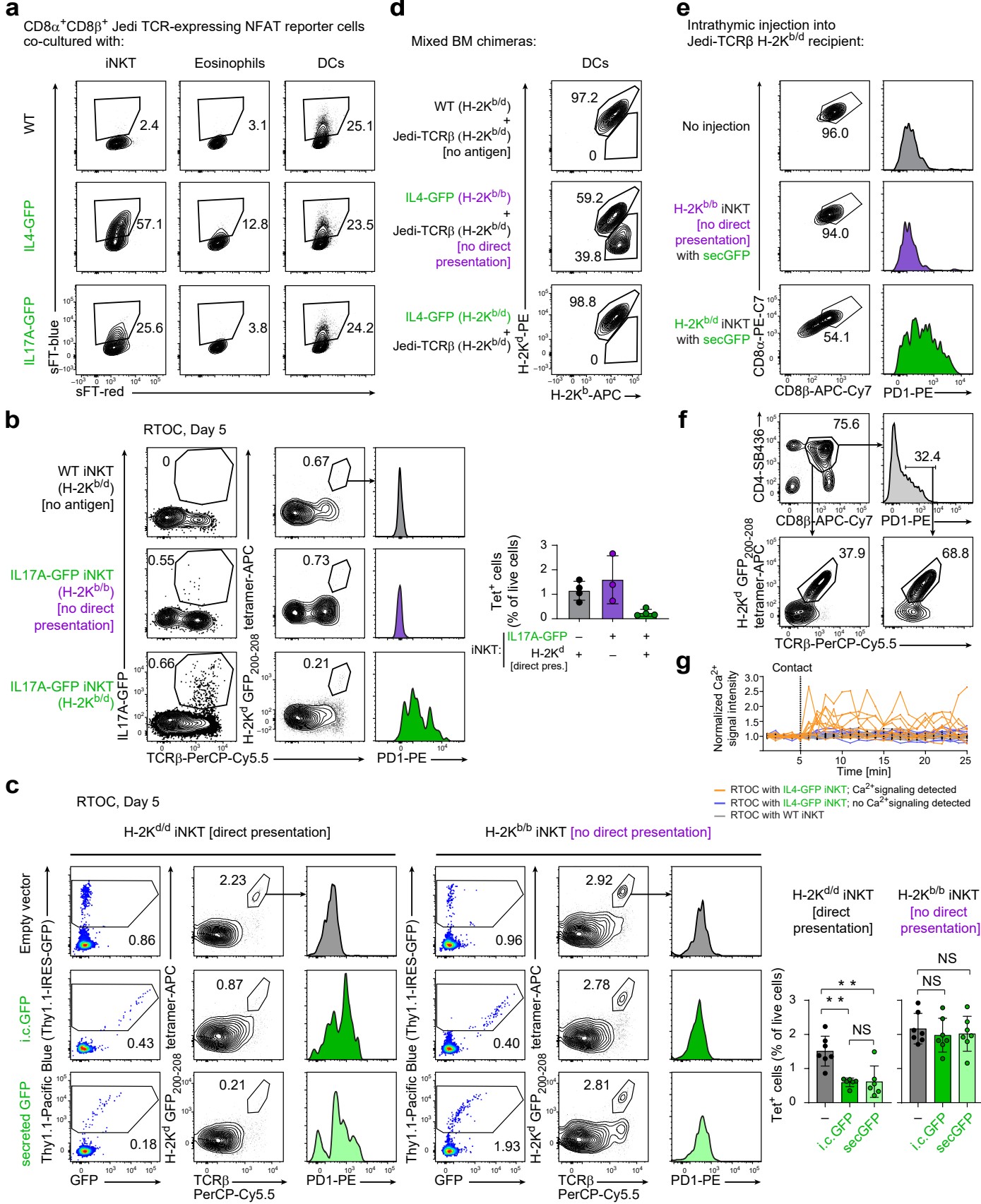

**Extended Data Fig. 6 | See next page for caption.**

**Extended Data Fig. 6 | Direct presentation by model antigen-expressing populations rather than cross-presentation by professional APCs is responsible for elimination of autoreactive GFP-specific CD8 thymocytes.**
**a**. iNKT cells, eosinophils and DCs were sorted from thymi of WT, IL4-GFP and IL17A-GFP mice and co-cultured overnight with slow fluorescent timer (sFT) NFAT reporter-containing 16.2c11 cells that were engineered to express the Jedi TCR, CD8α and CD8β. Activation of NFAT signaling was measured by sFT-Blue reporter upregulation. **b**. RTOCs with sorted Jedi DP thymocytes and iNKT cells were established and analyzed as in Fig. 6a, but using iNKT cells sorted from WT mice (on H2$^{b/d}$ background) or from IL17A-GFP mice on H2$^{b/d}$ and H2$^{b/b}$ backgrounds. Results of one experiment (n = 3 RTOCs for IL17A-GFP H-2$^{b/b}$ group, n = 4 for the other groups). **c**. Thymic iNKT cells from H2$^{d/d}$ (BALB/c background) or H2$^{b/b}$ (C57BL/6J background) mice were transduced with Thy1.1-IRES-i.c.GFP, Thy1.1-IRES-secGFP and Thy1.1-only retroviruses. Sorted Thy1.1$^+$ iNKT cells were added to RTOCs together with Jedi-TCRαβ DP thymocytes. Flow cytometric analysis

and quantification are shown. Results of one experiment (n = 6 RTOCs for H-2$^{d/d}$ i.c.GFP and H-2$^{d/d}$ secGFP groups, n = 7 for the other groups). **d**. Mixed BM chimeras were established as described in Fig. 7b. Expression of H-2K$^b$ and H2-K$^d$ on thymic DCs (gated as CD11c$^+$MHC II$^+$) was analyzed by flow cytometry. **e**. Thymic iNKT cells from H2$^{b/d}$ (CB6F1 background) or H2$^{b/b}$ (C57BL/6J background) mice were transduced with Bcl2-P2A-Thy1.1-IRES-secGFP-encoding retroviruses and injected into the thymi of Jedi-TCRβ mice (on CB6F1 background). Expression of CD8α, CD8β and PD1 on CD4$^-$ H2-K$^d$ GFP$_{200-208}$ tetramer-binding thymocytes is shown 7 days after transfer. **f**. Gating strategy showing utilization of low levels of PD1 as a surrogate marker for TCR expression on DP thymocytes from Jedi-TCRαβ mice. **g**. Normalized Ca$^{2+}$ signaling intensity in Jedi thymocytes interacting with iNKT cells as in Fig. 6f, but values for individual cells are shown. Data are presented as the mean ± s.d. with NS: non-significant ($P > 0.05$) and **$P < 0.01$. Data were analyzed by one-way ANOVA with Holm-Sidak's multiple comparisons test.

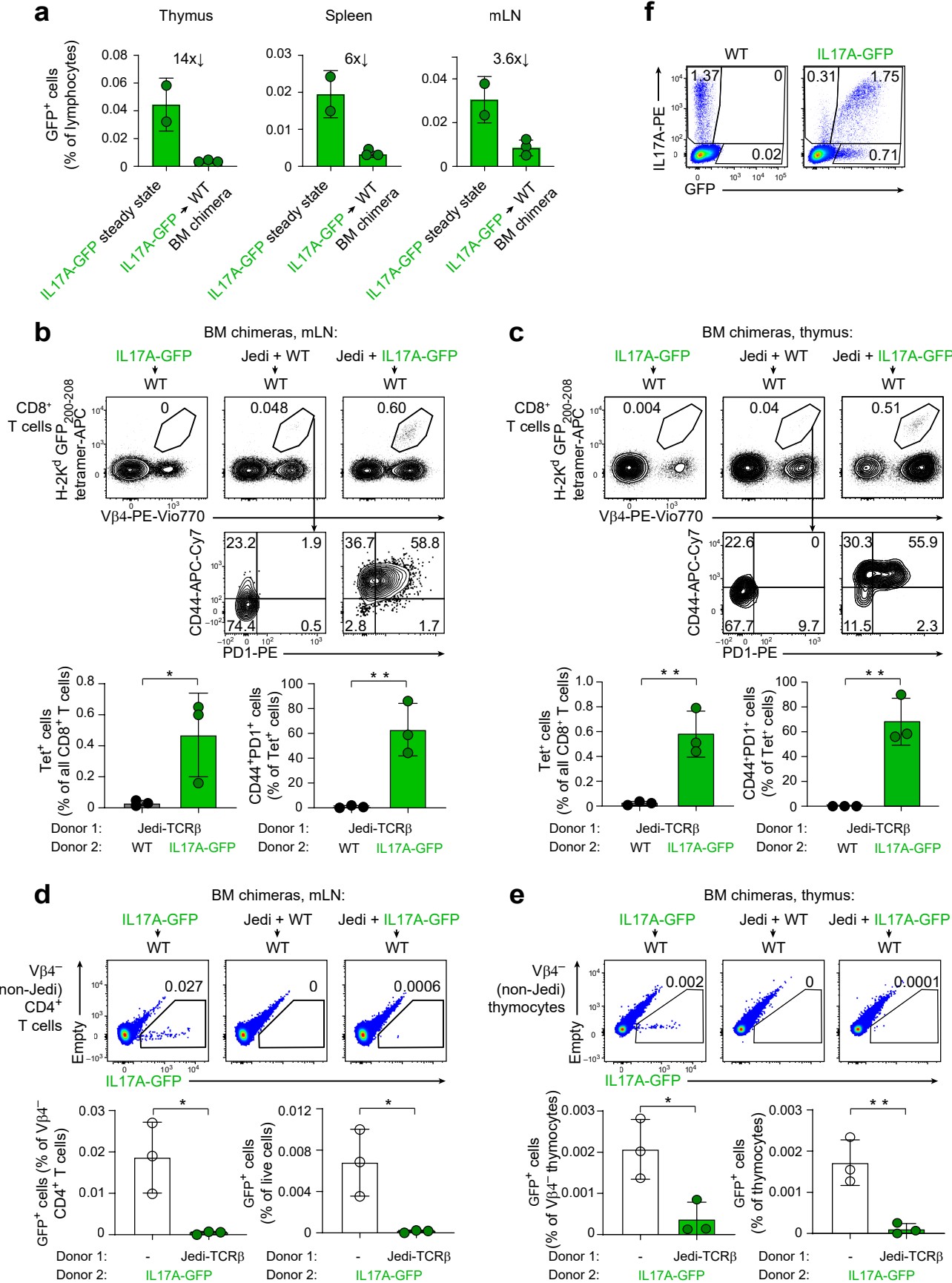

**Extended Data Fig. 7 | See next page for caption.**

**Extended Data Fig. 7 | Decreased thymic abundance of IL17A-GFP results in autoimmune elimination of GFP⁺ T_H17 cells. a**. Comparison of the frequencies of GFP⁺ lymphocytes in the thymi, spleens and mLNs of IL17A-GFP mice and in IL17A-GFP→WT BM chimeras (analyzed 8 weeks after reconstitution). One experiment with 2 IL17A-GFP mice and 3 BM chimeras. **b**–**e**. WT recipient mice were irradiated and transferred i.v. with T- and NK-cell depleted BM cells from either IL17A-GFP mice, or Jedi-TCRβ mice and WT mice, or Jedi-TCRβ mice and IL17A-GFP mice (all on CB6F1 background). Mesenteric lymph nodes (mLN) (b, d) and thymi (c, e) were analyzed 8 weeks after reconstitution. Same experiment

as in Fig. 7a–c. b, c. Frequency and cell surface phenotype of H2-K$^d$ GFP$_{200-208}$ tetramer-binding CD8 T cells. d, e. Frequency of IL17A-GFP expressing cells among total lymphocytes and among Vβ4⁻ (non-Jedi) CD4 T cells. Representative results of two independent experiments (n = 3 chimeras per group). **f**. T_H17 cells were differentiated in vitro from WT and IL17A-GFP CD4 T cells as in Fig. 7d. Intracellular expression of IL17A cytokine and GFP was analyzed after a short-term PMA/Ionomycin restimulation. Data are presented as the mean ± s.d. with *P < 0.05 and **P < 0.01. Data were analyzed by two-tailed Student's t test.

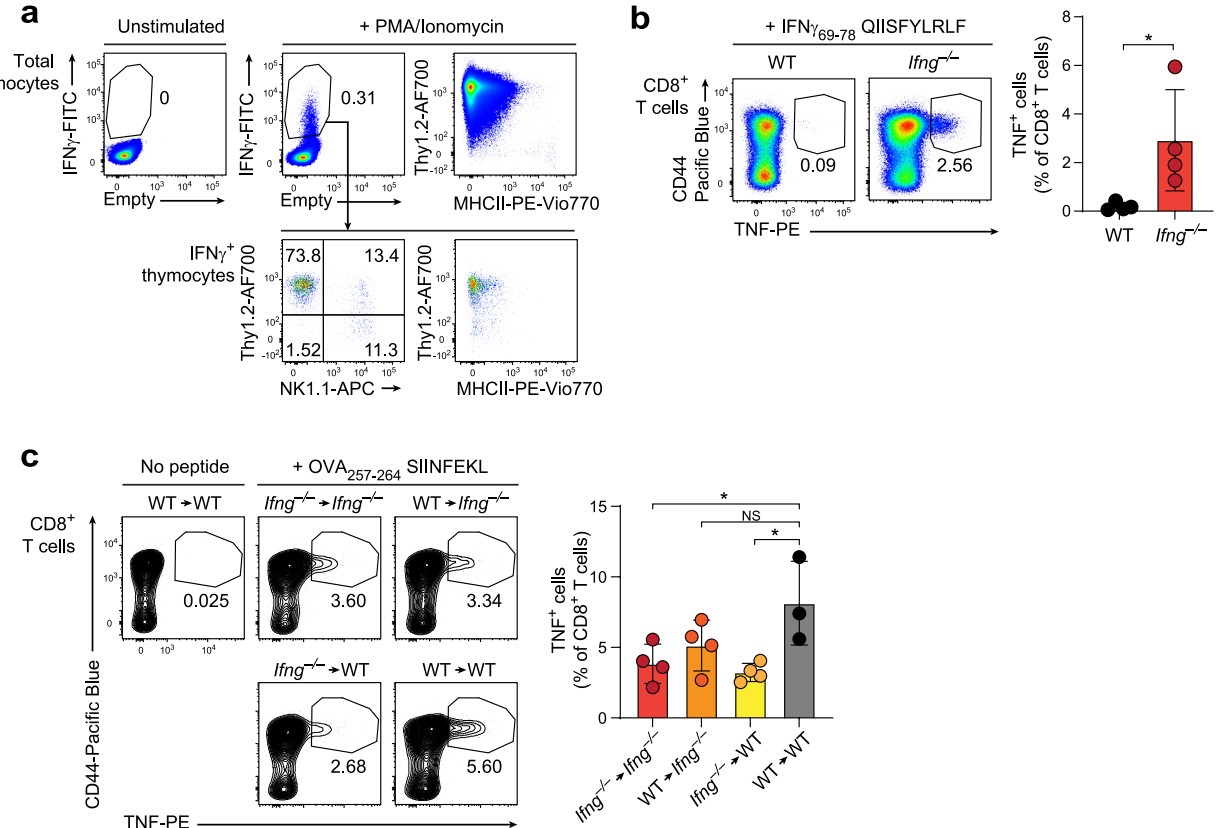

**Extended Data Fig. 8 | IFNγ as an endogenous T/NK/ILC-derived inflammation-associated self-antigen. a.** Expression of Thy1, NK1.1 and MHC II by IFNγ$^+$ and total WT thymocytes after PMA/Ionomycin stimulation. **b**. WT and *Ifng*$^{-/-}$ mice (on C57BL/6J background) were immunized with IFNγ$_{69-78}$ peptide using anti-CD40 and Poly(I:C) as adjuvant. 6 days after immunization, splenocytes were stimulated with IFNγ$_{69-78}$ peptide for 6 hours as described in Methods and production of TNF by CD8 T cells was analyzed by flow cytometry. **c**. WT and *Ifng*$^{-/-}$ mice (on C57BL/6J background) were lethally irradiated and reconstituted with syngeneic WT or *Ifng*$^{-/-}$ BM cells depleted of T- and NK-cells.

11 weeks after reconstitution, the resulting four groups of BM chimeras were immunized with IFNγ$_{69-78}$ and OVA$_{257-264}$ peptides using anti-CD40 and Poly(I:C) as adjuvant. 6 days after immunization, splenocytes were stimulated with OVA$_{257-264}$ peptide for 6 hours as described in Methods and production of TNF by CD8 T cells was analyzed by flow cytometry (n = 3 for WT→WT, n = 4 for the other groups). Representative results of two independent experiments. Data are presented as the mean ± s.d. with NS: non-significant (*P* > 0.05) and *\*P* < 0.05. Data were analyzed by two-tailed Student's t test (b) or one-way ANOVA with Holm-Sidak's multiple comparisons test (c).

# Reporting Summary

## Statistics

For all statistical analyses, confirm that the following items are present in the figure legend, table legend, main text, or Methods section.

| n/a | Confirmed | |
|---|---|---|
| ☐ | ☒ | The exact sample size (*n*) for each experimental group/condition, given as a discrete number and unit of measurement |
| ☐ | ☒ | A statement on whether measurements were taken from distinct samples or whether the same sample was measured repeatedly |
| ☐ | ☒ | The statistical test(s) used AND whether they are one- or two-sided<br>*Only common tests should be described solely by name; describe more complex techniques in the Methods section.* |
| ☐ | ☒ | A description of all covariates tested |
| ☐ | ☒ | A description of any assumptions or corrections, such as tests of normality and adjustment for multiple comparisons |
| ☐ | ☒ | A full description of the statistical parameters including central tendency (e.g. means) or other basic estimates (e.g. regression coefficient) AND variation (e.g. standard deviation) or associated estimates of uncertainty (e.g. confidence intervals) |
| ☐ | ☒ | For null hypothesis testing, the test statistic (e.g. *F*, *t*, *r*) with confidence intervals, effect sizes, degrees of freedom and *P* value noted<br>*Give P values as exact values whenever suitable.* |
| ☒ | ☐ | For Bayesian analysis, information on the choice of priors and Markov chain Monte Carlo settings |
| ☒ | ☐ | For hierarchical and complex designs, identification of the appropriate level for tests and full reporting of outcomes |
| ☒ | ☐ | Estimates of effect sizes (e.g. Cohen's *d*, Pearson's *r*), indicating how they were calculated |

*Our web collection on statistics for biologists contains articles on many of the points above.*

## Software and code

Policy information about availability of computer code

| Data collection | LSR Fortessa (BD Biosciences), FACSCanto™ II (BD Biosciences), Cytek Aurora (Cytek Biosciences), BD FACS Diva, Varioskan LUX Multimode Microplate Reader (ThermoFisher), LSM 800 system (Carl Zeiss), LSM880 microscope (Carl Zeiss), FlexMap3D instrument (Luminex Corp.). |
|---|---|
| Data analysis | FlowJo software v.10 (BD), Microsoft Office, Zen 2.3 Black Edition (Carl Zeiss), Imaris 10.1.0 (Bitplane), Scanpy 1.9.1, GraphPad Prism 9. |

For manuscripts utilizing custom algorithms or software that are central to the research but not yet described in published literature, software must be made available to editors and reviewers. We strongly encourage code deposition in a community repository (e.g. GitHub). See the Nature Portfolio guidelines for submitting code & software for further information.

## Data

Policy information about availability of data

All manuscripts must include a data availability statement. This statement should provide the following information, where applicable:
- Accession codes, unique identifiers, or web links for publicly available datasets
- A description of any restrictions on data availability
- For clinical datasets or third party data, please ensure that the statement adheres to our policy

Microarray datasets for Il4, Ifng, Gzmb expression by thymic cell subsets were obtained from ImmGen Consortium. Human and mouse thymus cell atlases were obtained from https://developmental.cellatlas.io/thymus-development.

# Human research participants

Policy information about studies involving human research participants and Sex and Gender in Research.

| | |
|---|---|
| Reporting on sex and gender | Not applicable |
| Population characteristics | Not applicable |
| Recruitment | Not applicable |
| Ethics oversight | Not applicable |

Note that full information on the approval of the study protocol must also be provided in the manuscript.

# Field-specific reporting

Please select the one below that is the best fit for your research. If you are not sure, read the appropriate sections before making your selection.

☒ Life sciences          ☐ Behavioural & social sciences          ☐ Ecological, evolutionary & environmental sciences

For a reference copy of the document with all sections, see nature.com/documents/nr-reporting-summary-flat.pdf

# Life sciences study design

All studies must disclose on these points even when the disclosure is negative.

| | |
|---|---|
| Sample size | No statistical test was used to pre-determine the sample size. For RTOC analyses, the sample size was dependent on the number of embryos and embryonic thymi that could be harvested, and the mating was set up based on prior experience to give sufficient number of embryos. For mouse phenotype analysis, the sample size was selected based on our previous experience and mouse availability. |
| Data exclusions | In Fig. 4b-d, 5c, 6a, Extended Data Fig. 3b, 3c, 6c, all RTOCs that were established with GFP-expressing cells but where no GFP+ cells were detectable during analysis, were excluded.<br>In Fig. 5d, all thymic lobes that were injected with IL4-GFP iNKT cells but where no GFP+ cells were detectable during analysis, were excluded.<br>In Fig. 6d-g, all DP thymocyte:iNKT cell interactions lasting for 2 frames or less were excluded from the analyses. |
| Replication | All experiments were repeated at least two times and gave similar results, with the exception of Caspase3 staining (Fig. 3d), WT->IL17A-GFP and IL17A-GFP->IL17A-GFP bone marrow chimera (Fig. 5b), RTOCs with IL4-GFP iNKT cells from different backgrounds (Fig. 6a), RTOCs with Actb-GFP CD4SP cells (Extended Data Fig. 3b) and IL4-GFP eosinophils (Extended Data Fig. 3c), and RTOCs with IL17A-GFP iNKT cells from different backgrounds (Extended Data Fig. 6b) which were analyzed in one experiment. |
| Randomization | After matching for age and sex, mice were randomly assigned to groups. |
| Blinding | Blinding was not feasible as most of the experiments were performed by one and the same person. |

# Reporting for specific materials, systems and methods

We require information from authors about some types of materials, experimental systems and methods used in many studies. Here, indicate whether each material, system or method listed is relevant to your study. If you are not sure if a list item applies to your research, read the appropriate section before selecting a response.

## Materials & experimental systems

| n/a | Involved in the study |
|---|---|
| ☐ | ☒ Antibodies |
| ☐ | ☒ Eukaryotic cell lines |
| ☒ | ☐ Palaeontology and archaeology |
| ☐ | ☒ Animals and other organisms |
| ☒ | ☐ Clinical data |
| ☒ | ☐ Dual use research of concern |

## Methods

| n/a | Involved in the study |
|---|---|
| ☒ | ☐ ChIP-seq |
| ☐ | ☒ Flow cytometry |
| ☒ | ☐ MRI-based neuroimaging |

# Antibodies

**Antibodies used**

Antibody/company/Dilution/Cat
Anti-mouse CD4, SB436 (RM4-8), Invitrogen (eBio)1:200 cat: 62-0042-82
Anti-mouse CD4, VioGreen (GK1.5), Miltenyi 1:50 cat: 130-123-899
Anti-mouse CD4, BV650 (GK1.5), Biolegend 1:200 cat: 100469
Anti-mouse CD4, APC (GK1.5), Biolegend 1:200 cat:100412
Anti-mouse CD4, APCCY7 (GK1.5), Biolegend 1:200 cat:100413
Anti-mouse CD8α PECY7 (53-6.7), Biolegend 1:200 cat: 100722
Anti-mouse CD8α BV605 (53-6.7), Biolegend 1:200 cat: 100744
Anti-mouse CD8β APC (YT5156.7.7), Biolegend 1:200 cat: 126614
Anti-mouse CD8β APCCY7 (YT5156.7.7), Biolegend 1:200 cat: 126620
Anti-mouse CD8β BV711 (YT5156.7.7), Biolegend 1:200 cat: 126633
Anti-mouse PD-1 APC (REA802), Miltenyi 1:50 cat: 130-111-801
Anti-mouse PD-1 PE (29F.1A12), Biolegend 1:200 cat: 135205
Anti-mouse CD24 APC (M1/69), Biolegend 1:200 cat: 101813
Anti-mouse CD24 BV510 (M1/69), Biolegend 1:200 cat: 101831
Anti-mouse CD44 Pacific Blue (IM7), Biolegend 1:400 cat: 103019
Anti-mouse CD44 APCCY7 (IM7), Biolegend 1:400 cat: 103027
Anti-mouse CD19 APC (6D5), Biolegend 1:200 cat: 115512
Anti-mouse CD19 BV786 (1D3), BD 1:200 cat: 563333
Anti-mouse CD11c APC (N418), Biolegend 1:200 cat: 117310
Anti-mouse CD45.1 PerCP cy5.5 (A20), Biolegend 1:200 cat: 110726
Anti-mouse CD45.1 BV421 (A20), Biolegend 1:200  cat: 110732
Anti-mouse CD45.1 BV605 (A20), Biolegend 1:200 cat: 110738
Anti-mouse CD45.1 FITC (A20), Biolegend 1:200  cat: 110705
Anti-mouse CD45.2 PECY7 (104), Biolegend 1:200 cat: 109830
Anti-mouse CD45.2 APCCY7 (104), Biolegend 1:200 cat: 109823
Anti-mouse TCRβ Pacific Blue (H57-597), Biolegend 1:200 cat: 109226
Anti-mouse TCRβ Percpcy5.5 (H57-597), Biolegend 1:200 cat: 109228
Anti-mouse TCRγδ PE-Cy7 (GL3), Biolegend 1:200 cat: 118124
Anti-mouse H-2Kb APC (AF 6-885), BioLegend 1:200 cat: 116517
Anti-mouse H-2Kd PE (SF1-1.1), BioLegend 1:200 cat: 116607
Anti-mouse cKit BV421 (2B8), Biolegend 1:200 cat:105828
Anti-mouse Ter119 APC (TER-119), Biolegend 1:200 cat: 116212
Anti-mouse NK1.1 APC (S17016D), Biolegend 1:200 cat: 156506
Anti-mouse SiglecF PE (E50-2440), BD Bioscience 1:200 cat: E50-2440
Anti-mouse MHCII BV421 (M5/114.15), Biolegend 1:200 cat: 107632
Anti-mouse MHCII PE-Vio770 (REA813), Miltenyi 1:200 130-112-389
Anti-mouse Ly51 PEcy7 (6C3), Biolegend 1:200 cat: 108313
Anti-mouse EPCAM APC (G8.8), Biolegend 1:200 cat: 118214
Anti-mouse TCRVα2 Pacific Blue (B20.1), Biolegend 1:200 cat: 127815
Anti-mouse TCRV β5.1, 5.2 APC (MR9-4), Biolegend 1:200 cat: 139505
Anti-mouse TCRVβ4 (REA729), Miltenyi 1:50 cat: 130-111-101
Anti-mouse Thy1.2 AF700 (30-H12), Biolegend 1:200 cat: 105319
Anti-mouse Thy1.1 Pacific Blue (OX7), Biolegend 1:400 cat: 202521
Anti-mouse Thy1.1 PE (OX7), Biolegend 1:200 cat: 202523
Anti-mouse Thy1.1 VioGreen (OX7), Miltenyi 1:50 cat: 130-112-879
Anti-mouse Ly49 FITC (14B11), Biolegend 1:200 cat:108205
Anti-mouse CD122 BV421 (TM-β1), BD 1:50 cat: 562960
Anti-mouse CCR7 PE (4B12), Miltenyi 1:100 cat: 130-126-035
Anti-mouse CD25 PEVio770 (REA568), Miltenyi 1:50 130-123-893
Anti-mouse CD80 PE (16-10A1), BD Biosciences 1:400 cat:553769
Anti-mouse CD86 PE (GL1), BD Biosciences 1:400 cat:553692
Anti-mouse CD80 PECy7 (16-10A1), Biolegend 1:200 cat:104733
Anti-mouse CD86 PECy7 (PO3.3), Miltenyi 1:50 cat: 130-116-518
PE Armenian Hamster IgG Isotype Ctrl (HTK888), Biolegend 1:400 cat:400907
PE/Cyanine7 Rat IgG2b, κ Isotype Ctrl (RTK4530), Biolegend 1:200 cat:400617
Anti-mouse IFNγ (REA638), Miltenyi 1:100 cat: 130-117-668
Anti-mouse IL4 FITC (11B11), Biolegend 1:200 cat: 504109
Anti-mouse IL17A PE (TC11-18H10) ,Miltenyi, 1:50 cat: 130-103-015
Anti-mouse caspase3 unconjugated (5A1E), Cell Signaling Technology 1:6400 cat: 9664S
Anti-rabbit IgG (H+L), F(ab')2 fragments PE,Cell Signaling Technology 1:1000 cat: 79408S
Streptavidin-APC ,BioLegend 1:500 cat: 405207
Biotinylated UEA1, Vector Laboratories 1:200. cat: B-1065

**Validation**

All antibodies are commercially available and validated by the vendors. Validation data are available on the vendors' websites.

# Eukaryotic cell lines

Policy information about cell lines and Sex and Gender in Research

| | |
|---|---|
| Cell line source(s) | 16.2c11 reporter cell line is a kind gift from Jan Kisielow. HEK293T cells were obtained from Research Institute of Molecular Pathology (IMP), Vienna. |
| Authentication | Cell lines were not authenticated. |
| Mycoplasma contamination | Cell lines were tested negative for mycoplasma. |
| Commonly misidentified lines (See ICLAC register) | No commonly misidentified lines were used. |

# Animals and other research organisms

Policy information about studies involving animals; ARRIVE guidelines recommended for reporting animal research, and Sex and Gender in Research

| | |
|---|---|
| Laboratory animals | IL4-GFP, IL17A-GFP and Ifng–/– mice were purchased from the Jackson Laboratory. Rag2–/– mice were purchased from Janvier Labs. PLZFlu/lu, OT-I, Actb-GFP, Jedi-TCRαβ and Tcrd–/–Cd1d–/–Mr1–/– mice were described previously. WT C57BL/6J and BALB/c mice were obtained from Janvier Labs or bred in house. Jedi-TCRβ only mice were generated from Jedi-TCRαβ mice by crossing with WT BALB/c mice and were further backcrossed to BALB/c background for more than 10 generations. WT C57BL/6J x BALB/c F1 mice (referred to as CB6F1), IL4-GFP CB6F1, IL17A-GFP CB6F1, Actb-GFP CB6F1 and Jedi-TCRβ CB6F1 mice were generated by crossing mice on C57BL/6J and BALB/c backgrounds. For MHC mismatch experiments, IL4-GFP CB6F1 mice were crossed with WT C57BL/6J mice to generate IL4-GFP H2b/d and IL4-GFP H2b/b mice. Tcrd–/–Cd1d–/–Mr1–/– mice were bred and maintained at the Würzburg Institute for Systems Immunology. All other mice were bred and maintained at the Comparative Medicine Biomedicum facility of Karolinska Institutet (Stockholm, Sweden). All mice were housed in SPF condition at 50% humidity, 22°C, with 6pm to 6am nocturnal dark light circle. |
| Wild animals | No wild animals were used in this study. |
| Reporting on sex | Both male and female mice were analyzed. |
| Field-collected samples | This study did not include any field-collected samples. |
| Ethics oversight | All mouse experiments were carried out according to valid project licenses, which were approved and regularly controlled by the Swedish Veterinary Authority. |

Note that full information on the approval of the study protocol must also be provided in the manuscript.

# Flow Cytometry

## Plots

Confirm that:

☒ The axis labels state the marker and fluorochrome used (e.g. CD4-FITC).

☒ The axis scales are clearly visible. Include numbers along axes only for bottom left plot of group (a 'group' is an analysis of identical markers).

☒ All plots are contour plots with outliers or pseudocolor plots.

☒ A numerical value for number of cells or percentage (with statistics) is provided.

## Methodology

| | |
|---|---|
| Sample preparation | Mouse organs were freshly harvested, and single cell suspensions were obtained by mincing through 70 μm cell strainers. Isolation of iIEL was performed as described previously. |
| Instrument | LSR Fortessa Flow Cytometer, FACS Aria III (BD Biosciences), or Cytek Aurora (Cytek Biosciences) |
| Software | Data collection: FACS Diva (v.8)<br>Data analysis: FlowJo (v.10) |
| Cell population abundance | The abundance of the post-sort fraction was determined by reanalysis. |
| Gating strategy | Live gate (FSC/SSC) included all live cells: from small lymphocytes to larger monocytes and SSC-hi granulocytes. For the most part, erythrocytes were gated out. Detailed gating strategies are described in the figures and figure legends. |

☒ Tick this box to confirm that a figure exemplifying the gating strategy is provided in the Supplementary Information.

