## [Peer Review File · Nature Immunology]

Peer Review Information

Journal: Nature Immunology

Manuscript Title: Direct presentation of inflammation-associated self-antigens by thymic innate-like T cells induces elimination of autoreactive CD8 thymocytes

Corresponding author name(s): Dr Taras Kreslavsky

Reviewer Comments & Decisions:

Decision Letter, initial version:
--

18th Jul 2023

Dear Dr Kreslavsky,

Thank you for providing a response to reviewers concerns regarding your article "Direct presentation of inflammation-associated self antigens by thymic innate-like T cells induces elimination of autoreactive CD8 thymocytes". We would be interested in considering a revised version that addresses these reviewers concerns as you outline in your response.

We hope you will find the referees' comments useful as you decide how to proceed. If you wish to submit a substantially revised manuscript, please bear in mind that we will be reluctant to approach the referees again in the absence of major revisions.

If you choose to revise your manuscript taking into account all reviewer and editor comments, please highlight all changes in the manuscript text file [OPTIONAL: in Microsoft Word format].

* If you have not done so already please begin to revise your manuscript so that it conforms to our Article format instructions at <http://www.nature.com/ni/authors/index.html>. Refer also to any

guidelines provided in this letter.

The Reporting Summary can be found here:

When submitting the revised version of your manuscript, please pay close attention to our [href="https://www.nature.com/nature-portfolio/editorial-policies/image-integrity">Digital Image Integrity Guidelines](https://www.nature.com/nature-portfolio/editorial-policies/image-integrity). and to the following points below:

[REDACTED]

If you wish to submit a suitably revised manuscript we would hope to receive it within 6 months. If you cannot send it within this time, please let us know. We will be happy to consider your revision so long as nothing similar has been accepted for publication at Nature Immunology or published elsewhere.

Nature Immunology is committed to improving transparency in authorship. As part of our efforts in this direction, we are now requesting that all authors identified as 'corresponding author' on published papers create and link their Open Researcher and Contributor Identifier (ORCID) with their account on the Manuscript Tracking System (MTS), prior to acceptance. ORCID helps the scientific community achieve unambiguous attribution of all scholarly contributions. You can create and link your ORCID from the home page of the MTS by clicking on 'Modify my Springer Nature account'. For more information please visit www.springernature.com/orcid.

Thank you for the opportunity to review your work.

Sincerely,

Stephanie Houston
Editor
Nature Immunology

Reviewers' Comments:

Reviewer #1:

Remarks to the Author:

The MS by You et al. addresses the modalities of tolerance induction to 'inflammation-associated' self-antigens whose expression is temporally and spatially linked to infection by pathogens. Such self-antigens are hence expected to be presented alongside 'foreign' in a highly immunogenic context, so the question of whether and how specific tolerance is induced is highly original and of great interest. The authors employ the neo-antigen GFP under control of two different 'inflammation associated' gene loci (IL-4 and IL-17A) as model systems for their study. In both constellations, GFP is expressed by a variety of rare 'innate-like' T cells as well as mTECs, which importantly mimics the physiological intrathymic expression pattern of the respective cytokines. Using TCR transgenic mice as well as specific MHC tetramers in conjunction with bone-marrow chimeras and RTOC experiments, it is shown that GFP expression in either hematopoietic cells or mTECs alone is sufficient for deletion of GFP specific CD8 T cells in a redundant manner. The authors go on to show that deletion as a consequence of GFP-expression in innate-like T cells is independent of cross-presentation, i.e. results from direct presentation by these cells. Finally, it is shown that when central tolerance to such antigens is experimentally abolished, i.e. the antigens are only expressed in the periphery but not in the thymus, the antigen-expressing subset of T cells appears to be eliminated by cytotoxic T cells. The experiments are all very well performed and the MS is well written. The enthusiasm is somewhat dampened by concerns that all conclusions of the MS are based on studies with a model antigen.

Major point

- 1) A major conceptual concern is in my opinion that it remains somewhat vague for which true self-antigens the neo-antigen GFP may be representative. In that context, the statement on page 20 that ..'mice deficient for PLZF ...did not reveal a breach in tolerance to endogenous inflammation-associated antigens..' is highly relevant. Which antigens were studied? How? The respective findings as such (which are unfortunately not further specified) may point towards redundancy between innate-like cells and mTECs, and the authors discuss this. Nonetheless, similar to what is done in the MS for IL-4 GFP or IL17A-GFP, it may still be possible to perform PLZFko or WT into endogenous 'antigen X' knockout mice, so that the contribution of mTECs is abolished. Has this been done for the respective endogenous antigens? Without knowing what the authors actually looked at in PLZFko mice, this point is difficult to specify any further.
- 2) Along these lines, a second caveat is that the neo-antigen GFP is expressed cytoplasmatically and hence does not reflect the subcellular distribution of IL-4 and IL-17A. This has implications for whether or not cross-presentation may occur or be necessary. The authors discuss this issue, but phrase the respective passages in the text as if it was 'unexpected' that endogenous presentation of cytoplasmic antigen on MHC class I is involved (At the same time, the authors do a good job in citing the relevant literature on negative selection by 'T cell antigens!')

Reviewer #2:

Remarks to the Author:

While the presence of innate-like T cells in the thymus has been acknowledged for a long time, there has been no clear explanation on their roles and requirements. Here, the authors propose a highly intriguing idea where thymic innate-like cells play a previously unappreciated role in establishing central tolerance to proinflammatory effector molecules, thus preventing autoreactivity to inflammatory effector T cells. Using GFP as a model antigen that is engineered to be expressed by innate-like cells, the authors document in IL-4-GFP reporter mice that GFP-specific CD8 T cells undergo clonal deletion during their development in the thymus. Consequently, GFP-reactive CD8 T cells are absent in peripheral lymphoid tissues, and anti-GFP immunity fails to be established. Remarkably, such negative selection in the thymus could be achieved without the contribution of thymic stromal cells but by GFP-expressing innate-like cells only, which was demonstrated in a series of BM chimera and RTOC experiments as well as by intrathymic injection of IL-4-GFP iNKT cells into GFP-specific TCR expressing thymocytes. Mechanistically, the authors exclude the possibility of antigen cross-presentation by DCs, but they suggest that direct GFP antigen presentation by innate-like T cells to DP thymocytes would be the basis of GFP-specific CD8 T cell deletion. Previously, thymic T-T interactions that are associated with strong agonistic TCR signals were found to drive the generation of PLZF⁺ innate-like T cells, such as iNKT and MAIT cells. But the current report significantly differs from these studies as the authors propose that the T-T interaction of DP cells and innate-like T cells leads to clonal deletion of immature thymocytes. Regarding its physiological significance, it is suggested that thymic innate-like cells trim the T cell repertoire to remove "inflammation-associated" specificities from the CD8 T cell pool to avoid "fratricide" of effector T cells by autoreactive CD8 T cells. This model has wide-ranging ramifications in our understanding how central tolerance is established and what roles thymic-resident innate-like T cells would play in this process.

This is a well-contemplated and nicely executed study that was a joy to read. The entire study, however, hinges on using GFP as an artificial "inflammation-associated" antigen, which brings in a couple of major concerns.

First, the IRES-GFP is a cytosolic protein that – unlike inflammatory IL-4 and IL-17- is not expressed into the secretory pathway. In this regard, it is difficult to imagine how inflammatory cytokines (or secreted effector molecules, such as granzyme B) could be loaded onto MHC-I for clonal deletion of effector molecule-specific CD8 T cells.

Second, the GFP-specific Jedi transgenic mouse has the same problem as the classical MHC-I-specific H-Y TCR transgene in that the TCR is ectopically expressed. Specifically, Fig. 3C (middle) shows a dramatic loss of immature DP thymocytes of Jedi mice in the presence of GFP, similarly to H-Y TCR male mice with premature TCR expression (Baldwin TA, 2005, JEM; Kreslavsky T., 2013, JEM).

Whether the loss of DP cells is a developmental arrest at DN stage, lineage redirection or clonal deletion can be discussed. But further evidence (more than PD1 expression) is necessary to attribute the loss of anti-GFP CD8 T cells in the Jedi + IL-4-GFP mice to negative selection. Also, assessing the developmental stage of Tet⁺ DN cells (Fig.3 C, middle row) by CCR7 staining or CD24 staining could be helpful to understand this issue.

The most puzzling issue, however, is to explain how the T-T interaction of DP thymocytes with GFP-expressing innate like cells would lead to negative selection? If this is indeed negative selection of GFP-specific thymocytes, does it require B7/CD28 co-stimulation and do innate-like cells and/or eosinophil express B7? Strong and/or persistent TCR signaling in immature thymocytes per se is

unlikely to induce negative selection. Rather, such strong/persistent TCR signals would lead to ThPOK expression and CD4 lineage commitment or lineage diversion. Mechanistic insights into how negative selection is induced by GFP-expressing innate-like cells (as well as by conventional GFP-expression CD4 T cells, i.e., CDSP of Actinb-GFP mice) should be provided.

Finally, based on the authors' conclusion, it would be reasonable to expect a breach in central tolerance to proinflammatory effector molecules - if innate-like T cells are absent in the thymus. This would be the case of Ja18-KO and CD1d-KO mice that lack iNKT cells or for PLZF-KO mice, which not only lack iNKT but also MAIT and innate-like gd T cells. For the latter case, the authors mention (lines 434-438) that they analyzed these mice but found no effects regarding inflammation-associated antigens. Because the results are not in agreement with the authors' postulation (line 438), these data should be explained and discussed in further detail.

Some minor points:

1. The RTOC system is a powerful tool to assess the stage-specific requirements of T cell development. However, RTOC necessarily also destroys the thymic architecture. Thus, RTOC artificially brings DP thymocytes and mature iNKT cells into close vicinity which might normally might not happen in vivo because the terminal maturation of iNKT cells occurs in the medulla (White AL., 2014, J Immunol; Lucas B., 2020, Nat. Comms) while DP thymocytes reside in the cortex.
2. Graph showing Tet+ CD8 T cell numbers is missing in Fig. 2a. This would be important to understand the actual magnitude of Tet+ CD8 T cell expansion upon peptide stimulation.
3. In line 458, the authors state that only a fraction of in vitro Th17-skewed IL17-IRES-GFP cells express GFP. This worries me because it questions whether GFP-positive cells indeed correspond to Th17 cells? Can the authors correlate GFP expression with IL-17 production before injection (Fig. 8d)
4. In Figure 8b, 8c and 8d, is the mixed BM set up with Jedi-TCRb – not Jedi-TCRab- T cells? If this the case, this should be clearly indicated in the figure to avoid confusion with experiments using Jedi-TCRab T cells.
5. The results as shown in Figure 8 are difficult to appreciate without seeing the data of Tet+ cells and GFP+ cells in the thymus. These data should be added and discussed.
6. This is an excellently written manuscript, but here are a few places where the conclusions are overstated. For example, in lines 243-246, the conclusion is overinterpreting the results. While the IL-4-GFP population is indeed enriched for innate-like T cells, at this point in the manuscript, these results do not preclude an involvement of professional APCs in the elimination of GFP-specific CD8 T cells, such as by antigen cross-presentation. I suggest dampening down the statement.

Author Rebuttal to Initial comments

See inserted PDF

We would like to thank both reviewers for their positive assessment of our work, constructive criticism and suggestions. We believe that the experiments performed in response to these suggestions substantially enhanced our story. We agree with both reviewers that utilization of intracellular GFP as the only antigen was an important limitation of the original manuscript. In the revised version, we aimed to address this aspect by three sets of experiments. First, to assess the effects of antigen localization, we compared intracellular vs secreted versions of the model antigen in RTOC and intrathymic transfer experiments. Second, as an important limitation of the Jedi/GFP system is its fixed affinity, we assessed the effects of affinity on the efficiency of negative selection induced by innate-like T cells by taking advantage of altered peptide ligands for the OT-I TCR. Finally, and most importantly, we sought to find evidence for or against the involvement of effector lymphocytes in the induction of tolerance to an endogenous inflammation-associated self antigen. Here, we are grateful to Reviewer 1 for the suggestion to exploit knockout mice for a T cell effector molecule. This idea, with some modifications, allowed us to move beyond model antigens and to demonstrate that hematopoietic cells are sufficient to induce tolerance to IFN γ . Taken together with T/NK/ILC-restricted expression of IFN γ , these results provide evidence for the role of these effector subsets in the induction of tolerance to an endogenous component of a T cell effector program.

Reviewer #1

(Remarks to the Author)

The MS by You et al. addresses the modalities of tolerance induction to ‘inflammation-associated’ self-antigens whose expression is temporally and spatially linked to infection by pathogens. Such self-antigens are hence expected to be presented alongside ‘foreign’ in a highly immunogenic context, so the question of whether and how specific tolerance is induced is highly original and of great interest.

We thank the reviewer for a positive assessment of the research question of our study.

The authors employ the neo-antigen GFP under control of two different ‘inflammation associated’ gene loci (IL-4 and IL-17A) as model systems for their study. In both constellations, GFP is expressed by a variety of rare ‘innate-like’ T cells as well as mTECs, which importantly mimics the physiological intrathymic expression pattern of the respective cytokines. Using TCR transgenic mice as well as specific MHC tetramers in conjunction with bone-marrow chimeras and RTOC experiments, it is shown that GFP expression in either hematopoietic cells or mTECs alone is sufficient for deletion of GFP specific CD8 T cells in a redundant manner. The authors go on to show that deletion as a consequence of GFP-expression in innate-like T cells is independent of cross-presentation, i.e. results from direct presentation by these cells. Finally, it is shown that when central tolerance to such antigens is experimentally abolished, i.e. the antigens are only expressed in the periphery but not in the thymus, the antigen-expressing subset of T cells appears to be eliminated by cytotoxic T cells.

The experiments are all very well performed and the MS is well written. The enthusiasm is somewhat dampened by concerns that all conclusions of the MS are based on studies with a model antigen.

We thank the reviewer for a positive assessment of the experiments and the manuscript and hope that we were able to address the concern about utilization of a model antigen in the revised version (see below).

Major point

1) A major conceptual concern is in my opinion that it remains somewhat vague for which true self-antigens the neo-antigen GFP may be representative. In that context, the statement on page 20 that ‘mice deficient for PLZF ...did not reveal a breach in tolerance to endogenous inflammation-associated antigens..’ is highly relevant. Which antigens were studied? How? The respective findings as such (which are unfortunately not further specified) may point towards redundancy between innate-like cells and mTECs, and the authors discuss this. Nonetheless, similar to what is done in the MS for IL-4 GFP or IL17A-GFP, it may still be possible to perform PLZFko or WT into endogenous ‘antigen X’ knockout mice, so that the contribution of mTECs is abolished. Has this been done for the respective endogenous antigens? Without knowing what the authors actually looked at in PLZFko mice, this point is difficult to specify any further.

We are grateful to the reviewer for this question – especially for the idea of an experiment with ‘antigen X’ knockout mice.

Negative results with PLZF-deficient and $Tcrd^{-/-}Cd1d^{-/-}Mr1^{-/-}$ mice.

In our initial attempts to identify a breach of tolerance in mice that are expected to have reduced numbers of innate-like T cells, we performed a number of experiments with $PLZF^{lu/lu}$ mice and $PLZF^{lu/lu} \rightarrow Rag2^{-/-}$ BM chimeras. This included peptide immunizations with epitopes from antigens predominantly expressed by innate-like T cells, repetitive transfer of *in vitro* polarized cells as well as antigen arrays to detect possible autoantibody responses. Negative results of some of these experiments are included at the end of this point-by-point response.

However, when we finally performed a simple stimulation of PLZF-deficient thymocytes with PMA and ionomycin, we realized that despite the near-absence of iNKT cells, thymi of $PLZF^{lu/lu}$ mice contained IFN γ - and IL17A-secreting cells at frequencies indistinguishable from that of WT mice. From the effector subsets, only the IL4-producing population was decreased but still clearly present in PLZF-deficient thymi (new Extended Data Fig. 9b). In line with the presence of IL4-producing cells (as well as with the redundancy between thymic innate-like T cells, eosinophils and TECs in the induction of tolerance to IL4-GFP), repetitive “immunizations” of $PLZF^{lu/lu} \rightarrow Rag2^{-/-}$ and WT $\rightarrow Rag2^{-/-}$ BM chimeras with *in vitro* generated T_H2 cells followed by an injection of CTV-labeled T_H2 cells for an “*in vivo* killing assay” failed to reveal a breach in tolerance to this effector subset (see new Extended Data Fig. 9c).

In a further attempt to find an innate-like T cell-deficient model, we now also collaborated with Wolfgang Kastenmüller’s lab that had generated CD1d/MR1/TCR δ triple knockout mice. In line with observations published by the Kastenmüller lab for peripheral lymphoid organs (Ataide et al, 2022, PMID: 36002023 Fig. S4E), no significant decrease was observed in either IFN γ -, IL4-, or IL17A-secreting populations (new Extended Data Fig. 9a). We

conclude that none of the existing genetic models allows to eliminate thymic innate-like lymphocytes.

'Antigen X' knockout experiments.

Although neither PLZF-deficient nor *Tcrd*^{-/-}*Cd1d*^{-/-}*Mr1*^{-/-} mice provided a suitable system for a complete or near-complete ablation of innate-like T cells, we realized that the chimera experiments with 'antigen X' knockout mice suggested by the reviewer can still be highly informative, provided that 'antigen X' expression in hematopoietic cells is restricted to effector lymphocytes.

We decided to utilize IFN γ -deficient mice for these experiments. Similar to many T cell effector program antigens, *Ifng* is expressed by innate and innate-like lymphocytes as well as by a small subset of TECs (Fig. 1a, b). Production of IFN γ upon PMA/Ionomycin stimulation of thymic cells was also restricted to Thy1- and/or NK1.1-positive cells and the IFN γ -producing cells did not include professional APCs as evidenced by the analysis of MHC II expression (new Extended Data Fig. 10f). We therefore reasoned that comparison of WT \rightarrow KO and KO \rightarrow KO chimeras can allow to assess the contribution of BM-derived cells (in case of IFN γ – predominantly if not exclusively T/NK/ILC cells) to the induction of tolerance to this antigen. We computationally predicted several H2-K^b and H2-D^b epitopes in IFN γ , validated them in MHC stabilization assays and tested their ability to induce responses in WT and *Ifng*^{-/-} mice. We managed to identify an H2-K^b epitope in IFN γ (IFN γ ₆₉₋₇₇) that was highly immunogenic in *Ifng*^{-/-} but not in WT mice, suggesting that a) this epitope is generated and presented *in vivo* and b) that its presentation in WT mice results in the induction of CD8 T cell tolerance.

We then established a full set of "reciprocal" BM chimeras (WT \rightarrow WT, WT \rightarrow KO, KO \rightarrow WT, KO \rightarrow KO) and immunized them with an irrelevant foreign peptide (OVA₂₅₇₋₂₆₄ SIINFEKL) and with the IFN γ ₆₉₋₇₇ epitope. We used intracellular staining for TNF after a brief peptide restimulation as a readout for antigen-specific CD8 T cell responses. All groups mounted a comparable response to SIINFEKL (new Extended Data Fig. 10g). However, IFN γ ₆₉₋₇₇ was highly immunogenic only in KO \rightarrow KO chimeras (new Fig. 8e), while WT \rightarrow KO, KO \rightarrow WT and WT \rightarrow WT groups all were equally tolerant to this self antigen. These results indicate that both radioresistant and radiosensitive cells can mediate tolerance to this antigen. Taken together with T/NK/ILC-restricted expression of IFN γ , these results also provide strong evidence for the role of these effector subsets in the induction of tolerance to an endogenous inflammation-associated self antigen. Importantly, while by itself this experiment cannot discriminate between central and peripheral tolerance mechanisms, the abundance of IFN γ -expressing effector subsets in the thymus and our results obtained with model antigens suggest that central tolerance is likely to contribute to the neutralization of IFN γ ₆₉₋₇₇-reactive CD8 T cells.

2) Along these lines, a second caveat is that the neo-antigen GFP is expressed cytoplasmatically and hence does not reflect the subcellular distribution of IL-4 and IL-17A. This has implications for whether or not cross-presentation may occur or be necessary. The authors discuss this issue, but phrase the respective passages in the text as if it was 'unexpected' that endogenous presentation of cytoplasmic antigen on MHC class I is

involved (At the same time, the authors do a good job in citing the relevant literature on negative selection by 'T cell antigens'!)

We fully agree with the reviewer that the utilization of IL4-GFP and IL17A-GFP as model inflammation self antigens has its limitations. We could identify two main constraints that precluded the generalization of our initial observations. First, as pointed out by the reviewer, while GFP 'mimics the physiological intrathymic expression pattern of the respective cytokines', it is not secreted and from that perspective models other intracellular components of effector T cell programs (e.g. transcription factors or signaling molecules) rather than the corresponding cytokines. Second, as the GFP₂₀₀₋₂₀₈ epitope is the same in both systems, the utilization of Jedi TCR provides a setting with a fixed (and likely relatively high) affinity.

Intracellular vs secreted antigen.

To bypass the first limitation, we now have generated retroviral constructs with secreted and intracellular versions of GFP, confirmed GFP secretion (new Extended Data Fig. 6c), transduced thymic iNKT cells with these constructs and tested their ability to induce elimination of Jedi DP cells. Both antigen versions induced comparable elimination of GFP-reactive thymocytes in the RTOC system (new Fig. 4c). As in the case of the secreted GFP version we expected to find contribution of cross-presentation, we also performed MHC mismatched RTOC experiments. Contrary to our expectations, these experiments demonstrated that for both intracellular and secreted GFP, the elimination of autoreactive thymocytes occurred exclusively through direct antigen presentation (new Extended Data Fig. 8c). Moreover, that was not a peculiarity of the RTOC system, as upon intrathymic transfer of GFP-secreting iNKT cells strong PD1 upregulation and CD8 downregulation on Jedi cells was observed only when iNKT cells were capable of direct GFP₂₀₀₋₂₀₈ presentation (new Extended Data Fig. 8e). It is important to note that this does not exclude a possibility of cross-presentation of other secreted antigens – for example in a setting when cross-presenting APC would express a receptor for a secreted antigen.

In addition, as discussed above, in the revised version of the manuscript we also provide evidence for the role of effector lymphocytes in the induction of tolerance to an endogenous secreted inflammation-associated self antigen – IFN γ (see response to point 1).

Effects of affinity.

We next sought to test if the negative selection by antigen expressed by innate-like T cells may be restricted to high affinity epitopes. To this end, we retrovirally transduced thymic iNKT cells with constructs encoding cytoplasmic OVA with either the WT version of the SIINFEKL epitope (N4), or its mutants (Q4, T4 and V4). These epitopes span a wide range of affinities to the OT-I TCR, as evidenced by about 20- (Q4), 70- (T4) and 700 (V4) -fold decrease in the reactivity of OT-I CD8 T cells (Zehn and Bevan, 2009, PMID: 19182777). Strikingly, the expression of N4, Q4 and T4 epitopes by iNKT cells resulted in a near-complete elimination of OT-I DP thymocytes in RTOCs (new Fig. 4d). Even expression of the weakest V4 epitope induced a loss of ~50% of OT-I cells (new Fig. 4d).

Taken together, these new results demonstrate that the highly efficient induction of tolerance by innate-like T cell-expressed antigens holds true for both secreted and intracellular antigens and operates across a wide range of affinities.

Reviewer #2

(Remarks to the Author)

While the presence of innate-like T cells in the thymus has been acknowledged for a long time, there has been no clear explanation on their roles and requirements. Here, the authors propose a highly intriguing idea where thymic innate-like cells play a previously unappreciated role in establishing central tolerance to proinflammatory effector molecules, thus preventing autoreactivity to inflammatory effector T cells. Using GFP as a model antigen that is engineered to be expressed by innate-like cells, the authors document in IL-4-GFP reporter mice that GFP-specific CD8 T cells undergo clonal deletion during their development in the thymus. Consequently, GFP-reactive CD8 T cells are absent in peripheral lymphoid tissues, and anti-GFP immunity fails to be established. Remarkably, such negative selection in the thymus could be achieved without the contribution of thymic stromal cells but by GFP-expressing innate-like cells only, which was demonstrated in a series of BM chimera and RTOC experiments as well as by intrathymic injection of IL-4-GFP iNKT cells into GFP-specific TCR expressing thymocytes. Mechanistically, the authors exclude the possibility of antigen cross-presentation by DCs, but they suggest that direct GFP antigen presentation by innate-like T cells to DP thymocytes would be the basis of GFP-specific CD8 T cell deletion. Previously, thymic T-T interactions that are associated with strong agonistic TCR signals were found to drive the generation of PLZF⁺ innate-like T cells, such as iNKT and MAIT cells. But the current report significantly differs from these studies as the authors propose that the T-T interaction of DP cells and innate-like T cells leads to clonal deletion of immature thymocytes. Regarding its physiological significance, it is suggested that thymic innate-like cells trim the T cell repertoire to remove “inflammation-associated” specificities from the CD8 T cell pool to avoid “fratricide” of effector T cells by autoreactive CD8 T cells. This model has wide-ranging ramifications in our understanding how central tolerance is established and what roles thymic-resident innate-like T cells would play in this process.

This is a well-contemplated and nicely executed study that was a joy to read. The entire study, however, hinges on using GFP as an artificial “inflammation-associated” antigen, which brings in a couple of major concerns.

We thank the reviewer for a positive assessment of the research question, the design of the experiments and the manuscript text. In the revised version of the manuscript, we provide evidence that the involvement of innate-like T cells in the elimination of autoreactive CD8 T cells is not restricted to a single intracellular high affinity model antigen (see below).

First, the IRES-GFP is a cytosolic protein that – unlike inflammatory IL-4 and IL-17- is not expressed into the secretory pathway. In this regard, it is difficult to imagine how inflammatory cytokines (or secreted effector molecules, such as granzyme B) could be loaded onto MHC-I for clonal deletion of effector molecule-specific CD8 T cells.

We fully agree with the reviewer that GFP in IL4-GFP and IL17A-GFP mice remains intracellular and from that perspective models intracellular components of effector

programs rather than the corresponding cytokines. Nevertheless, it is important to note that epitopes from secreted proteins can be subjected to MHC I presentation (as an example – pancreatic beta cells secreting insulin can be attacked by insulin-specific CD8 T cells). Mechanistically, this can be for example explained by the export of unfolded proteins from the ER back to the cytosol where they are subjected to proteasomal degradation (see PMID: 21962745 for review).

To compare the induction of tolerance by intracellular and secreted versions of the antigen, we now performed experiments with iNKT cells retrovirally transduced with constructs encoding intracellular or secreted forms of GFP. We did not find differences in these two GFP versions (new Fig. 4c) even in terms of their dependence on direct antigen presentation (new Extended Data Fig. 8c,e; see response to Reviewer 1 point 2 above for details).

Moreover, in the revised version of the manuscript we also provide evidence for the role of effector lymphocyte subsets in the induction of tolerance to an endogenous secreted inflammation-associated self antigen – IFN γ (see response to Reviewer 1 point 1 above).

Second, the GFP-specific Jedi transgenic mouse has the same problem as the classical MHC-I-specific H-Y TCR transgene in that the TCR is ectopically expressed. Specifically, Fig. 3C (middle) shows a dramatic loss of immature DP thymocytes of Jedi mice in the presence of GFP, similarly to H-Y TCR male mice with premature TCR expression (Baldwin TA, 2005, JEM; Kreslavsky T., 2013, JEM). Whether the loss of DP cells is a developmental arrest at DN stage, lineage redirection or clonal deletion can be discussed. But further evidence (more than PD1 expression) is necessary to attribute the loss of anti-GFP CD8 T cells in the Jedi + IL-4-GFP mice to negative selection. Also, assessing the developmental stage of Tet⁺ DN cells (Fig.3 C, middle row) by CCR7 staining or CD24 staining could be helpful to understand this issue.

Both negative selection and lineage diversion contribute to the induction of tolerance in a system with premature TCR expression.

Indeed, as noted by the reviewer the TCR α chain of the Jedi TCR exhibits premature expression at DN stages of T cell development resulting in an early expression of a complete $\alpha\beta$ TCR (Fig. 3c). Moreover, exactly as predicted by the reviewer, these DN4-like Jedi TCR⁺ thymocytes upregulated CCR7 and downregulated CD24 (new Extended Data Fig. 3e), indicating that these cells are not equivalent to the DN4 stage of T cell development and suggesting that they may represent cells undergoing lineage diversion due to premature TCR expression. In line with that notion, some CD8 α ⁻CD8 β ⁻ and CD8 α ⁺CD8 β ⁻ Jedi TCR⁺ cells were found among intestinal IELs, and this population was increased in numbers in the presence of antigen (in [IL4-GFP+Jedi]→WT BM chimeras; new Extended Data Fig. 4). These results indicate that in BM chimeras with a Jedi-TCR $\alpha\beta$ -expressing cells lineage diversion indeed makes some contribution to inactivation of autoreactive thymocytes.

As during the revision of this manuscript Al Singer's group published an elegant study (PMID: 37917689) identifying premature eviction of DP^{dull} cells from the thymus as a novel mechanism of CD8 T cell tolerance, we also investigated if Jedi cells acquire phenotypes associated with this process. While in [IL4-GFP+Jedi]→WT chimeras many Jedi cells did acquire the CD8^{int} phenotype (Fig. 3d and new Extended Data Fig. 3b) reported for evicted

cells, we could not detect peripheral DP^{dull} cells (Fig. 3d) or the acquisition of a CD122⁺Ly49⁺ phenotype by autoreactive CD8 T cells (new Extended Data Fig. 3c). It remains to be further investigated if clonal eviction of Jedi cells could result in phenotypic changes different from that described by Badr et al for autoreactive CD8 T cells in the HY system.

Importantly, despite the premature TCR expression by Jedi-TCR $\alpha\beta$ thymocytes, even in this system negative selection seems to make a prominent contribution to the elimination of autoreactive thymocytes. Indeed, our new results (new Fig. 3e) demonstrate that Jedi DP cells in [IL4-GFP+Jedi] \rightarrow WT mixed BM chimeras exhibit a strong increase in the frequency of cleaved caspase 3-positive cells – a “gold standard” for the detection of cells undergoing negative selection. Moreover, a very prominent reduction in peripheral Jedi TCR-expressing cells in these chimeras (Fig. 3d) also suggests that physical elimination is an important mechanism of inactivation of Jedi-TCR $\alpha\beta$ cells in the presence of antigen.

Physical elimination of autoreactive Jedi cells is the main mechanism of tolerance induction in a system with physiological timing of TCR expression.

We next decided to assess what happens in a setting with physiological timing of expression of the TCR chains. To this end, we took advantage of Jedi-TCR β only mice in which there is no pre-rearranged TCR α chain and the *Tcra* locus undergoes recombination physiologically, at the DP stage of T cell development. As was described in the original manuscript, a small but clearly distinguishable subset of developing CD8 T cells in these mice generate GFP₂₀₀₋₂₀₈-specific TCRs. We first compared T cell development in full Jedi-TCR $\alpha\beta$ and Jedi-TCR β only mice side by side. In line with the physiological timing of TCR expression, and in contrast to Jedi-TCR $\alpha\beta$ mice, GFP-specific thymocytes in Jedi-TCR β mice were found exclusively in DP and CD8SP compartments (new Extended Data Fig. 3d). We next established chimeras with BM from either full Jedi-TCR $\alpha\beta$ or Jedi-TCR β mice mixed with either WT or IL4-GFP BM. In the presence of antigen, GFP-specific Jedi-TCR β cells were completely eliminated in the thymus and were undetectable either in secondary lymphoid organs or in the iIEL compartment of [IL4-GFP+Jedi-TCR β] \rightarrow WT chimeras (new Extended Data Fig. 4). We conclude that while in a setting with premature TCR expression both lineage diversion and negative selection contribute to the inactivation of autoreactive Jedi cells, in case of physiological timing of TCR expression GFP-specific cells undergo complete or near-complete physical elimination in the thymus.

The most puzzling issue, however, is to explain how the T-T interaction of DP thymocytes with GFP-expressing innate like cells would lead to negative selection? If this is indeed negative selection of GFP-specific thymocytes, does it require B7/CD28 co-stimulation and do innate-like cells and/or eosinophil express B7? Strong and/or persistent TCR signaling in immature thymocytes per se is unlikely to induce negative selection. Rather, such strong/persistent TCR signals would lead to ThPOK expression and CD4 lineage commitment or lineage diversion. Mechanistic insights into how negative selection is induced by GFP-expressing innate-like cells (as well as by conventional GFP-expression CD4 T cells, i.e., CDSP of Actinb-GFP mice) should be provided.

We agree with the reviewer that a possible involvement of B7/CD28 co-stimulation in inactivation of autoreactive thymocytes in case of antigen presentation by non-professional APCs is a relevant question. As discussed above, several lines of evidence, including cell

death of autoreactive cells in the thymus and their loss from the periphery (as well as from thymus organoids), indicate that the physical elimination of Jedi cells plays an important role in this process.

As indicated by the reviewer, many studies had reported that co-stimulation through B7/CD28 interaction is an important factor for negative selection. However, despite its clear importance, the requirement of co-stimulation is not absolute. Indeed, in the absence of co-stimulation the overall frequency of cells undergoing negative selection seems to be reduced about two-fold (Breed et al, 2019, PMID: 31010850; Watanabe et al, 2020, PMID: 33293517) and cells with some autoreactive specificities are not at all affected by B7 deficiency (Watanabe et al, 2020, PMID: 33293517).

Following the reviewer's suggestion, we first assessed the expression of CD80 and CD86 by thymic innate-like T cells and eosinophils (new Extended Data Fig. 6a). Eosinophils were CD86^{-/lo} but expressed very high levels of CD80. We also detected substantial levels of CD80 and CD86 on the surface of a large fraction of thymic iNKT cells, despite little if any *Cd80* or *Cd86* mRNA expression was suggested by analysis of the Immgen database. Identical results were obtained with two different antibody clones for CD86 (GL1 and PO3.3) (new Extended Data Fig. 6a and data not shown). It was recently documented that despite the lack of *Cd80/Cd86* expression, developing thymocytes acquire CD80 and CD86 proteins from thymic APCs through trogocytosis (Watanabe et al, 2022, PMID: 36450247) – and in line with that we observed substantial levels of these molecules on the surface of single positive thymocytes. As thymic iNKT cells directly interact with thymic myeloid populations (Wang et al, 2019, PMID: 31611396) it seems likely that they also acquire CD80 and CD86 proteins from these cells. Lower CD80/CD86 expression was also observed on some $\gamma\delta$ T cells (new Extended Data Figure 6a).

We next utilized anti-CD80/CD86 antibodies (at concentrations previously reported to block CD28 signaling) in RTOCs to test if co-stimulation was required for negative selection of Jedi DP thymocytes. In line with a previous report that lack of co-stimulation enhances the generation of SP thymocytes (Vacchio et al, 2005, PMID: 15657954), CD80/CD86 blocking resulted in a significant increase in Jedi CD8SP cells in RTOCs without GFP, confirming that the antibodies interfered with CD80/CD86 function. However, in the presence of GFP-expressing iNKT cells blocking of co-stimulation did not affect the efficiency of negative selection (new Extended Data Fig. 6b). Taken together, these new results indicate that while co-stimulation by CD80/CD86 receptors was not required for the elimination of autoreactive Jedi thymocytes in the RTOC system, thymic innate-like T cells are equipped with these co-stimulatory molecules normally expressed by professional APCs.

Finally, based on the authors' conclusion, it would be reasonable to expect a breach in central tolerance to proinflammatory effector molecules - if innate-like T cells are absent in the thymus. This would be the case of Ja18-KO and CD1d-KO mice that lack iNKT cells or for PLZF-KO mice, which not only lack iNKT but also MAIT and innate-like gd T cells. For the latter case, the authors mention (lines 434-438) that they analyzed these mice but found no effects regarding inflammation-associated antigens. Because the results are not in agreement with the authors' postulation (line 438), these data should be explained and discussed in further detail.

We now added results showing that both PLZF-deficient and even *Tcrd*^{-/-}*Cd1d*^{-/-}*Mr1*^{-/-} mice have surprisingly normal frequencies of effector populations in the thymus (new Extended Data Fig. 9a,b, see also response to point 1 of Reviewer 1). We also included some experiments that we had performed with PLZF-deficient mice (before we realized that they have all major effector T cell programs well-represented in the thymus) at the end of this point-by-point response.

Nevertheless, during the revision of this manuscript we identified a setting which allowed us to provide evidence for the contribution of effector lymphocytes to the formation of tolerance to an endogenous inflammation-associated self antigen. Following a suggestion from Reviewer 1, we identified an MHC I epitope in IFN γ that was highly immunogenic in *Ifn γ* ^{-/-} but not in WT mice. IFN γ exhibited a T/NK/ILC-restricted pattern of expression in hematopoietic cells (Fig. 1 and new Extended Data Fig. 10f). Immunization of WT \rightarrow WT, WT \rightarrow KO, KO \rightarrow WT and KO \rightarrow KO BM chimeras with this epitope demonstrated that only KO \rightarrow KO chimeras showed a strong response to immunization, while all other groups of chimeras were equally tolerant to this epitope. The profound tolerance observed in WT \rightarrow KO chimeras indicated that IFN γ expression solely by hematopoietic cells is sufficient to inactivate autoreactive CD8 T cells. Taken together with the T/NK/ILC-restricted expression of IFN γ , these results suggest that antigen expression by these effector subsets is sufficient for the induction of tolerance to this endogenous inflammation-associated self antigen.

Some minor points:

1. The RTOC system is a powerful tool to assess the stage-specific requirements of T cell development. However, RTOC necessarily also destroys the thymic architecture. Thus, RTOC artificially brings DP thymocytes and mature iNKT cells into close vicinity which might normally might not happen *in vivo* because the terminal maturation of iNKT cells occurs in the medulla (White AL., 2014, J Immunol; Lucas B., 2020, Nat. Comms) while DP thymocytes reside in the cortex.

We agree that RTOCs do not fully recapitulate the thymic architecture – in particular the separation of cortex and medulla. However, nearly all our RTOC results are backed by similar observations *in vivo* (in BM chimeras and in intrathymic transfer experiments). Moreover, while a large fraction of IL4-producing iNKT cells indeed reside in the medulla, about 20% of these cells were reported to reside in the thymic cortex (Wang et al, 2019, PMID: 31611396). Indeed, our own new IF analysis of the distribution of IL4-GFP-expressing SiglecF⁺ (eosinophils) and SiglecF⁻ (innate-like T cells) in the thymus demonstrates that a substantial fraction of both populations is localized in the cortex (new Fig. 3b and new Extended Data Fig. 2d).

2. Graph showing Tet⁺ CD8 T cell numbers is missing in Fig. 2a. This would be important to understand the actual magnitude of Tet⁺ CD8 T cell expansion upon peptide stimulation.

We have now repeated this experiment and added the absolute numbers.

3. In line 458, the authors state that only a fraction of *in vitro* Th17-skewed IL17-IRES-GFP cells express GFP. This worries me because it questions whether GFP-positive cells indeed correspond to Th17 cells? Can the authors correlate GFP expression with IL-17 production before injection (Fig. 8d)

Point-by-point response. Manuscript NI-A36033.

We have now checked the correlation between the actual IL17A production and IL17-GFP reporter expression in the *in vitro* differentiated T_H17 cells (new Extended Data Figure 10e). The GFP reporter and the actual IL17A cytokine were largely co-expressed by the same cells.

4. In Figure 8b, 8c and 8d, is the mixed BM set up with Jedi-TCRb – not Jedi-TCRab- T cells? If this the case, this should be clearly indicated in the figure to avoid confusion with experiments using Jedi-TCRab T cells.

It is Jedi-TCR β only. To avoid confusion, we now refer to full Jedi TCR as “Jedi-TCR $\alpha\beta$ ” throughout the manuscript, to clearly distinguish it from the “Jedi-TCR β ” setting.

5. The results as shown in Figure 8 are difficult to appreciate without seeing the data of Tet+ cells and GFP+ cells in the thymus. These data should be added and discussed.

We now added these results as a new Extended Data Fig. 10b,d. The GFP⁺ cells, that were already very scarce in IL17A-GFP \rightarrow WT chimera thymi, were virtually absent from the thymi of [IL17A-GFP+Jedi-TCR β] \rightarrow WT chimeras. Thymic GFP-specific CD8 T cells, on the other hand, were not just present but also expanded in these chimeras – possibly reflecting the recruitment of peripherally activated cells (although we cannot exclude local activation). Note that this was in stark contrast to the situation in [IL4-GFP+Jedi-TCR β] \rightarrow WT chimeras in which GFP-specific cells were nearly completely eliminated (new Extended Data Fig. 4). These results could be consistent with a scenario in which autoreactive activated Jedi cells eliminate the few residual IL17A-GFP-expressing cells in the thymus of the chimeras which in turn may allow even more autoreactive GFP-specific cells to escape negative selection.

6. This is an excellently written manuscript, but here are a few places where the conclusions are overstated. For example, in lines 243-246, the conclusion is overinterpreting the results. While the IL-4-GFP population is indeed enriched for innate-like T cells, at this point in the manuscript, these results do not preclude an involvement of professional APCs in the elimination of GFP-specific CD8 T cells, such as by antigen cross-presentation. I suggest dampening down the statement.

We agree with the reviewer. Indeed, this statement was made before any evidence for direct antigen presentation was provided in the manuscript. We therefore removed this sentence.

Additional experiments that were performed with PLZF-deficient mice:

As both reviewers asked to provide further details on the experiments performed with PLZF-deficient mice, below we include some of the experiments we performed before we realized that PLZF-deficient mice have all major effector T cell programs well-represented in the thymus.

1. Peptide immunization experiments.

Using the Immgen database, we identified several antigens predominantly expressed in the thymus by innate-like T cells. We predicted MHC I epitopes, immunized $PLZF^{lu/lu} \rightarrow Rag2^{-/-}$ and $WT \rightarrow Rag2^{-/-}$ BM chimeras with these peptides and used IFN γ production after a brief *in vitro* restimulation as a readout for antigen-specific CD8 T cell responses:

These experiments failed to reveal differences between the two groups of chimeras. Moreover, immunization with some of the peptides resulted in relatively strong responses in both groups suggesting that these epitopes may not be naturally generated *in vivo* and therefore are never seen by the immune system under physiological conditions.

2. Repetitive transfers of *in vitro* differentiated effector subsets.

We also subjected $PLZF^{lu/lu} \rightarrow Rag2^{-/-}$ and $WT \rightarrow Rag2^{-/-}$ BM chimeras to repetitive *in vitro* transfers of *in vitro* polarized T_H1, T_H17 (below) and T_H2 (new Extended Data Fig. 9c) effector subsets. We then performed an “*in vivo* killing assay” by transferring the corresponding *in vitro* polarized subset mixed with naïve T cells labeled with different levels of CTV or CTY dyes. The survival of the transferred effector cells was comparable in both groups of chimeras:

3. Antigen arrays.

Finally, we also collaborated with PJ Utz' lab in Stanford to test if *PLZF^{lu/lu}* mice develop autoantibodies against cytokines. A custom bead-based antigen array was used to profile serum samples for antibodies against cytokines (as well as several other antigens), as previously described (Chang et al, 2021, PMID: 34521836). While antibodies against some cytokines were detectable in sera from MRL *Fas^{lpr/lpr}* mice that were used as a positive control, no such response was reproducibly detected in *PLZF*-deficient animals:

Decision Letter, first revision:

9th May 2024

Dear Dr. Kreslavsky,

Thank you for submitting your revised manuscript "Direct presentation of inflammation-associated self antigens by thymic innate-like T cells induces elimination of autoreactive CD8 thymocytes" (NI-A36033A). It has now been seen by the original referees and their comments are below. The reviewers find that the paper has improved in revision, and therefore we'll be happy in principle to publish it in Nature Immunology, pending minor revisions to satisfy the referees' final requests and to comply with our editorial and formatting guidelines.

We will now perform detailed checks on your paper and will send you a checklist detailing our editorial and formatting requirements in about a week. Please do not upload the final materials and make any revisions until you receive this additional information from us.

If you had not uploaded a Word file for the current version of the manuscript, we will need one before beginning the editing process; please email that to immunology@us.nature.com at your earliest convenience.

Thank you again for your interest in Nature Immunology. Please do not hesitate to contact me if you have any questions.

Sincerely,

Stephanie Houston, PhD
Senior Editor
Nature Immunology

Reviewer #1 (Remarks to the Author):

The authors have done a great job in further improving a MS that already in its initial version was highly exciting. Amazing piece of work.

Reviewer #2 (Remarks to the Author):

This is a well-conceived and nicely executed study that puts forward a new perspective on the role of innate-like T cells in the thymus and how they contribute to establishing central tolerance to proinflammatory effector molecules.

In the previous version, I had three major concerns, which were:

1. The exclusive use of GFP as a model antigen for testing a breach in central tolerance. The GFP results were convincing but insufficient to demonstrate the biological significance of the proposed mechanism.

2. How to reconcile the model GFP expression (which is cytosolic) with effector cytokine expression (which is secreted) to explain direct presentation and negative selection in the thymus
3. The need to test and demonstrate the breach in central tolerance to an inflammatory effector molecule in the absence of innate-like T cells in the thymus.

The authors did a great job of addressing these issues in their revision, introducing new mouse models, and conducting new experiments that convincingly support their conclusions.

Specifically, I appreciate the new experiment in Fig. 8d which identifies a new IFN γ epitope and further employs this peptide to demonstrate impaired central tolerance to proinflammatory IFN γ – in the absence of hematopoietic cell (or T/NK/ILC) derived IFN γ proteins. Also, the new Fig. 4c, demonstrating that both intracellular and secreted GFP can mediate negative selection is interesting. Finally, the major efforts to assess autoimmunity in mice lacking innate T cells (i.e., PLZF lu/lu mice and Tcrd $^{-/-}$ Cd1d $^{-/-}$ Mr1 $^{-/-}$ mice) are greatly appreciated. On one hand, it is disappointing that these mice cannot be used to test the lack of thymocyte-derived inflammatory effector molecules, on the other hand, it is interesting to learn that there is a redundancy and a strong feedback to maintain the expression of these effector molecules in the thymus. Overall, the study has greatly improved, and the narrative is much smoother and easier to follow, specifically when switching from “Jedi-TCRab” to the “Jedi-TCRb” model (p. 13, line 284), which was not so in the original manuscript.

I remain highly enthusiastic about this study, but I have a few minor points to suggest:

1. The results in the new Extended Data 9a, 9b are quite intriguing. What is the identity of inflammatory cytokine-producing cells in PLZF-lu/lu mice and innate-like cell-deficient mice? While the in-depth characterization of the cytokine-producing thymocytes in PLZF-KO or Tcrd, CD1d, Mr1-triple KO mice might be beyond the scope of this study, I wonder why the authors presume that they are T lineage cells (p. 26, line 579)? Instead, is there a possibility that they are ILCs? Providing evidence that affirms them as T lineage cells or rephrasing this sentence is recommended.
2. Regarding the cellular identity of CD8ab-low PD1hi cells (Fig. 6b and Fig. 4b,c,d), the authors mention a possible lineage diversion into “innate-like lineage” cells (page 21, line 462). But it is unclear to me what prompted this idea. Do these cells produce cytokines or express PLZF? Can the authors provide data supporting their hypothesis?
3. In Figure 4 b, it would be useful to show the frequency (or FACS plot) of iNKT cells added to the RTOC. While the WT iNKT cell mixture (top) shows that there are no IL4-GFP-producing cells in the culture, it is unclear if there are iNKT cells added to the RTOC. It would be reassuring to see that iNKT cells are present at the same frequency as for the IL4-GFP iNKT cell RTOC.
4. The antigen array experiment (Figure 3, additional experiments in the point-by-point response) is quite interesting. I would suggest including the data in the revised manuscript because the lack of autoantibodies in PLZF-KO mice is unexpected (based on the authors' model), but that can be now explained by the intact cytokine production in PLZF-KO mice.

Author Rebuttal, first revision:

See inserted PDF

We thank both reviewers for their very positive assessment of the revised version of the manuscript.

Reviewer #1 (Remarks to the Author):

The authors have done a great job in further improving a MS that already in its initial version was highly exciting. Amazing piece of work.

Reviewer #2 (Remarks to the Author):

This is a well-conceived and nicely executed study that puts forward a new perspective on the role of innate-like T cells in the thymus and how they contribute to establishing central tolerance to proinflammatory effector molecules.

In the previous version, I had three major concerns, which were:

1. The exclusive use of GFP as a model antigen for testing a breach in central tolerance. The GFP results were convincing but insufficient to demonstrate the biological significance of the proposed mechanism.
2. How to reconcile the model GFP expression (which is cytosolic) with effector cytokine expression (which is secreted) to explain direct presentation and negative selection in the thymus
3. The need to test and demonstrate the breach in central tolerance to an inflammatory effector molecule in the absence of innate-like T cells in the thymus.

The authors did a great job of addressing these issues in their revision, introducing new mouse models, and conducting new experiments that convincingly support their conclusions.

Specifically, I appreciate the new experiment in Fig. 8d which identifies a new IFN γ epitope and further employs this peptide to demonstrate impaired central tolerance to proinflammatory IFN γ – in the absence of hematopoietic cell (or T/NK/ILC) derived IFN γ proteins. Also, the new Fig. 4c, demonstrating that both intracellular and secreted GFP can mediate negative selection is interesting. Finally, the major efforts to assess autoimmunity in mice lacking innate T cells (i.e., PLZF lu/lu mice and Tcrd $^{-/-}$ -Cd1d $^{-/-}$ -Mr1 $^{-/-}$ mice) are greatly appreciated. On one hand, it is disappointing that these mice cannot be used to test the lack of thymocyte-derived inflammatory effector molecules, on the other hand, it is interesting to learn that there is a redundancy and a strong feedback to maintain the expression of these effector molecules in the thymus. Overall, the study has greatly improved, and the narrative is much smoother and easier to follow, specifically when

switching from “Jedi-TCRab” to the “Jedi-TCRb” model (p. 13, line 284), which was not so in the original manuscript.

We thank the reviewer for the constructive suggestions that allowed us to improve the manuscript and for the positive assessment of the changes. As we agree with the reviewer that the investigation of CD8 T cell tolerance to IFN γ was an important addition, we now show these results in a separate figure (new figure 8). To comply with the figure count guidelines, we now also merged figures 5 and 6 into one figure.

I remain highly enthusiastic about this study, but I have a few minor points to suggest.

1. The results in the new Extended Data 9a, 9b are quite intriguing. What is the identity of inflammatory cytokine-producing cells in PLZF-lu/lu mice and innate-like cell-deficient mice? While the in-depth characterization of the cytokine-producing thymocytes in PLZF-KO or Tcrd, CD1d, Mr1-triple KO mice might be beyond the scope of this study, I wonder why the authors presume that they are T lineage cells (p. 26, line 579)? Instead, is there a possibility that they are ILCs? Providing evidence that affirms them as T lineage cells or rephrasing this sentence is recommended.

The reviewer is absolutely right – in addition to T cells, the cytokine producing populations included sizable subsets of TCR-negative cells likely representing ILCs and/or NK cells. While we had detailed enough staining of surface markers for PLZF- but not Tcrd/CD1d/Mr1-deficient mice, even in WT animals some cytokine producing cells were TCR-negative. We added these results to the Supplementary Figure 3b (the former Extended Data Figure 9b) and changed the text as suggested by the reviewer.

2. Regarding the cellular identity of CD8ab-low PD1hi cells (Fig. 6b and Fig. 4b,c,d), the authors mention a possible lineage diversion into “innate-like lineage” cells (page 21, line 462). But it is unclear to me what prompted this idea. Do these cells produce cytokines or express PLZF? Can the authors provide data supporting their hypothesis?

We agree with the reviewer – we do not know if these cells are undergoing negative selection or lineage diversion. Our results provided in response to the previous comments indicate that negative selection rather than lineage diversion is the main mechanism to inactivate autoreactive thymocytes in our system – at least in the setting with physiological timing of TCR expression. As from that perspective CD8ab-low PD1hi cells do not seem to be physiologically relevant, we did not further investigate their fate but removed statements discussing their possible lineage diversion.

3. In Figure 4 b, it would be useful to show the frequency (or FACS plot) of iNKT cells added to the RTOC. While the WT iNKT cell mixture (top) shows that there are no IL4-GFP-producing cells in the culture, it is unclear if there are iNKT cells added to the RTOC. It would be reassuring to see that iNKT cells are present at the same frequency as for the IL4-GFP iNKT cell RTOC.

While we did not stain RTOCs with CD1d tetramers (as that would require dual tetramer staining or splitting a very small sample into two), in the experiment shown in Fig. 4b we had both Jedi (CD45.1⁺) and iNKT cells (CD45.2⁺) (both sorted in purity mode) congenically distinguishable from fetal thymic cells (CD45.1/CD45.2 heterozygous). We have now added a panel to Fig. 4b showing that both WT and IL4-GFP CD45.2⁺ iNKT cells were readily detectable in the RTOCs.

4. The antigen array experiment (Figure 3, additional experiments in the point-by-point response) is quite interesting. I would suggest including the data in the revised manuscript because the lack of autoantibodies in PLZF-KO mice is unexpected (based on the authors' model), but that can be now explained by the intact cytokine production in PLZF-KO mice.

We now added these data to the manuscript as Supplementary Figure 4b.

Final Decision Letter:

Dear Dr. Kreslavsky,

I am delighted to accept your manuscript entitled "Direct presentation of inflammation-associated self-antigens by thymic innate-like T cells induces elimination of autoreactive CD8 thymocytes" for publication in an upcoming issue of Nature Immunology.

Over the next few weeks, your paper will be copyedited to ensure that it conforms to Nature Immunology style. Once your paper is typeset, you will receive an email with a link to choose the appropriate publishing options for your paper and our Author Services team will be in touch regarding any additional information that may be required.

Please note that *Nature Immunology* is a Transformative Journal (TJ). Authors may publish their research with us through the traditional subscription access route or make their paper immediately open access through payment of an article-processing charge (APC). Authors will not be required to make a final decision about access to their article until it has been accepted. Find out more about Transformative Journals.

Your paper will be published online soon after we receive your corrections and will appear in print in

the next available issue.

Also, if you have any spectacular or outstanding figures or graphics associated with your manuscript - though not necessarily included with your submission - we'd be delighted to consider them as candidates for our cover. Simply send an electronic version (accompanied by a hard copy) to us with a possible cover caption enclosed.

If you have not already done so, we strongly recommend that you upload the step-by-step protocols used in this manuscript to protocols.io. protocols.io is an open online resource that allows researchers to share their detailed experimental know-how. All uploaded protocols are made freely available and are assigned DOIs for ease of citation. Protocols can be linked to any publications in which they are used and will be linked to from your article. You can also establish a dedicated workspace to collect all your lab Protocols. By uploading your Protocols to protocols.io, you are enabling researchers to more readily reproduce or adapt the methodology you use, as well as increasing the visibility of your protocols and papers. Upload your Protocols at <https://protocols.io>. Further information can be found at <https://www.protocols.io/help/publish-articles>.

Please note that we encourage the authors to self-archive their manuscript (the accepted version before copy editing) in their institutional repository, and in their funders' archives, six months after publication. Nature Portfolio recognizes the efforts of funding bodies to increase access of the research they fund, and strongly encourages authors to participate in such efforts. For information about our editorial policy, including license agreement and author copyright, please visit www.nature.com/ni/about/ed_policies/index.html

Sincerely,

Stephanie Houston, PhD
Senior Editor
Nature Immunology